# Coupled Data and Measurement Space Dynamics for Enhanced Diffusion Posterior Sampling

**Shayan Mohajer Hamidi**
Stanford University
smohajer@stanford.edu

**Ben Liang**
University of Toronto
linag@ece.utoronto.ca

**En-Hui Yang**
University of Waterloo
ehyang@uwaterloo.ca

## Abstract

Inverse problems, where the goal is to recover an unknown signal from noisy or incomplete measurements, are central to applications in medical imaging, remote sensing, and computational biology. Diffusion models have recently emerged as powerful priors for solving such problems. However, existing methods either rely on projection-based techniques that enforce measurement consistency through heuristic updates, or they approximate the likelihood $p(\boldsymbol{y} \mid \boldsymbol{x})$, often resulting in artifacts and instability under complex or high-noise conditions. To address these limitations, we propose a novel framework called *coupled data and measurement space diffusion posterior sampling* (C-DPS), which eliminates the need for constraint tuning or likelihood approximation. C-DPS introduces a forward stochastic process in the measurement space $\{\boldsymbol{y}_t\}$, evolving in parallel with the data-space diffusion $\{\boldsymbol{x}_t\}$, which enables the derivation of a closed-form posterior $p(\boldsymbol{x}_{t-1} \mid \boldsymbol{x}_t, \boldsymbol{y}_{t-1})$. This coupling allows for accurate and recursive sampling based on a well-defined posterior distribution. Empirical results demonstrate that C-DPS consistently outperforms existing baselines, both qualitatively and quantitatively, across multiple inverse problem benchmarks.

## 1 Introduction

Inverse problems, where the goal is to recover an unknown signal $\boldsymbol{x}_0$ from noisy or incomplete measurements $\boldsymbol{y}$, arise in a wide range of applications, including medical imaging [1, 2], remote sensing [3, 4], and audio signal processing [5, 6]. Mathematically, these problems are often modeled as $\boldsymbol{y} = \mathbf{A}\boldsymbol{x}_0 + \mathbf{n}$, where $\mathbf{A}$ is a known forward operator and $\mathbf{n}$ represents measurement noise. Inverse problems are inherently ill-posed, meaning that, without additional constraints, infinitely many solutions may satisfy the given measurements $\boldsymbol{y}$. As such, incorporating prior knowledge about the underlying signal and the noise model is essential for reliable reconstruction.

One principled approach to addressing this uncertainty is to treat inverse problems in a Bayesian framework, where the goal becomes sampling from the posterior distribution $p(\boldsymbol{x}_0|\boldsymbol{y})$. However, accurate and efficient posterior sampling remains a central challenge, particularly when the forward operator $\mathbf{A}$ is ill-conditioned or the measurement noise is significant [7]. Traditional sampling-based methods, such as Markov chain Monte Carlo (MCMC), often struggle with high-dimensional spaces or require careful tuning of proposal distributions [8]. More recently, diffusion-based techniques have emerged as powerful generative models for high-dimensional data, and several works have adapted diffusion processes to inverse problems by combining a learned data prior $p(\boldsymbol{x}_0)$ with a measurement-consistency term [7, 8, 9, 10, 11, 12, 13, 14, 15, 16, 17, 18, 19].

39th Conference on Neural Information Processing Systems (NeurIPS 2025).

Nevertheless, in standard diffusion-based frameworks, the measurement information must often be *retro-fitted* into the prior $p(\boldsymbol{x}_0)$ via projection-based techniques [8, 9, 10, 11, 12, 13, 14, 15] or approximation of the likelihood $p(\boldsymbol{y}|\boldsymbol{x})$ [7, 16, 17, 18, 19] (see Appendix A for details). Since these methods modify the iterative processes in data and measurement space in an uncoordinated manner, there is no guarantee that the generated samples lie on a valid data manifold while also being consistent with the observed measurements. This often leads to suboptimal reconstructions: the generated samples may either exhibit visual artifacts due to drifting off the data manifold, or fail to resemble the original measurements due to poor measurement fidelity. Examples of both failure modes are illustrated in Figure 1.

To address the limitations of conventional approaches, we propose C-DPS (**c**oupled data and measurement space **d**iffusion **p**osterior **s**ampling), a novel framework that extends the standard diffusion model by introducing a second, parallel diffusion process in the measurement space, alongside the conventional diffusion in the data space. While the data space process follows the traditional forward diffusion from an unknown signal $\boldsymbol{x}_0$, the measurement space process begins with the known observation $\boldsymbol{y}_0$ and progressively injects noise. This symmetric treatment mirrors the dynamics applied to $\boldsymbol{x}_0$, tightly coupling the two spaces throughout the diffusion trajectory. This coupling enables the derivation of a closed-form posterior expression for $p(\boldsymbol{x}_{t-1}|\boldsymbol{x}_t, \boldsymbol{y}_{t-1})$, thereby removing the need for ad-hoc measurement terms or learned likelihood approximations commonly used in prior work.

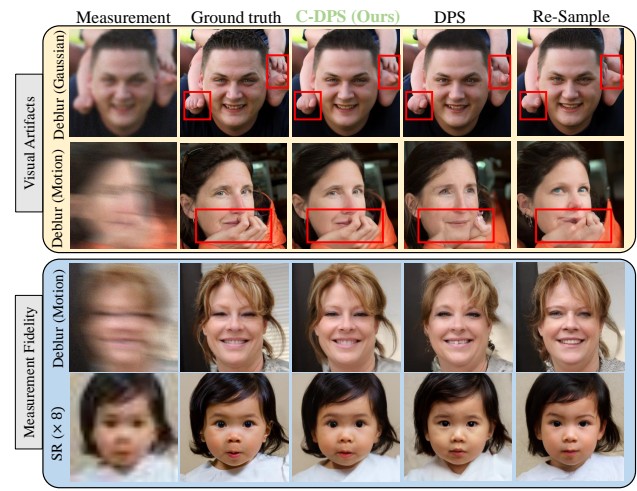

Figure 1: Visualization of reconstructed samples for some linear inverse problems using our proposed C-DPS method, compared with two leading baselines: DPS [7] and Resample [20]. C-DPS exhibits less visual artifacts and improved measurement consistency.

From a Bayesian perspective, the joint diffusion over both $\boldsymbol{x}$ and $\boldsymbol{y}$ defines a coherent generative model over the pair $(\boldsymbol{x}, \boldsymbol{y})$. In doing so, C-DPS naturally integrates both the learned prior and the forward measurement model into a single, unified framework.

• We propose C-DPS, a novel coupled stochastic framework that introduces a parallel diffusion process in the measurement space $\{\boldsymbol{y}_t\}$, evolving jointly with the data-space process $\{\boldsymbol{x}_t\}$. This formulation enables principled posterior sampling in diffusion models by treating $(\boldsymbol{x}, \boldsymbol{y})$ as a unified generative process.

• By explicitly constructing a Markov chain over $\boldsymbol{y}_t$, we derive a closed-form posterior transition $p(\boldsymbol{x}_{t-1} \mid \boldsymbol{x}_t, \boldsymbol{y}_{t-1})$, eliminating the need to approximate or learn the likelihood term $p(\boldsymbol{y} \mid \boldsymbol{x}_t)$. This allows for direct integration of the measurement model into the sampling procedure, ensuring consistent Bayesian updates at every diffusion step.

• We develop a scalable and efficient sampling algorithm for C-DPS based on a pre-whitened conjugate gradient solver. This matrix-free implementation retains the runtime efficiency of conventional DPS methods, despite the added complexity of coupled data-measurement diffusion.

• We validate C-DPS through extensive experiments on standard benchmarks, including FFHQ [21] and ImageNet [22]. Our method achieves state-of-the-art performance across multiple inverse problem settings—such as inpainting, deblurring, and super-resolution—both qualitatively and quantitatively.

**Notation**: Scalars are represented by non-bold letters, (e.g., $a$ or $A$), vectors by bold lowercase letters (e.g., $\mathbf{a}$), and matrices by bold uppercase letters (e.g., $\mathbf{A}$). The real axis is denoted by $\mathbb{R}$. The symbols $\mathbf{0}$ and $\mathbf{I}$ represent the zero vector and the identity matrix, respectively.

## 2 Background and Preliminaries

### 2.1 Diffusion Models

Diffusion models are built upon two key components: a forward noising process and a backward denoising process. These processes operate as Markov chains that progressively perturb and then recover data distributions. In the discrete formulation [23], the forward process gradually adds Gaussian noise to the data according to a predefined variance schedule $\{\beta_t\}_{t=1}^T$, and is described as

$$p(\boldsymbol{x}_{1:T}|\boldsymbol{x}_0) = \prod_{k=1}^T p(\boldsymbol{x}_t|\boldsymbol{x}_{t-1}), \tag{1a}$$

$$p(\boldsymbol{x}_t|\boldsymbol{x}_{t-1}) = \mathcal{N}(\sqrt{1-\beta_t}\boldsymbol{x}_{t-1}, \beta_t\mathbf{I}), \tag{1b}$$

where $\boldsymbol{x}_t \in \mathbb{R}^d$ and $\beta_t \in (0,1)$ is a monotonically increasing sequence controlling the rate of noise addition over time. Since each transition $p(\boldsymbol{x}_t|\boldsymbol{x}_{t-1})$ follows a linear Gaussian model, the marginal distribution $p(\boldsymbol{x}_t|\boldsymbol{x}_0)$ remains Gaussian, given by

$$p(\boldsymbol{x}_t|\boldsymbol{x}_0) \sim \mathcal{N}(\sqrt{\bar{\alpha}_t}\boldsymbol{x}_0, (1-\bar{\alpha}_t)\mathbf{I}), \tag{2}$$

where $\alpha_t = 1 - \beta_t$ and $\bar{\alpha}_t = \prod_{j=1}^t \alpha_j$ are derived from the variance schedule [23].

To generate data samples, a neural network $\boldsymbol{s}_\theta(\boldsymbol{x}_t, t)$ is trained to approximate the score function $\nabla_{\boldsymbol{x}_t} \log p(\boldsymbol{x}_t)$. The reverse denoising process is also formulated as a Markov chain, where the model iteratively reconstructs the data distribution. The backward process is expressed as

$$p_\theta(\boldsymbol{x}_{t-1}|\boldsymbol{x}_t) = \mathcal{N}\big(\boldsymbol{\mu}_\theta(\boldsymbol{x}_t), \boldsymbol{\Sigma}_\theta\big), \tag{3a}$$

$$\boldsymbol{\mu}_\theta(\boldsymbol{x}_t) = \frac{\sqrt{\alpha_t}(1-\bar{\alpha}_{t-1})}{1-\bar{\alpha}_t}\boldsymbol{x}_t + \frac{\sqrt{\bar{\alpha}_{t-1}}\beta_t}{1-\bar{\alpha}_t}\hat{\boldsymbol{x}}_0(\boldsymbol{x}_t), \tag{3b}$$

$$\boldsymbol{\Sigma}_\theta = \beta_t \frac{1-\bar{\alpha}_{t-1}}{1-\bar{\alpha}_t}\mathbf{I}, \tag{3c}$$

where $\hat{\boldsymbol{x}}_0(\boldsymbol{x}_t)$ is the predicted initial state of the data computed using Tweedie's formula [24]:

$$\hat{\boldsymbol{x}}_0(\boldsymbol{x}_t) = \frac{\boldsymbol{x}_t + (1-\bar{\alpha}_t)\boldsymbol{s}_\theta(\boldsymbol{x}_t, t)}{\sqrt{\bar{\alpha}_t}}. \tag{4}$$

### 2.2 Diffusion-Based Inverse Problem Solving

We focus on linear inverse problems, where the goal is to reconstruct an unknown signal $\boldsymbol{x}_0 \in \mathbb{R}^d$ from noisy, incomplete measurements $\boldsymbol{y} \in \mathbb{R}^m$, governed by the linear model $\boldsymbol{y} = \mathbf{A}\boldsymbol{x}_0 + \mathbf{n}$, where $\mathbf{A} \in \mathbb{R}^{m \times d}$ is a known measurement operator, and $\mathbf{n} \sim \mathcal{N}(\mathbf{0}, \boldsymbol{\Sigma_n})$ denotes additive Gaussian noise with known covariance $\boldsymbol{\Sigma_n}$. This leads to a Gaussian likelihood $p(\boldsymbol{y}|\boldsymbol{x}_0) = \mathcal{N}(\mathbf{A}\boldsymbol{x}_0, \boldsymbol{\Sigma_n})$.

Recent advances in applying diffusion models to inverse problems can be broadly categorized into two methodological paradigms: projection-based and likelihood-based approaches. These methods are summarized in Algorithm 1, and a comprehensive discussion is provided in Appendix A. We briefly describe each paradigm below to set the stage for our proposed method.

• **Projection-based approaches.** These approaches iteratively project the generated samples onto the feasible set defined by the measurement $\boldsymbol{y}$. At each reverse diffusion step, the generated sample $\boldsymbol{x}_t$ is updated using a projection operator $\mathcal{P}_{\boldsymbol{y}}$, such as

$$\boldsymbol{x}_t \leftarrow \mathcal{P}_{\boldsymbol{y}}(\boldsymbol{x}_t) := \arg\min_{\boldsymbol{z}} \|\boldsymbol{z} - \boldsymbol{x}_t\|^2 \quad \text{subject to} \quad \|\mathbf{A}\boldsymbol{z} - \boldsymbol{y}\|^2 \leq \epsilon,$$

or in simpler forms (e.g., for noiseless measurements),

$$\boldsymbol{x}_t \leftarrow \boldsymbol{x}_t - \mathbf{A}^\top(\mathbf{A}\mathbf{A}^\top)^{-1}(\mathbf{A}\boldsymbol{x}_t - \boldsymbol{y}).$$

This family includes methods such as ILVR [10], DDRM [14], and DSG [11], which leverage such projections to maintain measurement consistency throughout sampling.

| **Algorithm 1** Baseline methods | **Algorithm 2** C-DPS (one reverse step, clean form) |
|---|---|
| **Input:** # time steps $T$, $\boldsymbol{y}$, noise schedule $\{\beta_t\}$, measurement $\mathbf{A}$, $\{\tilde{\sigma}_t\}$. | **Input:** # steps $T$, measurements $\boldsymbol{y}$, schedule $\{\beta_t\}$, operator $\boldsymbol{A}$, noise covariance $\boldsymbol{\Sigma_n}$ |
| 1: $\boldsymbol{x}_N \sim \mathcal{N}(\mathbf{0}, \mathbf{I})$ | 1: Generate $\{\boldsymbol{y}_t\}_{t=0}^{T}$ as in Equation (5); draw $\boldsymbol{x}_N \sim \mathcal{N}(\mathbf{0}, \boldsymbol{I})$ |
| 2: **for** $t = T-1, T-2, \ldots, 0$ **do** | 2: **for** $t = T-1, T-2, \ldots, 0$ **do** |
| 3: $\quad \hat{\boldsymbol{s}} \leftarrow \boldsymbol{s}_\theta(\boldsymbol{x}_t, t)$ | 3: $\quad \hat{\boldsymbol{s}} \leftarrow \boldsymbol{s}_\theta(\boldsymbol{x}_t, t)$ |
| 4: $\quad \hat{\boldsymbol{x}}_0(\boldsymbol{x}_t) \leftarrow \frac{1}{\sqrt{\bar{\alpha}_t}}\left(\boldsymbol{x}_t + (1 - \bar{\alpha}_t)\hat{\boldsymbol{s}}\right)$ | 4: $\quad \boldsymbol{\Sigma}_{\boldsymbol{y}\mid\boldsymbol{x}} \leftarrow \bar{\alpha}_{t-1}\boldsymbol{\Sigma_n} + (1-\bar{\alpha}_{t-1})\boldsymbol{I}, \quad c_t \leftarrow \frac{1-\beta_t}{\beta_t}$ |
| 5: $\quad \boldsymbol{\mu}_\theta(\boldsymbol{x}_t) \leftarrow \frac{\sqrt{\alpha_t}(1-\bar{\alpha}_{t-1})}{1-\bar{\alpha}_t}\boldsymbol{x}_t + \frac{\sqrt{\bar{\alpha}_{t-1}}\beta_t}{1-\bar{\alpha}_t}\hat{\boldsymbol{x}}_0(\boldsymbol{x}_t)$. | 5: $\quad \boldsymbol{b}_{t-1} \leftarrow (1 - \bar{\alpha}_{t-1})\,\boldsymbol{A}\,\hat{\boldsymbol{s}}$ |
| 6: $\quad \boldsymbol{z} \sim \mathcal{N}(\mathbf{0}, \mathbf{I})$. | 6: $\quad$ Define the matrix–free precision operator $\Lambda_t \boldsymbol{u} \leftarrow c_t\,\boldsymbol{u} + \boldsymbol{A}^\top \boldsymbol{\Sigma}_{\boldsymbol{y}\mid\boldsymbol{x}}^{-1}\boldsymbol{A}\,\boldsymbol{u}.$ |
| 7: $\quad \boldsymbol{x}'_{t-1} \leftarrow \boldsymbol{\mu}_\theta(\boldsymbol{x}_t) + \tilde{\sigma}_t \boldsymbol{z}$ | 7: $\quad$ Solve for $\boldsymbol{\mu}_{\text{post}}$ with CG |
| 8: $\quad \boldsymbol{x}_{t-1}$ is obtained from $\boldsymbol{x}'_{t-1}$ by either a projection step [8, 9, 10, 11, 12, 13, 14, 15] or likelihood approximation [7, 16, 17, 18, 19]. | 8: $\quad \boldsymbol{v} \leftarrow \text{PW-CG}(\Lambda_t, \boldsymbol{A}, \boldsymbol{\Sigma}_{\boldsymbol{y}\mid\boldsymbol{x}}^{-1}, c_t)$ (Algorithm 3) |
| 9: **end for** | 9: $\quad$ **Update:** $\boldsymbol{x}_{t-1} \leftarrow \boldsymbol{\mu}_{\text{post}} + \boldsymbol{v}$ |
| **Output:** $\boldsymbol{x}_0$. | 10: **end for** |
| | **Output:** $\boldsymbol{x}_0$ |

• **Likelihood-based approaches.** These methods aim to recover the unknown signal $\boldsymbol{x}_0$ by approximately sampling from the posterior distribution $p(\boldsymbol{x}_0 \mid \boldsymbol{y})$, where $\boldsymbol{y}$ denotes the measurement. Using Bayes' rule, the posterior can be expressed as

$$p(\boldsymbol{x}_0 \mid \boldsymbol{y}) \propto p(\boldsymbol{y} \mid \boldsymbol{x}_0)p(\boldsymbol{x}_0).$$

Here, $p(\boldsymbol{x}_0)$ is modeled using a diffusion model trained on clean data, and $p(\boldsymbol{y} \mid \boldsymbol{x}_0)$ represents the likelihood induced by the forward measurement process (e.g., a linear operator with additive noise).

Since the likelihood is typically intractable to evaluate directly in diffusion models, several methods approximate it or its gradient. For example, DPS [7] and related methods [16, 17, 18, 19] use Tweedie's formula or Bayesian denoising estimators to approximate $\nabla_{\boldsymbol{x}_t} \log p(\boldsymbol{y} \mid \boldsymbol{x}_t)$ at intermediate timesteps $t$. This approximation is then used to adjust the reverse dynamics by modifying the model's predicted mean. Specifically, the mean used in the reverse step is replaced by $\tilde{\boldsymbol{\mu}}_\theta(\boldsymbol{x}_t, \boldsymbol{y}) = \boldsymbol{\mu}_\theta(\boldsymbol{x}_t) + \lambda\nabla_{\boldsymbol{x}_t} \log p(\boldsymbol{y} \mid \boldsymbol{x}_t)$, where $\boldsymbol{\mu}_\theta(\boldsymbol{x}_t)$ is the original predicted mean, and $\lambda$ is a scaling factor.

## 3 Methodology

### 3.1 Motivation

As discussed in Section 2.2, most existing approaches to solving inverse problems with diffusion models rely on retro-fitting measurement information into the learned data prior $p(\boldsymbol{x}_0)$. This is typically done either through heuristic projection-based techniques or by approximating the likelihood $p(\boldsymbol{y} \mid \boldsymbol{x})$. However, both approaches introduce fundamental limitations: projection-based constraints often require manual tuning and lack theoretical justification, while likelihood approximations can distort the posterior geometry, especially under high noise levels (see Figure 1).

To address these challenges, we propose C-DPS, a principled framework that integrates the measurement process directly into the diffusion dynamics. Central to our idea is an auxiliary forward stochastic process in the measurement space, denoted by $\{\boldsymbol{y}_t\}_{t=0}^{T}$. By coupling the data-space process $\{\boldsymbol{x}_t\}_{t=0}^{T}$ and $\{\boldsymbol{y}_t\}_{t=0}^{T}$, C-DPS enables the derivation of a closed-form posterior distribution $p(\boldsymbol{x}_{t-1} \mid \boldsymbol{x}_t, \boldsymbol{y}_{t-1})$. This eliminates the need for manually tuning constraint terms or approximating the likelihood. Crucially, unlike the data-space diffusion process which requires training a neural network to approximate reverse dynamics, the measurement-space process $\{\boldsymbol{y}_t\}$ is purely forward and analytically defined. It evolves from the observed measurement $\boldsymbol{y}$, not to generate $\boldsymbol{y}_0$, but to propagate measurement information coherently across diffusion steps. This coupling allows C-DPS to incorporate observation structure directly into posterior updates, resulting in more stable, interpretable, and accurate reconstructions across a wide range of inverse problems.

It is worth noting that similar measurement-space diffusions have been discussed conceptually in Appendix I.4 of [25] and further explored in [26] for medical imaging, but neither derives an analytic posterior or provides a tractable sampling rule. In contrast, C-DPS formulates a closed-form posterior $p(\boldsymbol{x}_{t-1} \mid \boldsymbol{x}_t, \boldsymbol{y}_{t-1})$, removing heuristic updates and manual tuning.

In the following subsections, we elaborate how the sequences $\{\boldsymbol{y}_t\}_{t=0}^T$ and $\{\boldsymbol{x}_t\}_{t=0}^T$ are generated.

## 3.2 Constructing a Markov Chain in the Measurement Space

Similarly to the diffusion process in the data space which is defined with Markov chain in Equation (1b), we define a Markov chain $\{\boldsymbol{y}_t\}_{t=0}^T$ that starts from the real-world measurement $\boldsymbol{y}_0$ and progressively adds Gaussian noise at each forward step. Specifically, we let

$$\boldsymbol{y}_0 = \mathbf{A}\boldsymbol{x}_0 + \mathbf{n}, \tag{5a}$$

$$\boldsymbol{y}_t = \sqrt{1 - \beta_t}\,\boldsymbol{y}_{t-1} + \sqrt{\beta_t}\,\boldsymbol{z}_t, \tag{5b}$$

where $\boldsymbol{z}_t \sim \mathcal{N}(\mathbf{0}, \mathbf{I})$ and $\beta_t \in (0, 1)$ determines the noise schedule. The linear operator $\mathbf{A}$ maps the unknown data $\boldsymbol{x}_0$ to the measurement space, and $\mathbf{n}$ is the measurement noise.

**Remark 1** (Choice of noise schedule). *We use the same noise schedule $\{\beta_t\}$ for both the data-space diffusion $\{\boldsymbol{x}_t\}$ and the measurement-space diffusion $\{\boldsymbol{y}_t\}$. This design choice keeps the two processes synchronized, which simplifies the derivation of the backward update $p(\boldsymbol{x}_{t-1} \mid \boldsymbol{x}_t, \boldsymbol{y}_{t-1})$ and avoids introducing additional hyperparameters. Conceptually, using a shared schedule ensures that the same fraction of noise is injected at each time step in both domains, preserving a one-to-one correspondence between $\boldsymbol{x}_t$ and $\boldsymbol{y}_t$. Although it is possible in principle to define separate schedules for the data and measurement spaces, we find that a shared schedule leads to strong empirical performance and a cleaner theoretical formulation.*

From the standard forward-diffusion identity, $\boldsymbol{y}_t$ could be written in terms of $\boldsymbol{y}_0$ as

$$\boldsymbol{y}_t = \sqrt{\bar{\alpha}_t}\,\boldsymbol{y}_0 + \sqrt{1 - \bar{\alpha}_t}\,\boldsymbol{\zeta}, \tag{6}$$

where $\boldsymbol{\zeta} \sim \mathcal{N}(\mathbf{0}, \mathbf{I})$, and $\bar{\alpha}_t$ is defined in Equation (2). Further noting $\boldsymbol{y}_0 = \mathbf{A}\,\mathbf{x}_0 + \mathbf{n}$ with $\mathbf{n} \sim \mathcal{N}(\mathbf{0}, \boldsymbol{\Sigma}_{\mathbf{n}})$, we have

$$\boldsymbol{y}_t = \sqrt{\bar{\alpha}_t}\,(\mathbf{A}\,\mathbf{x}_0 + \mathbf{n}) + \sqrt{1 - \bar{\alpha}_t}\,\boldsymbol{\zeta}. \tag{7}$$

From Equation (7), it is easy to obtain the following conclusion:

**Proposition 1** (Distribution of $\{\boldsymbol{y}_t\}$). *For the sequence $\{\boldsymbol{y}_t\}_{t=0}^T$ defined in Equation (5), the distribution of $\boldsymbol{y}_t$ is a Gaussian whose mean and covariance at step $t$ are given by*

$$\mu_{\boldsymbol{y},t} = \sqrt{\bar{\alpha}_t}\,\mathbf{A}\,\boldsymbol{x}_0, \tag{8a}$$

$$\boldsymbol{\Sigma}_{\boldsymbol{y},t} = \bar{\alpha}_t\,\boldsymbol{\Sigma}_{\mathbf{n}} + (1 - \bar{\alpha}_t)\,\mathbf{I}. \tag{8b}$$

## 3.3 Generating $\{\boldsymbol{x}_t\}$ Consistent with $\{\boldsymbol{y}_t\}$

Given the measurement sequence $\{\boldsymbol{y}_t\}$, we construct a consistent sequence of latent states $\{\boldsymbol{x}_t\}$ that evolves under diffusion while remaining aligned with the observations.

• **Initialization.** We start by sampling

$$\boldsymbol{x}_T \sim \mathcal{N}(\mathbf{0}, \mathbf{I}).$$

• **Backward Recursion.** For each $t$ from $T$ down to 1, we generate $\boldsymbol{x}_{t-1}$ by drawing a sample from $p(\boldsymbol{x}_{t-1} | \boldsymbol{x}_t, \boldsymbol{y}_{t-1})$. Based on the Bayesian update rule for the posterior, we can write[1]

$$p(\boldsymbol{x}_{t-1} \mid \boldsymbol{x}_t, \boldsymbol{y}_{t-1}) \propto p(\boldsymbol{x}_t \mid \boldsymbol{x}_{t-1})\,p(\boldsymbol{y}_{t-1} \mid \boldsymbol{x}_{t-1}). \tag{9}$$

To find $p(\boldsymbol{y}_{t-1} | \boldsymbol{x}_{t-1})$, at an intermediate diffusion step, we use the formula in Equation (7), and replace $\boldsymbol{x}_0$ in this formula with its estimate in terms of $\boldsymbol{x}_{t-1}$ using Equation (4). Thus, we get

$$\boldsymbol{y}_{t-1} = \sqrt{\bar{\alpha}_{t-1}}\left(\mathbf{A}\left[\frac{\boldsymbol{x}_{t-1} + (1 - \bar{\alpha}_{t-1})\boldsymbol{s}_\theta(\boldsymbol{x}_{t-1}, t-1)}{\sqrt{\bar{\alpha}_{t-1}}}\right] + \mathbf{n}\right) + \sqrt{1 - \bar{\alpha}_{t-1}}\,\boldsymbol{\zeta} \tag{10}$$

$$= \mathbf{A}\boldsymbol{x}_{t-1} + \mathbf{A}\big((1 - \bar{\alpha}_{t-1})\boldsymbol{s}_\theta(\boldsymbol{x}_{t-1}, t-1)\big) + \sqrt{\bar{\alpha}_{t-1}}\mathbf{n} + \sqrt{1 - \bar{\alpha}_{t-1}}\boldsymbol{\zeta}. \tag{11}$$

---

[1]Please refer to Remark 2 to see why the prior $p(\boldsymbol{x}_{t-1})$ is dropped from Equation (9).

Since $\mathbf{n} \sim \mathcal{N}(\mathbf{0}, \mathbf{\Sigma_n})$ and $\boldsymbol{\zeta} \sim \mathcal{N}(\mathbf{0}, \mathbf{I})$, $p(\boldsymbol{y}_{t-1}|\boldsymbol{x}_{t-1})$ is Gaussian with the following parameters:

$$p(\boldsymbol{y}_{t-1}|\boldsymbol{x}_{t-1}) \sim \mathcal{N}\Big(\boldsymbol{\mu_{y|x}}, \mathbf{\Sigma_{y|x}}\Big), \tag{12a}$$

$$\text{where} \quad \boldsymbol{\mu_{y|x}} = \mathbf{A}\boldsymbol{x}_{t-1} + (1 - \bar{\alpha}_{t-1})\mathbf{A}\boldsymbol{s}_\theta(\boldsymbol{x}_{t-1}, t - 1), \tag{12b}$$

$$\text{and} \quad \mathbf{\Sigma_{y|x}} = \bar{\alpha}_{t-1}\mathbf{\Sigma_n} + (1 - \bar{\alpha}_{t-1})\mathbf{I}. \tag{12c}$$

Substituting Equation (1b) and Equation (12a) into Equation (9), the posterior $p\big(\boldsymbol{x}_{t-1}|\boldsymbol{x}_t, \boldsymbol{y}_{t-1}\big)$ could be obtained using Bayes' rule and dropping the normalizing constant:

$$p(\boldsymbol{x}_{t-1}|\boldsymbol{x}_t, \boldsymbol{y}_{t-1}) \propto \exp\Big[-\tfrac{1}{2\beta_t}\big\|\boldsymbol{x}_t - \sqrt{1 - \beta_t}\,\boldsymbol{x}_{t-1}\big\|^2 - \tfrac{1}{2}\big(\boldsymbol{y}_{t-1} - \boldsymbol{\mu_{y|x}}\big)^\top \mathbf{\Sigma_{y|x}}^{-1}\big(\boldsymbol{y}_{t-1} - \boldsymbol{\mu_{y|x}}\big)\Big]. \tag{13}$$

The expression in Equation (13) defines the exact un-normalized posterior density. The first term is quadratic in $\boldsymbol{x}_{t-1}$ and arises from the diffusion prior. However, the second term involves the score network $\boldsymbol{s}_\theta(\boldsymbol{x}_{t-1}, t - 1)$, making the conditional mean $\boldsymbol{\mu_{y|x}}$ a nonlinear function of $\boldsymbol{x}_{t-1}$.

To make the posterior distribution in Equation (13) Gaussian, we apply a common approximation in diffusion-based inference and *freeze the score network* at the current iterate:

$$\boldsymbol{s}_\theta(\boldsymbol{x}_{t-1}, t - 1) \quad \longrightarrow \quad \boldsymbol{s}_\theta(\boldsymbol{x}_t, t). \tag{14}$$

This substitution avoids evaluating the score at $\boldsymbol{x}_{t-1}$, which is not yet known during sampling, and instead uses the available point $\boldsymbol{x}_t$. Intuitively, since $\boldsymbol{x}_{t-1}$ and $\boldsymbol{x}_t$ are close for small $\beta_t$, this approximation preserves consistency. While this approximation has been used in prior works [10, 12], we further empirically justify it in the context of our work in Appendix B.

Using Equation (14), $\boldsymbol{\mu_{y|x}}$ becomes affine in $\boldsymbol{x}_{t-1}$,

$$\boldsymbol{\mu_{y|x}} = \mathbf{A}\boldsymbol{x}_{t-1} + \underbrace{\big(1 - \bar{\alpha}_{t-1}\big)\mathbf{A}\,\boldsymbol{s}_\theta(\boldsymbol{x}_t, t)}_{\triangleq\,\boldsymbol{b}_{t-1}}, \tag{15}$$

so the product of the two Gaussians in Equation (9) remains Gaussian $\mathcal{N}\big(\boldsymbol{\mu}_{\text{post}}, \mathbf{\Sigma}_{\text{post}}\big)$. Collecting quadratic and linear terms gives

$$\mathbf{\Sigma}_{\text{post}}^{-1} = \frac{1 - \beta_t}{\beta_t}\mathbf{I} + \mathbf{A}^\top \mathbf{\Sigma_{y|x}}^{-1}\mathbf{A}, \tag{16a}$$

$$\boldsymbol{\mu}_{\text{post}} = \mathbf{\Sigma}_{\text{post}}\Big[\frac{\sqrt{1 - \beta_t}}{\beta_t}\,\boldsymbol{x}_t + \mathbf{A}^\top \mathbf{\Sigma_{y|x}}^{-1}\big(\boldsymbol{y}_{t-1} - \boldsymbol{b}_{t-1}\big)\Big]. \tag{16b}$$

Hence, C-DPS samples from $\mathcal{N}(\boldsymbol{\mu}_{\text{post}}, \mathbf{\Sigma}_{\text{post}})$ at each reverse step. The full procedure is shown in Algorithm 2, with the latent variant, termed LC-DPS, detailed in Appendix C. In addition, in Appendix D, we extend the above approach for the case of non-linear measurement.

**Remark 2.** *Note that the exact form of Equation* (9) *includes the prior term* $p(\boldsymbol{x}_{t-1})$. *In diffusion models, after sufficient noising steps under linear or cosine schedules, the marginal* $p(\boldsymbol{x}_{t-1})$ *is close to* $\mathcal{N}(\mathbf{0}, \boldsymbol{I})$ *since the noise dominates. Including this term multiplies Equation* (13) *by* $\exp(-\tfrac{1}{2}\|\boldsymbol{x}_{t-1}\|_2^2)$, *which adds* $\boldsymbol{I}$ *to the posterior precision, that is* $\Lambda_t \leftarrow \frac{1-\beta_t}{\beta_t}\boldsymbol{I} + \boldsymbol{A}^\top\mathbf{\Sigma_{y|x}}^{-1}\boldsymbol{A} + \boldsymbol{I}$. *Since* $\beta_t \ll 1$, *the extra* $\boldsymbol{I}$ *changes each diagonal by at most a small fraction* $O(\beta_t)$, *which is numerically negligible. We therefore omit* $p(\boldsymbol{x}_{t-1})$ *in Equation* (9) *for clarity.*

## 3.4 Efficient sampling

Our goal is to draw $\boldsymbol{x}_{t-1} \sim p(\boldsymbol{x}_{t-1} \mid \boldsymbol{x}_t, \boldsymbol{y}_{t-1})$ as in Equation (16) without forming dense factorizations. Define the posterior precision operator

$$\Lambda_t = \mathbf{\Sigma}_{\text{post}}^{-1} = c_t\,\boldsymbol{I} + \boldsymbol{A}^\top\mathbf{\Sigma_{y|x}}^{-1}\boldsymbol{A}, \qquad c_t = \tfrac{1-\beta_t}{\beta_t}. \tag{17}$$

Direct Cholesky on dense $\mathbf{\Sigma_{y|x}}$ is $\mathcal{O}(d^3)$ and impractical. We therefore use two matrix-free conjugate-gradient (CG) solves per reverse step, implemented by Algorithms 2 and 3.

**Step 1: mean solve.** Compute $\boldsymbol{\mu}_{\text{post}}$ by solving

$$\Lambda_t\,\boldsymbol{\mu}_{\text{post}} \;=\; c_t\,\boldsymbol{x}_t \;+\; \boldsymbol{A}^\top \boldsymbol{\Sigma}_{\boldsymbol{y}|\boldsymbol{x}}^{-1}\big(\boldsymbol{y}_{t-1} - \boldsymbol{b}_{t-1}\big), \qquad \boldsymbol{b}_{t-1} = (1 - \bar{\alpha}_{t-1})\,\boldsymbol{A}\,\hat{\boldsymbol{s}}, \qquad (18)$$

with $\hat{\boldsymbol{s}} = \boldsymbol{s}_\theta(\boldsymbol{x}_t, t)$.

**Step 2: noise draw (PW-CG).** Draw $\boldsymbol{v} \sim \mathcal{N}(\boldsymbol{0}, \boldsymbol{\Sigma}_{\text{post}})$ by solving $\Lambda_t \boldsymbol{v} = \boldsymbol{z}$ for a synthetic right-hand side $\boldsymbol{z}$ satisfying $\text{cov}(\boldsymbol{z}) = \Lambda_t$. Algorithm 3 shows how to build such a $\boldsymbol{z}$ from two standard Gaussians using a whitening operator.

**Step 3: update.** Set

$$\boldsymbol{x}_{t-1} \;=\; \boldsymbol{\mu}_{\text{post}} \;+\; \boldsymbol{v}. \qquad (19)$$

**Pre-whitened conjugate gradient (PW-CG).** Let $\boldsymbol{W}$ satisfy $\boldsymbol{W}^\top \boldsymbol{W} = \boldsymbol{\Sigma}_{\boldsymbol{y}|\boldsymbol{x}}^{-1}$. For many structured noise models a closed form exists. For example, if $\boldsymbol{\Sigma}_{\boldsymbol{n}} = \sigma^2 \boldsymbol{I}$, then

$$\boldsymbol{\Sigma}_{\boldsymbol{y}|\boldsymbol{x}} \;=\; \big(\bar{\alpha}_{t-1}\sigma^2 + 1 - \bar{\alpha}_{t-1}\big)\boldsymbol{I} \;\triangleq\; \gamma_t \boldsymbol{I}, \qquad \boldsymbol{W} = \gamma_t^{-1/2}\boldsymbol{I}. \qquad (20)$$

Define $\tilde{\boldsymbol{A}} = \boldsymbol{W}\boldsymbol{A}$ so that $\Lambda_t = c_t \boldsymbol{I} + \tilde{\boldsymbol{A}}^\top \tilde{\boldsymbol{A}}$. PW-CG samples

$$\boldsymbol{\varepsilon}_1 \sim \mathcal{N}(\boldsymbol{0}, \boldsymbol{I}_d), \qquad \boldsymbol{\varepsilon}_2 \sim \mathcal{N}(\boldsymbol{0}, \boldsymbol{I}_m), \qquad \boldsymbol{z} = \sqrt{c_t}\,\boldsymbol{\varepsilon}_1 + \tilde{\boldsymbol{A}}^\top \boldsymbol{\varepsilon}_2, \qquad (21)$$

which gives $\text{cov}(\boldsymbol{z}) = \Lambda_t$. Solving $\Lambda_t \boldsymbol{v} = \boldsymbol{z}$ by CG then yields $\boldsymbol{v} \sim \mathcal{N}(\boldsymbol{0}, \Lambda_t^{-1}) = \mathcal{N}(\boldsymbol{0}, \boldsymbol{\Sigma}_{\text{post}})$.

**Practicalities and cost.** Each CG iteration applies $\boldsymbol{A}$ and $\boldsymbol{A}^\top$ once. A diagonal preconditioner $\boldsymbol{P}_t = \text{diag}(\Lambda_t)$ works well. CG requires $\mathcal{O}(\sqrt{\kappa})$ iterations with $\kappa$ the condition number of $\boldsymbol{P}_t^{-1}\Lambda_t$. Empirically $\kappa < 50$ across our tasks, so the PW-CG cost per step is $\mathcal{O}\big(\sqrt{\kappa} \cdot \text{nnz}(\boldsymbol{A})\big)$. In our setup, score evaluation dominates the runtime, while the linear solves add a small overhead.

To contextualize this cost, we note that evaluating the score network $\boldsymbol{s}_\theta(\boldsymbol{x}_t, t)$—typically implemented as a U-Net or Transformer—takes approximately 100–200 milliseconds per step on a modern GPU (e.g., NVIDIA V100 or P100) for a $256 \times 256$ input. In contrast, our linear solver runs significantly faster, often requiring under 10 milliseconds. Its contribution to the total runtime is therefore negligible compared with the dominant cost of score evaluation.

---

**Algorithm 3** PW-CG (draw $\boldsymbol{v} \sim \mathcal{N}(\boldsymbol{0}, \boldsymbol{\Sigma}_{\text{post}})$ without Cholesky)

---

**Input:** precision operator $\Lambda_t$, matrix $\boldsymbol{A}$, action of $\boldsymbol{\Sigma}_{\boldsymbol{y}|\boldsymbol{x}}^{-1}$ (or its square root), scalar $c_t$
  1: Draw $\boldsymbol{\varepsilon}_1 \sim \mathcal{N}(\boldsymbol{0}, \boldsymbol{I}_d), \quad \boldsymbol{\varepsilon}_2 \sim \mathcal{N}(\boldsymbol{0}, \boldsymbol{I}_m), \quad$ independent
  2: (Prewhiten) define a whitening operator $\boldsymbol{W}$ with $\boldsymbol{W}^\top \boldsymbol{W} = \boldsymbol{\Sigma}_{\boldsymbol{y}|\boldsymbol{x}}^{-1}$, and set $\tilde{\boldsymbol{A}} = \boldsymbol{W}\boldsymbol{A}$

  3: **Form** $\boldsymbol{z} \leftarrow \sqrt{c_t}\,\boldsymbol{\varepsilon}_1 + \tilde{\boldsymbol{A}}^\top \boldsymbol{\varepsilon}_2$
  4: **Solve** $\boldsymbol{v} \leftarrow \text{CG-solve}(\Lambda_t, \boldsymbol{z})$             (that is, solve $\Lambda_t \boldsymbol{v} = \boldsymbol{z}$)
**Output:** $\boldsymbol{v}$             (then $\text{cov}(\boldsymbol{v}) = \Lambda_t^{-1} = \boldsymbol{\Sigma}_{\text{post}}$)

---

## 4 Experiments

### 4.1 Quantitative Results

● **Experimental setup.** Following prior work [7, 27], we evaluate our method on the FFHQ $256 \times 256$ [21] and ImageNet $256 \times 256$ [22] datasets, using 1,000 validation images from each. All images are normalized to the $[0, 1]$ range. To ensure a fair comparison, we adopt the same experimental settings used in [7] across all evaluated methods. The measurement data is corrupted with additive Gaussian noise of zero mean and standard deviation $\sigma = 0.05$. During inference, we use a fixed number of reverse diffusion steps $T = 1000$, following standard practice in the literature. For score estimation, we utilize the pre-trained model from [7] for FFHQ and the model from [28] for ImageNet.

Table 1: Quantitative results on the 1k validation sets of FFHQ $256 \times 256$ and ImageNet $256 \times 256$. **Bold** and underline indicate the best and second-best results, respectively. Green and red denote performance improvements and degradations relative to the best baseline.

| Dataset | Method | Inpaint (Random) | | | Inpaint (Box) | | | Deblur (Gaussian) | | | Deblur (Motion) | | | SR ($4\times$) | | |
|---|---|---|---|---|---|---|---|---|---|---|---|---|---|---|---|---|
| | | FID↓ | LPIPS↓ | SSIM↑ | FID↓ | LPIPS↓ | SSIM↑ | FID↓ | LPIPS↓ | SSIM↑ | FID↓ | LPIPS↓ | SSIM↑ | FID↓ | LPIPS↓ | SSIM↑ |
| | | | | | | | | *Pixel-Domain Methods* | | | | | | | | |
| FFHQ | DPS | 21.19 | 0.212 | 0.851 | 33.12 | 0.168 | 0.873 | 44.05 | 0.257 | 0.811 | 39.92 | 0.242 | 0.859 | 39.35 | 0.214 | 0.852 |
| | ΠGDM | 21.27 | 0.221 | 0.840 | 34.79 | 0.179 | 0.860 | 40.21 | 0.242 | 0.825 | 33.24 | 0.221 | 0.887 | 34.98 | 0.202 | 0.854 |
| | DDRM | 69.71 | 0.587 | 0.319 | 42.93 | 0.204 | 0.869 | 74.92 | 0.332 | 0.767 | – | – | – | 62.15 | 0.294 | 0.835 |
| | MCG | 29.26 | 0.286 | 0.751 | 40.11 | 0.309 | 0.703 | 101.2 | 0.340 | 0.051 | – | – | – | 87.64 | 0.520 | 0.559 |
| | ILVR | 25.74 | 0.231 | 0.672 | 37.24 | 0.175 | 0.854 | 52.93 | 0.297 | 0.784 | – | – | – | 47.59 | 0.253 | 0.844 |
| | ReSample | 21.25 | 0.202 | 0.847 | 33.51 | 0.160 | 0.866 | 37.05 | 0.251 | 0.822 | 31.19 | 0.220 | 0.892 | 30.48 | 0.204 | 0.851 |
| | PnP-ADMM | 123.6 | 0.692 | 0.325 | 151.9 | 0.406 | 0.642 | 90.42 | 0.441 | 0.812 | – | – | – | 66.52 | 0.353 | 0.855 |
| | Score-SDE | 76.54 | 0.612 | 0.437 | 60.06 | 0.331 | 0.678 | 109.0 | 0.403 | 0.109 | – | – | – | 96.72 | 0.563 | 0.617 |
| | ADMM-TV | 181.5 | 0.463 | 0.784 | 68.94 | 0.322 | 0.814 | 186.7 | 0.507 | 0.801 | – | – | – | 110.6 | 0.428 | 0.803 |
| | PnP-DM | 21.15 | 0.208 | 0.858 | 32.21 | 0.155 | 0.877 | 41.92 | 0.251 | 0.816 | 37.21 | 0.233 | 0.871 | 36.21 | 0.210 | 0.859 |
| | DAPS | 20.77 | 0.201 | 0.869 | 29.44 | 0.144 | 0.882 | 35.84 | 0.242 | 0.830 | 30.26 | 0.215 | 0.911 | 30.15 | 0.202 | 0.854 |
| | DMPlug | **20.12** | 0.197 | 0.877 | 27.12 | 0.140 | **0.888** | 32.44 | 0.230 | 0.830 | 27.55 | 0.210 | **0.925** | 28.55 | 0.199 | **0.862** |
| | C-DPS | 20.14 | **0.195** | **0.881** | **26.33** | **0.132** | 0.871 | **32.24** | 0.238 | **0.832** | **27.29** | 0.217 | 0.921 | **28.41** | **0.196** | 0.855 |
| ImageNet | DPS | 35.87 | 0.303 | 0.739 | 38.82 | 0.262 | 0.794 | 62.72 | 0.444 | 0.706 | 56.08 | 0.389 | 0.634 | 50.66 | 0.337 | 0.781 |
| | ΠGDM | 41.82 | 0.356 | 0.705 | 42.26 | 0.284 | 0.752 | 59.79 | 0.425 | **0.717** | 54.18 | 0.373 | **0.675** | 54.26 | 0.352 | 0.765 |
| | DDRM | 114.9 | 0.665 | 0.403 | 45.95 | 0.245 | **0.814** | 63.02 | 0.427 | 0.705 | – | – | – | 59.57 | 0.339 | 0.790 |
| | MCG | 39.19 | 0.414 | 0.546 | 39.74 | 0.330 | 0.633 | 95.04 | 0.550 | 0.441 | – | – | – | 144.5 | 0.637 | 0.227 |
| | ILVR | 38.27 | 0.372 | 0.656 | 39.51 | 0.278 | 0.726 | 71.24 | 0.421 | 0.662 | – | – | – | 95.3 | 0.532 | 0.498 |
| | ReSample | 33.47 | 0.289 | 0.730 | 39.54 | 0.259 | 0.799 | 61.24 | 0.439 | 0.708 | 55.76 | 0.370 | 0.637 | 49.19 | 0.339 | 0.777 |
| | PnP-ADMM | 114.7 | 0.677 | 0.300 | 78.24 | 0.367 | 0.657 | 100.6 | 0.519 | 0.669 | – | – | – | 97.27 | 0.433 | 0.761 |
| | Score-SDE | 127.1 | 0.659 | 0.517 | 54.07 | 0.354 | 0.612 | 120.3 | 0.667 | 0.436 | – | – | – | 170.7 | 0.701 | 0.256 |
| | ADMM-TV | 189.3 | 0.510 | 0.676 | 87.69 | 0.319 | 0.785 | 155.7 | 0.588 | 0.634 | – | – | – | 130.9 | 0.523 | 0.679 |
| | PnP-DM | 34.92 | 0.296 | 0.736 | 37.67 | 0.258 | 0.797 | 61.06 | 0.433 | 0.707 | 55.33 | 0.372 | 0.636 | 50.10 | 0.336 | 0.786 |
| | DAPS | 33.94 | 0.282 | 0.741 | 35.46 | 0.248 | 0.801 | 60.12 | 0.419 | 0.709 | 54.82 | 0.365 | 0.639 | 49.62 | 0.333 | 0.789 |
| | DMPlug | 32.85 | 0.226 | 0.748 | 34.28 | 0.247 | 0.804 | 57.42 | 0.407 | 0.714 | 53.13 | 0.366 | 0.642 | 48.96 | 0.324 | 0.793 |
| | C-DPS | **32.37** | **0.214** | **0.755** | **33.24** | **0.236** | 0.807 | **56.36** | **0.391** | 0.712 | **52.06** | **0.352** | 0.644 | **47.30** | **0.316** | **0.795** |
| | | | | | | | | *Latent-Domain Methods* | | | | | | | | |
| FFHQ | PSLD | 47.21 | 0.221 | 0.809 | 43.02 | 0.158 | 0.813 | 89.51 | 0.316 | 0.631 | 96.15 | 0.336 | 0.678 | 74.36 | 0.287 | 0.649 |
| | ReSample | 39.85 | 0.140 | 0.746 | 53.21 | 0.184 | 0.749 | 71.69 | 0.255 | 0.714 | **44.72** | **0.198** | **0.823** | 93.18 | 0.392 | 0.594 |
| | RLSD | 38.25 | 0.142 | 0.808 | 44.08 | 0.153 | 0.812 | 68.92 | 0.244 | 0.750 | 49.10 | 0.284 | 0.810 | 61.37 | 0.203 | 0.774 |
| | LC-DPS | **36.67** | **0.137** | **0.815** | 42.11 | **0.144** | **0.821** | **65.71** | **0.232** | **0.759** | 46.57 | 0.272 | 0.819 | 58.41 | **0.197** | **0.787** |
| ImageNet | PSLD | 83.21 | 0.337 | 0.783 | 146.53 | 0.465 | 0.694 | 91.39 | 0.390 | 0.688 | 124.67 | 0.511 | 0.594 | 97.45 | 0.360 | **0.694** |
| | ReSample | 59.87 | 0.143 | 0.756 | 127.84 | 0.262 | 0.631 | **65.35** | 0.254 | 0.703 | 66.89 | 0.227 | 0.738 | 113.42 | 0.370 | 0.576 |
| | RLSD | 60.44 | 0.141 | 0.787 | 130.16 | 0.259 | 0.705 | 67.34 | 0.249 | 0.712 | 68.43 | 0.236 | 0.735 | 92.73 | 0.317 | 0.680 |
| | LC-DPS | 60.02 | **0.140** | **0.790** | **113.67** | **0.252** | **0.714** | 67.21 | **0.244** | **0.718** | **63.94** | **0.216** | **0.748** | 88.15 | **0.288** | 0.683 |

The measurement models used in our experiments follow those described in [7]: ($i$) Inpainting: For box-type inpainting, a central $128 \times 128$ region is masked out; for random-type inpainting, 92% of pixels (across all RGB channels) are randomly masked. ($ii$) Super-resolution (SR): Low-resolution measurements are obtained by applying bicubic downsampling. ($iii$) Gaussian blur: A Gaussian kernel of size $61 \times 61$ with a standard deviation of 5.0 is convolved with the image. ($iv$) Motion blur: Motion blur kernels of size $61 \times 61$ are generated using the code from [29], with an intensity parameter of 0.5. These kernels are convolved with the ground-truth images to produce the measurements.

● **Benchmark methods.** For pixel-based diffusion model experiments, we use DPS [7], ΠGDM [8], DDRM [14], MCG [30], ILVR [10], ReSample [20], PnP-ADMM [31] , Score-SDE [25], total-variation sparsity regularized optimization method (ADMM-TV), PnP-DM [32], DAPS [33], and DMPlug [34]. Furthermore, for latent diffusion experiments, we compare LC-DPS with PSLD [35], the latent version of the ReSample, and RLSD [36]. To ensure a fair comparison, all methods use the same pre-trained score function.

● **Evaluation metrics.** To evaluate the performance of different methods, we adopt the metrics used in [7]: ($i$) learned perceptual image patch similarity (LPIPS) [37], ($ii$) Frechet inception distance (FID) [38], and ($iii$) structure similarity index measure (SSIM) [39]. In addition, we provide results for peak signal-to-noise ratio (PSNR) in Appendix F. All experiments are conducted using a single NVIDIA P100 GPU with 12 GB of memory.

● **Results.** The quantitative results for both datasets are presented in Table 1. Across nearly all tasks, C-DPS (or LC-DPS) outperforms the baseline methods. In a few settings, DMPlug attains scores that are competitive with ours. We emphasize that although DMPlug can be close in accuracy, its runtime is approximately $4\times$ slower than C-DPS, which limits its practical utility; see Appendix F.1 for run-time details. We also conduct a similar set of experiments under varying Gaussian noise levels, presented in Appendix H.1.

## 4.2 Visual Comparison

In this subsection, we provide a visual comparison of reconstructions produced by C-DPS, DPS, and Re-Sample. We choose to include DPS and ReSample in these visual comparisons because, as shown in Table 1, they achieve the strongest quantitative performance among the evaluated baselines. We randomly select five images from the FFHQ test set and corrupt them using the measurement models described in Section 4.1, with minor adjustments to enhance visual clarity: for super-resolution, we apply an $8\times$ downsampling factor, and for random inpainting, 95% of the pixels are masked. The reconstructed images are presented in Figure 2, where each row corresponds to a different measurement setting. More qualitative results are presented in Appendix H.2.

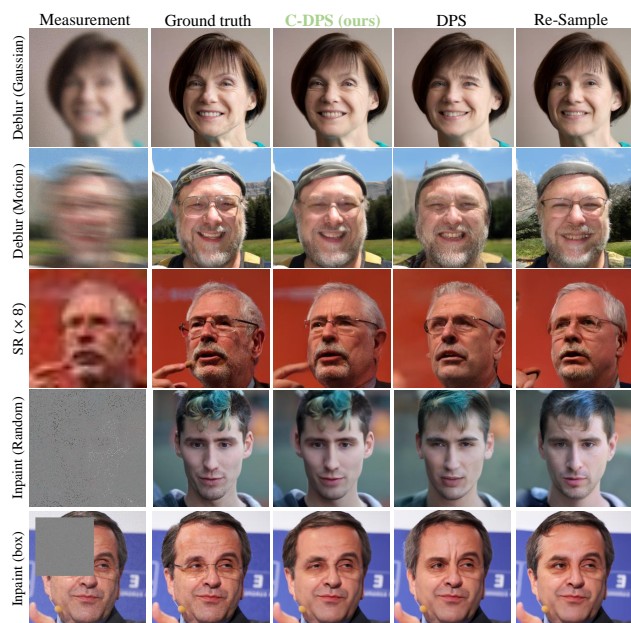

Figure 2: Qualitative results on FFHQ dataset.

As observed, C-DPS consistently produces reconstructions that more closely resemble the ground truth, exhibiting fewer artifacts and better perceptual quality compared with DPS and ReSample.

## 4.3 Measurement Fidelity

In this subsection, we demonstrate that C-DPS achieves higher measurement fidelity compared with DPS and ReSample. Specifically, we show that C-DPS recovers solutions $x$ that better satisfy the measurement model.

To this end, we conduct a motion deblurring experiment under a noiseless setting, where the Gaussian noise is removed (i.e., $y = \mathbf{A}x_0$). We then track the reconstruction error $\|y - \mathbf{A}x_t\|_2^2$ as a function of sampling progress. The progress is expressed as a percentage relative to the total number of reverse diffusion steps taken by each method.

We run each method over 100 instances and report the average error along with the shaded region representing the range of observed values. The results are shown in Figure 4. As illustrated, C-DPS consistently maintains lower measurement error throughout the sampling trajectory, indicating better alignment with the forward measurement model.

## 4.4 Why C-DPS Works: Posterior Recovery on a Ground-Truth Benchmark

In this subsection, we aim to demonstrate that the superior performance of C-DPS stems from its improved approximation of the true posterior distribution. To support this claim, we compare its ability to recover the posterior against the best benchmarks DPS and ReSample.

To this end, we construct a toy dataset where the data distribution $p(x_0)$ is defined as a mixture of 25 Gaussian components.[2] The means and variances of the mixture components are detailed in Appendix G, where we also explain how the ground-truth posterior can be computed in closed form for any given observation $y$, measurement matrix $\mathbf{A}$, and noise level $\sigma$.

To systematically evaluate each method, we generate measurement models $(y, \mathbf{A}) \in \mathbb{R}^m \times \mathbb{R}^{m \times d}$ across combinations of signal dimension, number of measurements, and noise levels. Specifically, we consider $(d, m, \sigma) \in \{8, 80, 800\} \times \{1, 2, 4\} \times \{10^{-2}, 10^{-1}, 10^0\}$, yielding 27 distinct settings. All Gaussian mixture components are equally weighted to ensure a balanced posterior.

---

[2]We follow the dataset construction procedure in [40, 16].

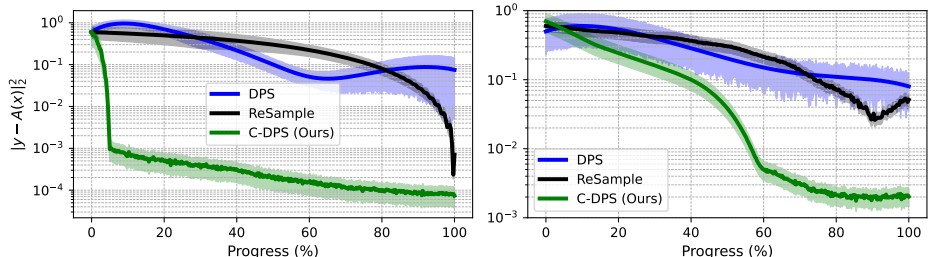

Figure 4: Evolution of the measurement error $\|\boldsymbol{y} - \mathbf{A}\boldsymbol{x}_t\|_2^2$ over the sampling trajectory for DPS, ReSample, and C-DPS on the FFHQ (left) and ImageNet (right) datasets.

Table 2: Sliced Wasserstein (SW) distance between the true and estimated posterior distributions across different dimensions $d$, numbers of measurements $m$, and noise levels $\sigma$. Lower is better.

|  | Method | $d = 8$ | | | $d = 80$ | | | $d = 800$ | | |
|---|---|---|---|---|---|---|---|---|---|---|
|  |  | $m = 1$ | $m = 2$ | $m = 4$ | $m = 1$ | $m = 2$ | $m = 4$ | $m = 1$ | $m = 2$ | $m = 4$ |
| $\sigma = 10^{-2}$ | C-DPS | **2.2** | **1.5** | **0.5** | **2.9** | **1.7** | **0.4** | **3.3** | **2.5** | **0.3** |
|  | DPS | 4.7 | 1.8 | 0.7 | 5.6 | 3.2 | 1.2 | 5.8 | 3.5 | 1.4 |
|  | ReSample | 2.6 | 2.1 | 3.8 | 3.2 | 2.8 | 0.6 | 3.5 | 3.1 | 0.4 |
| $\sigma = 10^{-1}$ | C-DPS | **1.8** | **0.9** | **0.6** | **2.5** | **1.7** | **0.4** | **2.8** | **2.3** | **0.4** |
|  | DPS | 4.7 | 1.5 | 0.8 | 5.1 | 3.1 | 1.0 | 5.7 | 3.1 | 1.3 |
|  | ReSample | 2.2 | 1.6 | 3.8 | 2.9 | 2.7 | 0.6 | 3.3 | 2.7 | **0.4** |
| $\sigma = 10^{0}$ | C-DPS | **1.2** | 1.9 | 0.9 | 1.7 | **1.2** | **0.8** | **1.6** | **1.5** | 0.7 |
|  | DPS | 5.2 | 3.5 | 2.5 | 6.9 | 3.9 | 1.7 | 6.8 | 4.7 | 0.9 |
|  | ReSample | 1.5 | 2.3 | 1.8 | **1.6** | 1.4 | 0.9 | 2.0 | 2.0 | **0.6** |

For each configuration, we generate 1000 samples from the true posterior and apply C-DPS, DPS and ReSample to approximate the posterior using 1000 steps. We then assess the quality of each approximation using the sliced Wasserstein (SW) distance [41], a metric that captures high-dimensional differences. The SW distance is computed using $10^4$ random projections per method.

Table 2 reports the mean SW distances along with 95% confidence intervals, calculated over 20 randomly sampled measurement matrices $\mathbf{A}$ per configuration. Additionally, Figure 3 visualizes the estimated posterior in the first two dimensions for the setting $(d = 80, m = 1, \sigma = 0.1)$, using a single randomly drawn measurement matrix. As illustrated, C-DPS more accurately recovers the true posterior, successfully capturing all mixture modes. In contrast, DPS and ReSample either miss certain modes or fail to represent the full posterior geometry, highlighting the advantage of our coupled diffusion formulation.

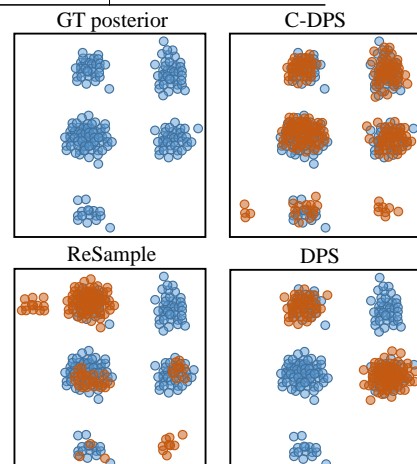

Figure 3: Visualizing the first two dimensions of the estimated posterior distributions for the configuration ($d = 80$, $m = 1$, $\sigma = 10^{-1}$) for a random $\mathbf{A}$.

## 5 Conclusion

We have introduced coupled data and measurement space diffusion posterior sampling (C-DPS), a novel framework that addresses the limitations of likelihood approximation in diffusion-based inverse problem solvers. By introducing a forward stochastic process in the measurement space $\{\boldsymbol{y}_t\}$, which evolves in parallel with the data-space diffusion $\{\boldsymbol{x}_t\}$, C-DPS enables the derivation of a closed-form posterior distribution. This coupling allows for accurate, recursive sampling without relying on approximations of the likelihood or heuristic constraints. Empirical results demonstrate that C-DPS consistently outperforms state-of-the-art baselines, delivering robust and high-fidelity reconstructions across a wide range of inverse problems. We also discuss limitations and failure cases in Appendix I, highlighting opportunities for future improvement.

## Acknowledgments

This work was supported in part by the Natural Sciences and Engineering Research Council of Canada (NSERC).

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

# A  Related Works

In many practical applications, we often encounter the underdetermined regime where $m < d$, making the inverse problem ill-posed. To obtain meaningful reconstructions in such settings, incorporating prior knowledge is essential. Within the Bayesian framework, this is addressed by modeling a prior $p(\boldsymbol{x}_0)$ and forming the posterior $p(\boldsymbol{x}_0|\boldsymbol{y})$ via Bayes' rule:

$$p(\boldsymbol{x}_0|\boldsymbol{y}) \propto p(\boldsymbol{y}|\boldsymbol{x}_0)p(\boldsymbol{x}_0).$$

When employing diffusion models as priors, one can extend the reverse-time stochastic differential euqtion (SDE) used in unconditional models,

$$d\boldsymbol{x} = \left[-\tfrac{\beta(t)}{2}\boldsymbol{x} - \beta(t)\nabla_{\boldsymbol{x}_t} \log p_t(\boldsymbol{x}_t)\right] dt + \sqrt{\beta(t)}d\bar{\boldsymbol{w}},$$

by incorporating an additional likelihood gradient to perform posterior sampling:

$$d\boldsymbol{x} = \left[-\tfrac{\beta(t)}{2}\boldsymbol{x} - \beta(t)\left(\nabla_{\boldsymbol{x}_t} \log p_t(\boldsymbol{x}_t) + \nabla_{\boldsymbol{x}_t} \log p_t(\boldsymbol{y}|\boldsymbol{x}_t)\right)\right] dt + \sqrt{\beta(t)}d\bar{\boldsymbol{w}}. \tag{22}$$

This follows directly from the identity:

$$\nabla_{\boldsymbol{x}_t} \log p_t(\boldsymbol{x}_t|\boldsymbol{y}) = \nabla_{\boldsymbol{x}_t} \log p_t(\boldsymbol{x}_t) + \nabla_{\boldsymbol{x}_t} \log p_t(\boldsymbol{y}|\boldsymbol{x}_t).$$

In this formulation, two quantities are needed: the prior score $\nabla_{\boldsymbol{x}_t} \log p_t(\boldsymbol{x}_t)$, which can be obtained from a pre-trained diffusion model, and the likelihood score $\nabla_{\boldsymbol{x}_t} \log p_t(\boldsymbol{y}|\boldsymbol{x}_t)$. The latter is generally intractable, since the data $\boldsymbol{y}$ is conditionally dependent only on $\boldsymbol{x}_0$, not directly on $\boldsymbol{x}_t$. As a result, estimating this likelihood gradient becomes a key challenge. In the following sections, we explore various strategies for approximating this term.

**Score-ALD [42] and Score-SDE [25].**  Among the earliest approaches for solving linear inverse problems with diffusion models are Score-ALD and Score-SDE, both of which estimate the gradient of the log-likelihood to steer the reverse diffusion trajectory. The key difference lies in how they compute the residual. Score-ALD uses a deterministic correction:

$$\nabla_{\boldsymbol{x}_t} \log p(\boldsymbol{y}|\boldsymbol{x}_t) \approx -\frac{\mathbf{A}^\top (\boldsymbol{y} - \mathbf{A}\boldsymbol{x}_t)}{\sigma_{\boldsymbol{y}}^2 + \gamma_t^2},$$

where $\gamma_t$ is a tunable parameter regulating the guidance strength. On the other hand, Score-SDE adds Gaussian noise to the measurements before evaluating the discrepancy:

$$\nabla_{\boldsymbol{x}_t} \log p(\boldsymbol{y}|\boldsymbol{x}_t) \approx -\mathbf{A}^\top (\boldsymbol{y} + \sigma_t \boldsymbol{\epsilon} - \mathbf{A}\boldsymbol{x}_t), \quad \boldsymbol{\epsilon} \sim \mathcal{N}(\mathbf{0}, I_m).$$

While both methods use the measurement error to guide the reverse process, Score-SDE introduces stochasticity by perturbing the observations, effectively pushing the samples toward a noisy target $\boldsymbol{y}_t = \boldsymbol{y} + \sigma_t \boldsymbol{\epsilon}$.

**ILVR [10].**  ILVR, originally introduced for super-resolution tasks, applies a similar principle by guiding the reverse process with a noised version of the measurements. Its gradient approximation takes the form

$$\nabla_{\boldsymbol{x}_t} \log p(\boldsymbol{y}|\boldsymbol{x}_t) \approx -\mathbf{A}^\dagger (\boldsymbol{y}_t - \mathbf{A}\boldsymbol{x}_t) = -(\mathbf{A}^\top \mathbf{A})^{-1} \mathbf{A}^\top (\boldsymbol{y}_t - \mathbf{A}\boldsymbol{x}_t),$$

where $\mathbf{A}^\dagger$ denotes the Moore-Penrose pseudo-inverse of $A$, and $\boldsymbol{y}_t = \boldsymbol{y} + \sigma_t \boldsymbol{\epsilon}$. Compared with Score-SDE, ILVR can be viewed as a preconditioned variant, replacing the simple adjoint $\mathbf{A}^\top$ with a full pseudo-inverse to achieve a more accurate projection onto the measurement-consistent space.

**DPS [7].**  While the previous methods are tailored for *linear* inverse problems, DPS extends to *non-linear* settings and is among the most widely used reconstruction techniques in this domain. The core approximation behind DPS is

$$\nabla_{\boldsymbol{x}_t} \log p(\boldsymbol{y}|\boldsymbol{x}_t) \approx \nabla_{\boldsymbol{x}_t} \log p\left(\boldsymbol{y} \mid \boldsymbol{x}_0 = \mathbb{E}[\boldsymbol{x}_0 \mid \boldsymbol{x}_t]\right).$$

Assuming additive Gaussian noise and a forward operator $\mathbf{A}$, the likelihood becomes

$$p\left(\boldsymbol{y} \mid \boldsymbol{x}_0 = \mathbb{E}[\boldsymbol{x}_0 \mid \boldsymbol{x}_t]\right) = \mathcal{N}\left(\boldsymbol{y}; \mathbf{A}(\mathbb{E}[\boldsymbol{x}_0 \mid \boldsymbol{x}_t]), \sigma_{\boldsymbol{y}}^2 I\right),$$

yielding the approximation

$$\nabla_{\boldsymbol{x}_t} \log p(\boldsymbol{y}|\boldsymbol{x}_t) \approx \frac{1}{\sigma_{\boldsymbol{y}}^2} \nabla_{\boldsymbol{x}_t}^\top \mathbf{A}(\mathbb{E}[\boldsymbol{x}_0 \mid \boldsymbol{x}_t]) \left(\mathbf{A}(\mathbb{E}[\boldsymbol{x}_0 \mid \boldsymbol{x}_t]) - \boldsymbol{y}\right).$$

For linear inverse problems with $\mathbf{A}(\boldsymbol{x}) = \mathbf{A}\boldsymbol{x}$, this simplifies to

$$\nabla_{\boldsymbol{x}_t} \log p(\boldsymbol{y}|\boldsymbol{x}_t) \approx -\frac{1}{\sigma_{\boldsymbol{y}}^2} \nabla_{\boldsymbol{x}_t}^\top \mathbb{E}[\boldsymbol{x}_0 \mid \boldsymbol{x}_t] \mathbf{A}^\top \left(\boldsymbol{y} - A\mathbb{E}[\boldsymbol{x}_0 \mid \boldsymbol{x}_t]\right).$$

Using Tweedie's formula, the gradient term can be further written as

$$\nabla_{\boldsymbol{x}_t} \log p(\boldsymbol{y}|\boldsymbol{x}_t) \approx -\frac{1}{\sigma_{\boldsymbol{y}}^2} \left(I + \nabla_{\boldsymbol{x}_t}^2 \log p_t(\boldsymbol{x}_t)\right)^\top \mathbf{A}^\top \left(\boldsymbol{y} - A\mathbb{E}[\boldsymbol{x}_0 \mid \boldsymbol{x}_t]\right).$$

In practice, DPS does not rely on the theoretical guidance strength above, and instead employs an adaptive step size inversely scaled by the norm of the measurement error.

**MCG [12], DSG [13], and MPGD [43].** Several recent works build on DPS by incorporating geometric insights to improve reconstruction quality. MCG [12] provides a geometric interpretation of DPS, demonstrating that its approximation helps maintain samples on the data manifold. Building on this, DSG [13] introduces a theoretically grounded step size derived from the MCG perspective, and combines it with projected gradient descent to enhance sample fidelity. More recently, MPGD [43] further improves performance by constraining the update steps to lie within a learned low-dimensional subspace using autoencoding, effectively regularizing the sampling path to remain close to the underlying manifold.

**ΠGDM [8].** The DPS approximation effectively replaces the posterior $p(\boldsymbol{x}_0|\boldsymbol{x}_t)$ with a Dirac delta centered at its mean:
$$p(\boldsymbol{x}_0|\boldsymbol{x}_t) \approx \delta\left(\boldsymbol{x}_0 - \mathbb{E}[\boldsymbol{x}_0|\boldsymbol{x}_t]\right).$$
In contrast, ΠGDM [8] proposes a more expressive approximation using a Gaussian distribution:

$$p(\boldsymbol{x}_0|\boldsymbol{x}_t) \approx \mathcal{N}\left(\mathbb{E}[\boldsymbol{x}_0|\boldsymbol{x}_t], r_t^2 I_n\right),$$

where $r_t$ is a tunable hyperparameter. For linear inverse problems, this leads to the marginal likelihood

$$p(\boldsymbol{y}|\boldsymbol{x}_t) \approx \mathcal{N}\left(\mathbf{A}\mathbb{E}[\boldsymbol{x}_0|\boldsymbol{x}_t], r_t^2 \mathbf{A}\mathbf{A}^\top + \sigma_{\boldsymbol{y}}^2 I\right),$$

and the corresponding gradient becomes

$$\nabla_{\boldsymbol{x}_t} \log p(\boldsymbol{y}|\boldsymbol{x}_t) \approx -\frac{\partial \mathbb{E}[\boldsymbol{x}_0|\boldsymbol{x}_t]}{\partial \boldsymbol{x}_t} (r_t^2 \mathbf{A}\mathbf{A}^\top + \sigma_{\boldsymbol{y}}^2 I)^{-1} \mathbf{A}^\top \left(\boldsymbol{y} - \mathbf{A}\mathbb{E}[\boldsymbol{x}_0|\boldsymbol{x}_t]\right).$$

This formulation softens the delta approximation of DPS and introduces a covariance-aware correction, improving flexibility in modeling uncertainty.

**Moment Matching [44].** The ΠGDM method assumes an isotropic Gaussian approximation for $p(\boldsymbol{x}_0|\boldsymbol{x}_t)$, ignoring the structure of its true covariance. Moment Matching [44] improves upon this by leveraging an exact expression for the posterior variance

$$V[\boldsymbol{x}_0|\boldsymbol{x}_t] = \sigma_t^4 H(\log p_t(\boldsymbol{x}_t)) + \sigma_t^2 I_n = \sigma_t^2 \nabla_{\boldsymbol{x}_t} \mathbb{E}[\boldsymbol{x}_0|\boldsymbol{x}_t],$$

resulting in an anisotropic Gaussian approximation:

$$p(\boldsymbol{x}_0|\boldsymbol{x}_t) \approx \mathcal{N}(\mathbb{E}[\boldsymbol{x}_0|\boldsymbol{x}_t], V[\boldsymbol{x}_0|\boldsymbol{x}_t]).$$

For linear inverse problems, this yields a refined estimate for the measurement score:

$$\nabla_{\boldsymbol{x}_t} \log p(\boldsymbol{y}|\boldsymbol{x}_t) \approx -\nabla_{\boldsymbol{x}_t} \mathbb{E}[\boldsymbol{x}_0|\boldsymbol{x}_t]^\top \mathbf{A}^\top \left(\sigma_{\boldsymbol{y}}^2 I + \mathbf{A}\sigma_t^2 \nabla_{\boldsymbol{x}_t} \mathbb{E}[\boldsymbol{x}_0|\boldsymbol{x}_t]\mathbf{A}^\top\right)^{-1} \left(\boldsymbol{y} - \mathbf{A}\mathbb{E}[\boldsymbol{x}_0|\boldsymbol{x}_t]\right).$$

In high-dimensional settings, explicitly computing the full Jacobian $\nabla_{\boldsymbol{x}_t} \mathbb{E}[\boldsymbol{x}_0|\boldsymbol{x}_t]$ is computationally prohibitive. To address this, the authors employ automatic differentiation to evaluate Jacobian-vector products efficiently, avoiding the need to form the full matrix.

**SNIPS [45] and DDRM [14].** These methods reformulate linear inverse problems as noisy inpainting in the spectral domain via the singular value decomposition (SVD) of the measurement matrix, $\mathbf{A} = U\Sigma V^\top$. With this, the measurement model becomes

$$\bar{\boldsymbol{y}} = \Sigma\bar{\boldsymbol{x}} + \sigma_{\boldsymbol{y}}\bar{\boldsymbol{z}}, \quad \text{where} \quad \bar{\boldsymbol{x}} = V^\top\boldsymbol{x},\ \bar{\boldsymbol{y}} = U^\top\boldsymbol{y},\ \bar{\boldsymbol{z}} = U^\top\boldsymbol{z}.$$

SNIPS solves the inverse problem in this transformed space by performing annealed Langevin dynamics to sample from $p(\bar{\boldsymbol{x}}|\bar{\boldsymbol{y}})$, using the approximation

$$\nabla_{\bar{\boldsymbol{x}}_t} \log p(\bar{\boldsymbol{y}}|\bar{\boldsymbol{x}}_t) \approx -\Sigma^\top \left|\sigma_{\boldsymbol{y}}^2 I - \sigma_t^2 \Sigma\Sigma^\top\right|^\dagger (\bar{\boldsymbol{y}} - \Sigma\bar{\boldsymbol{x}}_t).$$

DDRM improves upon SNIPS by replacing the noisy iterate $\bar{\boldsymbol{x}}_t$ with the posterior mean $\bar{\boldsymbol{x}}_{0|t} = V^\top\mathbb{E}[\boldsymbol{x}_0|\boldsymbol{x}_t]$ in the above expression. Furthermore, DDRM introduces a sampling rule based on conditional Gaussian distributions over the spectral components $\bar{x}_t^{(i)}$, which adapt based on the singular value $s_i$, the diffusion noise level $\sigma_t$, and a hyperparameter $\eta \in (0, 1]$ to control stochasticity:

$$\bar{x}_t^{(i)} \sim \begin{cases} \mathcal{N}(\bar{x}_{0|t+1}^{(i)} + \sqrt{1-\eta^2}\sigma_t \frac{\bar{x}_{t+1}^{(i)} - \bar{x}_{0|t+1}^{(i)}}{\sigma_{t+1}}, \eta^2\sigma_t^2), & s_i = 0 \\ \mathcal{N}(\bar{x}_{0|t+1}^{(i)} + \sqrt{1-\eta^2}\sigma_t \frac{\bar{y}^{(i)} - \bar{x}_{0|t+1}^{(i)}}{\sigma_{\boldsymbol{y}}/s_i}, \eta^2\sigma_t^2), & \sigma_t < \frac{\sigma_{\boldsymbol{y}}}{s_i} \\ \mathcal{N}(\bar{y}^{(i)}, \sigma_t^2 - \frac{\sigma_{\boldsymbol{y}}^2}{s_i^2}), & \sigma_t \geq \frac{\sigma_{\boldsymbol{y}}}{s_i} \end{cases}.$$

When $\eta = 1$, this sampling reduces to the deterministic posterior formulation used in the original DDRM algorithm.

**DDS [46] and DiffPIR [47].** Both DDS and DiffPIR approximate the posterior mean via a proximal formulation:

$$\mathbb{E}[\boldsymbol{x}_0|\boldsymbol{x}_t, \boldsymbol{y}] \approx \arg\min_{\boldsymbol{x}} \frac{1}{2}\|\boldsymbol{y} - \mathbf{A}\boldsymbol{x}\|^2 + \frac{\lambda_t}{2}\|\boldsymbol{x} - \mathbb{E}[\boldsymbol{x}_0|\boldsymbol{x}_t]\|^2.$$

This balances fidelity to the measurements with proximity to the diffusion-based prior. The methods differ in solving this objective and selecting $\lambda_t$. DDS solves it approximately via a few conjugate gradient (CG) steps, motivated by replacing DPS gradient updates with CG under data manifold assumptions, and uses a fixed $\lambda_t$. In contrast, DiffPIR adopts a closed-form solution and schedules $\lambda_t$ proportional to the signal-to-noise ratio at time $t$, specifically $\lambda_t = \sigma_t\zeta$ with $\zeta$ as a constant.

Another line of work tackles inverse problems via variational inference, where the true posterior $p(\boldsymbol{x}_0|\boldsymbol{y})$ is approximated by a tractable distribution $q$, optimized by minimizing the KL divergence.

**RED-Diff [48].** RED-Diff approximates $p(\boldsymbol{x}_0|\boldsymbol{y})$ using a Gaussian $q = \mathcal{N}(\boldsymbol{\mu}, \sigma^2 I)$, minimizing their KL divergence as follows

$$\min_q \mathcal{D}_{\mathrm{KL}}(q(\boldsymbol{x}_0|\boldsymbol{y}) \,\|\, p(\boldsymbol{x}_0|\boldsymbol{y})).$$

This leads to the variational bound

$$\min_{\mu} \frac{\|\boldsymbol{y} - \mathbf{A}(\boldsymbol{\mu})\|^2}{2\sigma_{\boldsymbol{y}}^2} + \mathbb{E}_{t,\boldsymbol{\epsilon}}[\lambda_t\|\boldsymbol{\epsilon}_\theta(\boldsymbol{x}_t, t) - \boldsymbol{\epsilon}\|^2],$$

which combines a reconstruction loss and a score-matching term.

**Score Prior [49].** This approach uses a normalizing flow $q_\phi$ as the variational distribution and minimizes

$$\mathbb{E}_{\boldsymbol{z}\sim\mathcal{N}(0,I)}\left[-\log p(\boldsymbol{y}|G_\phi(\boldsymbol{z})) - \log p_\theta(G_\phi(\boldsymbol{z})) + \log\pi(\boldsymbol{z}) - \log|\det\frac{dG_\phi}{d\boldsymbol{z}}|\right].$$

The prior term $\log p_\theta$ is computed via the PF-ODE formulation, though it is computationally expensive and must be optimized per measurement.

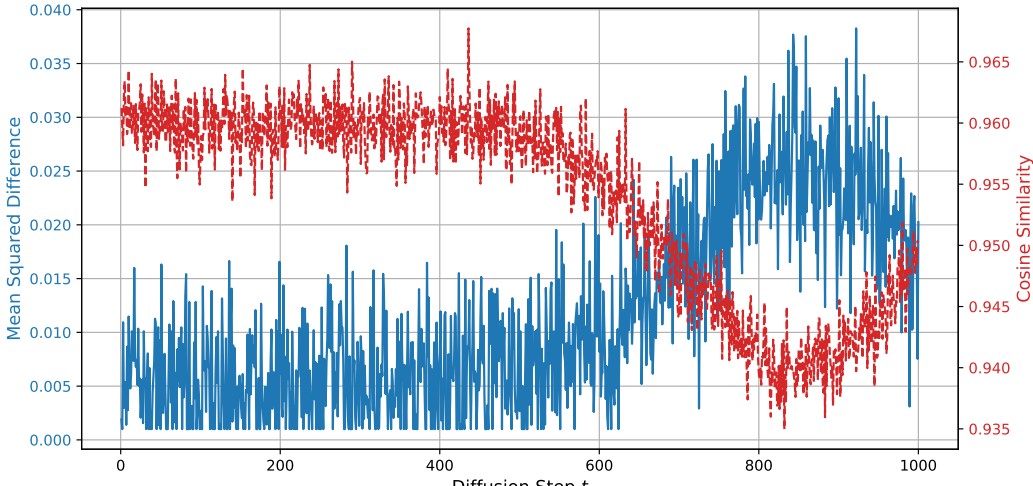

Figure 5: Cosine similarity and MSE between score vectors. Cosine similarity remains high and MSE stays low across diffusion steps, supporting the approximation $s_\theta(\boldsymbol{x}_{t-1}, t-1) \approx s_\theta(\boldsymbol{x}_t, t)$.

**Efficient Score Prior [50].**   To reduce cost, this variant replaces the exact likelihood with a surrogate lower bound

$$b_\theta(\boldsymbol{x}_0) = \mathbb{E}_{p(\boldsymbol{x}_T | \boldsymbol{x}_0)}[\log \pi(\boldsymbol{x}_T)] - \frac{1}{2} \int_0^T g(t)^2 h(t)\, dt,$$

where $h(t)$ includes the denoising loss and other tractable terms. This approximation drastically reduces computation while maintaining training quality.

As seen, variational methods approximate the true posterior with a simpler distribution whose parameters can be efficiently optimized using standard techniques. While this enables tractable inference, it may limit expressiveness, as the chosen distribution might fail to capture the full complexity of the true posterior.

## B    Empirical Justification for the Score Approximation in Equation (14)

In Section 3, we introduced a practical approximation to enable tractable posterior sampling:

$$s_\theta(\boldsymbol{x}_{t-1}, t-1) \quad \longrightarrow \quad s_\theta(\boldsymbol{x}_t, t).$$

This simplification avoids computing the score at the unknown $\boldsymbol{x}_{t-1}$ by reusing the available value at $\boldsymbol{x}_t$. To empirically validate the accuracy of this approximation, we measure how closely the two score vectors align in both direction and magnitude across the diffusion trajectory.

We quantify this with the C-DPS cosine similarity and C-DPS mean squared error (MSE) between the score vectors $s_\theta(\boldsymbol{x}_{t-1}, t-1)$ and $s_\theta(\boldsymbol{x}_t, t)$ at each diffusion step $t$, where $\boldsymbol{x}_{t-1}$ is generated from $\boldsymbol{x}_t$ via a standard reverse diffusion step. Specifically, we compute the cosine similarity as

$$\text{CosSim}(t) = \frac{\langle s_\theta(\boldsymbol{x}_{t-1}, t-1), s_\theta(\boldsymbol{x}_t, t) \rangle}{\| s_\theta(\boldsymbol{x}_{t-1}, t-1) \|_2 \cdot \| s_\theta(\boldsymbol{x}_t, t) \|_2},$$

and the MSE as

$$\text{MSE}(t) = \frac{1}{d} \| s_\theta(\boldsymbol{x}_{t-1}, t-1) - s_\theta(\boldsymbol{x}_t, t) \|_2^2,$$

where $d$ is the dimensionality of the data.

Figure 5 shows both metrics averaged over 100 randomly sampled instances from the FFHQ dataset, evaluated across the full reverse diffusion trajectory (i.e., $t = 1000 \rightarrow 0$).

As seen, the C-DPS cosine similarity remains high (above 0.94) throughout the diffusion process. This indicates that the two score vectors are consistently aligned in direction, even if their magnitudes differ. Since the reverse process relies more on the direction of the gradient, this supports the validity of the approximation.

In addition, The C-DPS MSE remains low across the trajectory, peaking modestly during the middle diffusion steps (approximately $t \in [400, 600]$). This is expected since

- In early steps (high $t$), the noise level is large, and score estimates change slowly.
- In later steps (low $t$), the signal is nearly denoised, and the score stabilizes.
- The middle regime exhibits faster dynamics as the model transitions from noise to structure, hence slightly higher variation in scores.

These empirical findings demonstrate that our approximation in Equation (14) introduces minimal directional or magnitude error, particularly in regions where diffusion steps are small. This justifies its use in our closed-form posterior sampling framework without sacrificing empirical performance.

## C    Latent C-DPS

In the latent version of C-DPS, both the diffusion process and the measurement model are defined over the latent space of a pretrained generative model, rather than the pixel space. This modification allows us to leverage the benefits of a compressed latent representation, improving computational efficiency and robustness, particularly for high-resolution data.

Algorithm 4 presents the detailed steps for the latent C-DPS framework. Compared with its pixel-domain counterpart (Algorithm 2), the following changes are made:

- The unknown signal is represented by latent variables $\{\ell_t\}_{t=0}^{T}$ instead of pixel variables $\{x_t\}_{t=0}^{T}$.
- The forward operator $\mathbf{A}$ and the measurement $y$ are defined in the latent space.
- The score function $s_\theta$ is evaluated on latent variables rather than pixel variables.
- All sampling, posterior updates, and noise schedules are performed in the latent space following the same principles as in the pixel domain.

Despite these modifications, the overall structure of the algorithm remains the same: a forward diffusion process is applied to the measurement, followed by a backward sampling procedure using closed-form posterior updates that couple the data and measurement spaces at each time step.

---

**Algorithm 4** Latent C-DPS (ours)

---

**Input:** # time steps $T$, latent measurement $y$, noise schedule $\{\beta_t\}$, forward operator $\mathbf{A}$ in latent space, and noise covariance $\Sigma_{\mathbf{n}}$.
1: Generate $\{y_t\}_{t=0}^{T}$ as per latent-space version of Equation (5)
2: $\ell_T \sim \mathcal{N}(\mathbf{0}, \boldsymbol{I})$
3: **for** $t = T - 1, T - 2, \ldots, 0$ **do**
4:     $\hat{s} \leftarrow s_\theta(\ell_t, t)$
5:     $\boldsymbol{\mu}_{y|\ell} \leftarrow \mathbf{A}\ell_{t-1} + (1 - \bar{\alpha}_{t-1})\mathbf{A}\hat{s}$
6:     $\boldsymbol{\Sigma}_{y|\ell} \leftarrow \bar{\alpha}_{t-1}\Sigma_{\mathbf{n}} + (1 - \bar{\alpha}_{t-1})\mathbf{I}$
7:     Compute $\boldsymbol{\mu}_{\text{post}}$ and $\boldsymbol{\Sigma}_{\text{post}}$ via (16), replacing $x_t$ with $\ell_t$
8:     $z \sim \mathcal{N}(\mathbf{0}, \boldsymbol{I})$
9:     $\ell_{t-1} \leftarrow \boldsymbol{\mu}_{\text{post}} + \boldsymbol{\Sigma}_{\text{post}} z$
10: **end for**
**Output:** $\ell_0$

---

## D    Extension to Nonlinear Forward Models

**Locally linear C–DPS.**    Let the measurement model be $y = g(x_0) + n$ with differentiable $g : \mathbb{R}^d \to \mathbb{R}^m$ and $n \sim \mathcal{N}(0, \Sigma_n)$. The coupled construction still applies, while the Gaussian closed form in Eq. (16) holds after a local linearization of $g$ at the current iterate $x_t$:

$$g(x) \approx g(x_t) + J_t(x - x_t), \quad J_t = \nabla g\big|_{x=x_t}. \tag{23}$$

With the same $\Sigma_{y|x} = \bar{\alpha}_{t-1}\Sigma_n + (1 - \bar{\alpha}_{t-1})I$ as in the linear case, the conditional becomes

$$p(\boldsymbol{y}_{t-1} \mid \boldsymbol{x}_{t-1}) \approx \mathcal{N}(J_t\boldsymbol{x}_{t-1} + \boldsymbol{c}_t, \Sigma_{y|x}), \quad \boldsymbol{c}_t = g(\boldsymbol{x}_t) - J_t\boldsymbol{x}_t + (1 - \bar{\alpha}_{t-1})J_t\boldsymbol{s}_\theta(\boldsymbol{x}_t, t). \tag{24}$$

Combining this with the diffusion prior $p(\boldsymbol{x}_t \mid \boldsymbol{x}_{t-1}) = \mathcal{N}(\sqrt{1 - \beta_t}\,\boldsymbol{x}_{t-1}, \beta_t I)$ yields a Gaussian posterior, identical in form to Eq. (16) with $A \leftarrow J_t$ and $b_{t-1} \leftarrow \boldsymbol{c}_t$:

$$\Sigma_{\text{post}}^{-1} = \frac{1 - \beta_t}{\beta_t} I + J_t^\top \Sigma_{y|x}^{-1} J_t, \tag{25}$$

$$\boldsymbol{\mu}_{\text{post}} = \Sigma_{\text{post}}\left[\frac{\sqrt{1 - \beta_t}}{\beta_t}\boldsymbol{x}_t + J_t^\top \Sigma_{y|x}^{-1}(\boldsymbol{y}_{t-1} - \boldsymbol{c}_t)\right]. \tag{26}$$

This is a Gauss–Newton or extended-Kalman smoothing view inside the C–DPS recursion. It adds one Jacobian–vector product per step, which modern autodiff provides efficiently. The dominant cost remains the score call. In our setup, the score pass costs about 100 to 200 ms and the linear solve is under 10 ms per step, so the added Jacobian–vector product ($< 15$ ms) is modest.

| Phase Retrieval | | | | |
|---|---|---|---|---|
| Method | FID ↓ | LPIPS ↓ | SSIM ↑ | PSNR ↑ |
| DAPS | 42.71 | 0.139 | 0.851 | 30.63 |
| DPS | 104.5 | 0.410 | 0.441 | 17.64 |
| RED–diff | 167.4 | 0.596 | 0.398 | 15.60 |
| PnP–DM | 99.4 | 0.335 | 0.581 | 19.69 |
| DMPlug | 41.21 | 0.124 | 0.894 | 31.25 |
| **C–DPS (local linear)** | **41.32** | **0.129** | **0.903** | **31.91** |
| Nonlinear Deblur | | | | |
| Method | FID ↓ | LPIPS ↓ | SSIM ↑ | PSNR ↑ |
| DAPS | 49.38 | 0.155 | 0.783 | 28.29 |
| DPS | 91.31 | 0.278 | 0.623 | 23.39 |
| RED–diff | 43.84 | 0.160 | 0.795 | 30.86 |
| PnP–DM | 68.96 | 0.193 | 0.742 | 27.81 |
| DMPlug | 47.28 | 0.135 | 0.792 | 29.55 |
| **C–DPS (local linear)** | **47.52** | **0.130** | **0.802** | **30.27** |

**Preliminary nonlinear results (FFHQ 256×256).** These early results show that locally linear C–DPS retains the advantages of the linear model while remaining computationally feasible. We also observed DMPlug to be close in accuracy but about $5\times$ slower in our setup.

# E   Implementation Details

## E.1   Implementation Details of Baseline Methods

We evaluate all baseline methods using their official implementations and default settings. The corresponding repositories are listed below:

- DPS [7] & MCG [30] & ΠGDM [8]: `https://github.com/DPS2022/diffusion-posterior-sampling`

- DDRM [14]: `https://github.com/bahjat-kawar/ddrm`

- ADMM-PnP [31]: `https://github.com/kanglin755/plug_and_play_admm`

- ILVR [10]: `https://github.com/jychoi118/ilvr_adm`

- ReSample [20]: `https://github.com/soominkwon/resample/tree/main`

- PSLD [35]: We follow the original code from the repository provided by [35] with the pretrained LDMs on FFHQ and ImageNet datasets. We use the default hyperparameters as implied in [35]. For some tasks, the hyperparameters were tuned by grid search, and the results according to the optimal hyperparameters are reported.

## E.2 Construction of Whitening Operator W

To solve linear systems involving the posterior precision $\mathbf{\Lambda}_t = \frac{1-\beta_t}{\beta_t}\mathbf{I} + \mathbf{A}^\top\mathbf{\Sigma}_{\boldsymbol{y}|\boldsymbol{x}}^{-1}\mathbf{A}$, we employ a pre-whitened formulation based on the factorization

$$\mathbf{\Sigma}_{\boldsymbol{y}|\boldsymbol{x}}^{-1} = \mathbf{W}^\top\mathbf{W},$$

where $\mathbf{W}$ is a *whitening operator*. This appendix describes how to construct $\mathbf{W}$ under common structural assumptions[3] on the measurement noise covariance $\mathbf{\Sigma}_{\boldsymbol{n}}$, which is used to define $\mathbf{\Sigma}_{\boldsymbol{y}|\boldsymbol{x}} = \bar{\alpha}_{t-1}\mathbf{\Sigma}_{\boldsymbol{n}} + (1 - \bar{\alpha}_{t-1})\mathbf{I}$.

**Case 1: Diagonal or isotropic noise.** If $\mathbf{\Sigma}_{\boldsymbol{n}} = \sigma^2\mathbf{I}$, then

$$\mathbf{\Sigma}_{\boldsymbol{y}|\boldsymbol{x}} = \left(\bar{\alpha}_{t-1}\sigma^2 + 1 - \bar{\alpha}_{t-1}\right)\mathbf{I} \triangleq \gamma_t\mathbf{I}, \quad \text{so} \quad \mathbf{W} = \gamma_t^{-1/2}\mathbf{I}.$$

**Case 2: Low-rank plus diagonal noise.** Suppose $\mathbf{\Sigma}_{\boldsymbol{n}} = \mathbf{U}\mathbf{U}^\top + \sigma^2\mathbf{I}$ for a tall matrix $\mathbf{U} \in \mathbb{R}^{m\times r}$ with $r \ll m$. Then,

$$\mathbf{\Sigma}_{\boldsymbol{y}|\boldsymbol{x}} = \bar{\alpha}_{t-1}(\mathbf{U}\mathbf{U}^\top + \sigma^2\mathbf{I}) + (1 - \bar{\alpha}_{t-1})\mathbf{I} = \bar{\alpha}_{t-1}\mathbf{U}\mathbf{U}^\top + \delta_t\mathbf{I},$$

where $\delta_t = \bar{\alpha}_{t-1}\sigma^2 + 1 - \bar{\alpha}_{t-1}$. The inverse can be computed using the Woodbury identity:

$$\mathbf{\Sigma}_{\boldsymbol{y}|\boldsymbol{x}}^{-1} = \delta_t^{-1}\left(\mathbf{I} - \mathbf{U}\left(\mathbf{I} + \frac{\bar{\alpha}_{t-1}}{\delta_t}\mathbf{U}^\top\mathbf{U}\right)^{-1}\frac{\bar{\alpha}_{t-1}}{\delta_t}\mathbf{U}^\top\right).$$

To match the form $\mathbf{\Sigma}^{-1} = \mathbf{W}^\top\mathbf{W}$, define $\mathbf{W}$ as a matrix whose Gram matrix gives the above (e.g., via Cholesky of the RHS).

**Case 3: Convolutional (circulant) noise.** If $\mathbf{\Sigma}_{\boldsymbol{n}}$ is circulant (e.g., for colored Gaussian noise in image processing), then it is diagonalized by the Discrete Fourier Transform (DFT):

$$\mathbf{\Sigma}_{\boldsymbol{n}} = \mathbf{F}^\dagger\mathbf{D}\mathbf{F}, \tag{27}$$

$$\text{so} \quad \mathbf{\Sigma}_{\boldsymbol{y}|\boldsymbol{x}} = \mathbf{F}^\dagger\mathbf{D}'\mathbf{F}, \tag{28}$$

where $\mathbf{F}$ is the DFT matrix and $\mathbf{D}' = \bar{\alpha}_{t-1}\mathbf{D} + (1 - \bar{\alpha}_{t-1})\mathbf{I}$ is diagonal. Then

$$\mathbf{\Sigma}_{\boldsymbol{y}|\boldsymbol{x}}^{-1} = \mathbf{F}^\dagger\mathbf{D}'^{-1}\mathbf{F} \quad \text{and} \quad \mathbf{W} = \mathbf{D}'^{-1/2}\mathbf{F}.$$

This allows $\mathbf{W}\boldsymbol{v}$ and $\mathbf{W}^\top\boldsymbol{v}$ to be computed efficiently using FFTs.

**Implementation note.** Although we derive closed-form expressions for the whitening operator $\mathbf{W}$ in all three cases (diagonal, low-rank, and convolutional noise), we do not instantiate $\mathbf{W}$ as an explicit matrix. Instead, we implement the action of $\mathbf{W}$ and its transpose through matrix-free routines, which apply $\boldsymbol{v} \mapsto \mathbf{W}\boldsymbol{v}$ and $\boldsymbol{v} \mapsto \mathbf{W}^\top\boldsymbol{v}$ directly using efficient operations. For instance, in the convolutional case, $\mathbf{W}$ is implemented via FFTs; in the low-rank case, via structured low-rank multiplications using the Woodbury identity; and in the diagonal case, via elementwise scaling. This matrix-free formulation is sufficient because our conjugate gradient solver requires only matrix-vector products with the precision matrix $\mathbf{\Lambda}_t = \frac{1-\beta_t}{\beta_t}\mathbf{I} + \mathbf{A}^\top\mathbf{W}^\top\mathbf{W}\mathbf{A}$. By avoiding explicit construction of $\mathbf{W}$ and leveraging fast vectorized operations, we ensure that each CG solve has negligible memory overhead and runtime comparable to a single evaluation of the score network, even in high-dimensional settings.

# F  Additional PSNR Results on Pixel-Domain Tasks

## F.1  Runtime Comparison on FFHQ 256×256

We report wall-clock runtimes on a single NVIDIA P100 (12 GB GPU), batch size = 1, and 1,000 reverse-diffusion steps in the pixel domain. All methods use the same pretrained score network for a fair comparison. Times are averaged over 50 images.

C–DPS runs within approximately 15% of DPS, and it is significantly faster than recent methods such as DMPlug. It is slightly slower than DPS due to the added posterior conditioning, however the overall runtime remains comparable given the improved accuracy and the principled posterior formulation.

---

[3]These cases are representative of real-world applications such as compressed sensing, deblurring, and MRI reconstruction.

Table 3: PSNR (dB) on FFHQ $256 \times 256$. **Bold** is best, underline is second-best for each column.

| Method | Inpaint (Rand) | Inpaint (Box) | Deblur (Gauss) | Deblur (Motion) | SR ($4\times$) |
|---|---|---|---|---|---|
| DPS | 25.23 | 22.51 | 24.25 | 24.92 | 25.86 |
| DDRM | 9.19 | 22.26 | 24.93 | – | 26.58 |
| MCG | 21.57 | 19.97 | 6.72 | – | 18.20 |
| ReSample | 25.90 | 23.81 | 26.33 | 26.00 | 25.22 |
| PnP-ADMM | 8.41 | 11.65 | 24.93 | – | 26.55 |
| Score-SDE | 13.52 | 18.51 | 7.12 | – | 17.62 |
| ADMM-TV | 22.03 | 17.81 | 22.37 | – | 23.86 |
| PnP-DM | 26.20 | 22.41 | 25.71 | 25.80 | 27.90 |
| DAPS | 28.33 | 24.07 | 29.19 | 29.66 | 29.07 |
| DMPlug | 28.71 | **28.92** | 30.02 | **29.91** | **30.25** |
| **C-DPS (ours)** | **28.95** | 28.69 | **30.13** | 29.85 | 30.12 |

Table 4: Time per image on FFHQ $256 \times 256$ (seconds, lower is better).

| Method | Time [s] $\downarrow$ |
|---|---|
| DPS | 130 |
| MCG | 142 |
| DAPS | 113 |
| PnP-DM | 181 |
| DMPlug | 550 |
| **C–DPS (ours)** | **152** |

# G   Toy dataset

Table 5: Sliced Wasserstein for VE-DDPM.

| $d$ | $m$ | $\sigma = 0.01$ | | | $\sigma = 0.1$ | | | $\sigma = 1.0$ | | |
|---|---|---|---|---|---|---|---|---|---|---|
| | | C-DPS | ReSample | DPS | C-DPS | ReSample | DPS | C-DPS | ReSample | DPS |
| 8 | 1 | **1.9** ± 0.5 | 2.6 ± 0.9 | 4.7 ± 1.5 | **1.4** ± 0.6 | 2.2 ± 0.9 | 4.7 ± 1.6 | **1.2** ± 0.6 | 1.5 ± 0.4 | 5.2 ± 1.3 |
| 8 | 2 | **0.8** ± 0.4 | 2.1 ± 1.0 | 1.8 ± 1.5 | **1.0** ± 0.4 | 1.6 ± 0.6 | 1.5 ± 0.9 | **1.0** ± 0.3 | 2.3 ± 0.4 | 3.5 ± 1.2 |
| 8 | 4 | **0.4** ± 0.2 | 3.8 ± 2.3 | 0.7 ± 0.6 | **0.2** ± 0.2 | 3.8 ± 2.2 | 0.8 ± 0.6 | **0.7** ± 0.3 | 1.8 ± 0.3 | 2.5 ± 0.9 |
| 80 | 1 | **2.7** ± 0.7 | 3.2 ± 1.0 | 5.6 ± 1.8 | **2.4** ± 0.8 | 2.9 ± 0.8 | 5.1 ± 1.8 | **1.5** ± 0.7 | 1.6 ± 0.5 | 6.9 ± 1.8 |
| 80 | 2 | **1.1** ± 0.6 | 2.8 ± 1.3 | 3.2 ± 1.9 | **1.3** ± 0.4 | 2.7 ± 1.2 | 3.1 ± 1.9 | **1.0** ± 0.3 | 1.4 ± 0.2 | 3.9 ± 1.2 |
| 80 | 4 | **0.4** ± 0.2 | 0.6 ± 0.4 | 1.2 ± 1.1 | **0.5** ± 0.3 | 0.6 ± 0.4 | 1.0 ± 1.1 | **0.9** ± 0.3 | 0.9 ± 0.2 | 1.7 ± 0.6 |
| 800 | 1 | **3.1** ± 0.7 | 3.5 ± 1.1 | 5.8 ± 1.6 | **3.0** ± 0.5 | 3.3 ± 0.9 | 5.7 ± 1.6 | **1.4** ± 0.5 | 2.0 ± 0.4 | 6.8 ± 1.0 |
| 800 | 2 | **1.5** ± 0.5 | 3.1 ± 1.1 | 3.5 ± 1.7 | **1.2** ± 0.4 | 2.7 ± 0.9 | 3.1 ± 1.4 | **1.3** ± 0.4 | 2.0 ± 0.5 | 4.7 ± 1.3 |
| 800 | 4 | 0.5 ± 0.3 | **0.4** ± 0.2 | 1.4 ± 1.0 | **0.3** ± 0.2 | 0.4 ± 0.2 | 1.3 ± 0.9 | 0.9 ± 0.2 | **0.7** ± 0.3 | 0.9 ± 0.4 |

Table 6: Sliced Wasserstein for the GMM case for the reverse VE SDEs discretized with Euler-Maruyama.

| $d$ | $m$ | $\sigma = 0.01$ | | | $\sigma = 0.1$ | | | $\sigma = 1.0$ | | |
|---|---|---|---|---|---|---|---|---|---|---|
| | | C-DPS | ReSample | DPS | C-DPS | ReSample | DPS | C-DPS | ReSample | DPS |
| 8 | 1 | **1.6** ± 0.4 | 1.5 ± 0.4 | 5.7 ± 2.2 | **1.3** ± 0.4 | 1.2 ± 0.4 | 5.6 ± 2.1 | **0.8** ± 0.3 | 0.9 ± 0.3 | 0.9 ± 0.3 |
| 8 | 2 | **0.6** ± 0.3 | 0.4 ± 0.3 | 6.2 ± 0.8 | **1.0** ± 0.4 | 0.5 ± 0.3 | 6.2 ± 2.4 | **0.8** ± 0.2 | 1.0 ± 0.3 | 1.2 ± 0.4 |
| 8 | 4 | **0.4** ± 0.2 | 0.1 ± 0.1 | - | **0.4** ± 0.2 | 0.1 ± 0.0 | 8.4 ± 3.1 | **0.7** ± 0.2 | 0.2 ± 0.1 | 0.3 ± 0.2 |
| 80 | 1 | **2.5** ± 0.7 | 2.9 ± 1.4 | 9.1 ± 1.3 | **2.1** ± 0.8 | 2.1 ± 1.1 | 4.7 ± 1.8 | **1.4** ± 0.7 | 1.8 ± 0.8 | 1.9 ± 0.9 |
| 80 | 2 | **1.2** ± 0.4 | 0.8 ± 0.7 | 2.2 ± 0.9 | **1.1** ± 0.5 | 0.8 ± 0.7 | 6.0 ± 2.1 | **1.3** ± 0.3 | 1.3 ± 0.5 | 1.5 ± 0.5 |
| 80 | 4 | **0.4** ± 0.1 | 0.1 ± 0.0 | - | **0.3** ± 0.2 | 0.1 ± 0.1 | 4.4 ± 1.6 | **0.8** ± 0.3 | 0.4 ± 0.2 | 0.5 ± 0.3 |
| 800 | 1 | **3.2** ± 0.6 | 3.2 ± 1.0 | 6.8 ± 1.2 | **2.8** ± 0.5 | 2.8 ± 0.7 | 6.4 ± 1.5 | **1.4** ± 0.4 | 1.3 ± 0.3 | 1.3 ± 0.3 |
| 800 | 2 | **1.4** ± 0.3 | 0.8 ± 0.5 | 7.4 ± 0.9 | **1.2** ± 0.3 | 0.8 ± 0.4 | 6.4 ± 1.9 | **1.3** ± 0.4 | 1.1 ± 0.3 | 1.1 ± 0.3 |
| 800 | 4 | **0.4** ± 0.2 | 0.6 ± 0.5 | - | **0.3** ± 0.2 | 0.1 ± 0.0 | 5.8 ± 1.4 | **0.8** ± 0.3 | 0.4 ± 0.2 | 0.4 ± 0.2 |

The generation of this dataset is inspired from [16].

As explained earlier in the paper, we model $p_0(\boldsymbol{x}_0)$ as a mixture of 25 Gaussian distributions. Each of these Gaussian components has a mean vector $\mathbf{U}_{i,j}$ in $\mathbb{R}^d$, defined as $\mathbf{U}_{i,j} = (8i, 8j, \ldots, 8i, 8j)$

for each pair $(i, j)$ where $i$ and $j$ take values from the set $\{-2, -1, 0, 1, 2\}$. All components have the same variance of 1. The unnormalized weight associated with each component is $\omega_{i,j} = 1.0$. Additionally, we have set the variance of the noise, $\sigma_\delta^2$, to $10^{-4}$.

Recall that the distribution $p_t(\boldsymbol{x}_t)$ can be expressed as an integral: $p_t(\boldsymbol{x}_t) = \int p_{t|0}(\boldsymbol{x}_t|\boldsymbol{x}_0)p_0(\boldsymbol{x}_0)d\boldsymbol{x}_0$. Since $p_0(\boldsymbol{x}_0)$ is a mixture of Gaussian distributions, $p_t(\boldsymbol{x}_t)$ is also a mixture of Gaussians. The means of these Gaussians are given by $\sqrt{\alpha_t}\mathbf{U}_{i,j}$, and each Gaussian has unit variance. By using automatic differentiation libraries, we can efficiently compute the gradient $\nabla_{\boldsymbol{x}_t} \log p_t(\boldsymbol{x}_t)$.

We have set the parameters $\beta_{\max} = 500.0$ and $\beta_{\min} = 0.1$, and we use 1000 timesteps to discretize the time domain. For a given pair of dimensions and a chosen observation noise standard deviation $(d, m, \sigma)$, the measurement model $(\boldsymbol{y}, \mathbf{A})$ is generated as follows:

• Matrix $\mathbf{A}$: First, we sample a random matrix $\tilde{\mathbf{A}}$ from a Gaussian distribution $\mathcal{N}(\mathbf{0}_{m \times d}, \mathbf{I}_{m \times d})$. We then compute its singular value decomposition (SVD), $\tilde{\mathbf{A}} = \mathbf{U}\mathbf{S}\mathbf{V}^\top$. For each pair $(i, j)$ in $\{-2, -1, 0, 1, 2\}^2$, we draw a singular value $s_{i,j}$ from a uniform distribution on the interval $[0, 1]$. Finally, we construct the matrix $\mathbf{A} = \mathbf{U}\text{diag}(\{s_{i,j}\}_{(i,j) \in \{-2, -1, 0, 1, 2\}^2})\mathbf{V}^\top$.

• Observation vector $\boldsymbol{y}$: Next, we sample a vector $\boldsymbol{x}_*$ from the distribution $p_0$. The observation vector $\boldsymbol{y}$ is then obtained by applying the matrix $\mathbf{A}$ to $\boldsymbol{x}_*$ and adding Gaussian noise $\boldsymbol{z}$, where $\boldsymbol{z}$ is sampled from $\mathcal{N}(\mathbf{0}, \sigma^2\mathbf{I}_m)$.

Once we have drawn both $\boldsymbol{x}_* \sim p_0$ and $(\boldsymbol{y}, \mathbf{A}, \sigma)$, the posterior can be exactly calculated using Bayes formula and gives a mixture of Gaussians with mixture components $c_{i,j}$ and associated weights $\tilde{\omega}_{i,j}$,

$$c_{i,j} := \mathcal{N}(\boldsymbol{\Sigma}(\mathbf{A}^\top \boldsymbol{y}/\sigma^2 + \mathbf{U}_{i,j}), \boldsymbol{\Sigma}), \tag{29}$$

$$\tilde{\omega}_i := \omega_i \mathcal{N}(\boldsymbol{y}; \mathbf{A}\mathbf{U}_{i,j}, \sigma_\delta^2\mathbf{I}_d + \mathbf{A}\mathbf{A}^\top), \tag{30}$$

where $\boldsymbol{\Sigma} = (\mathbf{I}_d + \frac{1}{\sigma_\delta^2}\mathbf{A}^\top\mathbf{A})^{-1}$.

• **SW Distance Calculation.** To compare the posterior distribution estimated by each algorithm with the target posterior distribution, we use $10^4$ slices for the SW distance and compare 1000 samples of the true posterior distribution.

Table 5 and Table 6 indicate the 95% confidence intervals obtained by considering 20 randomly selected measurement models ($\mathbf{A}$) for each setting $(d, m, \sigma)$.

## H  More Experiments

### H.1  Additional Quantitative Results for Other Noise Levels

Tables 7 and 8 report the same set of metrics as Table 1 in the main paper, but for Gaussian measurement noise levels $\sigma = 0.07$ and $\sigma = 0.03$, respectively. Both tables include pixel-domain methods in the upper block and latent-domain methods in the lower block, covering the same five inverse problems: random inpainting, box inpainting, Gaussian deblurring, motion deblurring, and $4\times$ super-resolution.

$\sigma = 0.07$.  When the noise standard deviation is increased from 0.05 to 0.07 (Table 7), all methods experience moderate degradation: FID and LPIPS rise while SSIM falls. Nevertheless, C-DPS remains the top performer in every task on both datasets. The performance gap between C-DPS and the best baseline widens in most cases, indicating that our closed-form posterior update is more robust to heavier measurement noise than the projection- or likelihood-based alternatives.

$\sigma = 0.03$.  With milder noise ($\sigma = 0.03$, Table 8), absolute scores improve across the board, but C-DPS still delivers the best or second-best results in all settings. The advantage is most pronounced for inpainting and Gaussian deblurring, where C-DPS attains the lowest FID and LPIPS and the highest SSIM. These trends confirm that the coupled posterior formulation benefits both high- and low-noise regimes.

**Overall observation.**  Across $\sigma \in \{0.03, 0.05, 0.07\}$, C-DPS consistently outperforms state-of-the-art baselines, showing graceful performance degradation as noise increases and the largest gains under

Table 7: Quantitative results for Gaussian noise level $\sigma = 0.07$ on the 1k validation sets of FFHQ $256 \times 256$ and ImageNet $256 \times 256$. **Bold** and underline indicate the best and second-best results, respectively.

| | | Inpaint (Random) | | | Inpaint (Box) | | | Deblur (Gaussian) | | | Deblur (Motion) | | | SR ($4\times$) | | |
|---|---|---|---|---|---|---|---|---|---|---|---|---|---|---|---|---|
| | | \multicolumn{15}{c}{Pixel-Domain Methods} | | | | | | | | | | | | | |
| Dataset | Method | FID↓ | LPIPS↓ | SSIM↑ | FID↓ | LPIPS↓ | SSIM↑ | FID↓ | LPIPS↓ | SSIM↑ | FID↓ | LPIPS↓ | SSIM↑ | FID↓ | LPIPS↓ | SSIM↑ |
| FFHQ | DPS | 23.1 | 0.228 | 0.833 | 36.9 | 0.183 | 0.858 | 48.4 | 0.277 | 0.792 | 44.3 | 0.262 | 0.842 | 42.8 | 0.232 | 0.836 |
| | IIGDM | 23.3 | 0.238 | 0.820 | 38.9 | 0.194 | 0.845 | 44.4 | 0.264 | 0.807 | 36.3 | 0.243 | 0.872 | 38.3 | 0.218 | 0.836 |
| | DDRM | 76.2 | 0.619 | 0.294 | 46.3 | 0.227 | 0.843 | 80.6 | 0.352 | 0.748 | – | – | – | 66.8 | 0.318 | 0.814 |
| | MCG | 32.4 | 0.308 | 0.727 | 44.0 | 0.330 | 0.677 | 110.3 | 0.360 | 0.048 | – | – | – | 93.7 | 0.544 | 0.535 |
| | ILVR | 28.5 | 0.251 | 0.648 | 40.3 | 0.190 | 0.835 | 58.2 | 0.318 | 0.764 | – | – | – | 52.3 | 0.273 | 0.827 |
| | ReSample | 23.2 | 0.217 | 0.827 | 36.9 | 0.173 | 0.845 | 40.8 | 0.271 | 0.806 | 34.3 | 0.238 | 0.874 | 33.4 | 0.224 | 0.835 |
| | PnP-ADMM | 135.9 | 0.723 | 0.306 | 168.3 | 0.430 | 0.621 | 99.8 | 0.470 | 0.791 | – | – | – | 72.8 | 0.379 | **0.848** |
| | Score-SDE | 84.9 | 0.639 | 0.418 | 66.4 | 0.355 | 0.659 | 121.4 | 0.431 | 0.105 | – | – | – | 103.2 | 0.593 | 0.596 |
| | ADMM-TV | 196.8 | 0.486 | 0.760 | 74.9 | 0.346 | 0.793 | 202.1 | 0.532 | 0.780 | – | – | – | 121.8 | 0.455 | 0.781 |
| | **C-DPS** | **22.1** | **0.209** | **0.862** | **28.9** | **0.150** | 0.859 | **35.4** | **0.259** | **0.814** | **30.3** | **0.236** | **0.904** | **31.0** | **0.214** | 0.841 |
| | C-DPS vs. Best | -1.1 | -0.008 | 0.035 | -8.0 | -0.023 | -0.014 | -5.4 | -0.012 | 0.008 | -4.0 | -0.002 | 0.030 | -2.4 | -0.010 | -0.006 |
| ImageNet | DPS | 39.8 | 0.327 | 0.713 | 42.9 | 0.280 | 0.768 | 67.6 | 0.472 | 0.683 | 60.8 | 0.414 | 0.614 | 55.4 | 0.361 | 0.756 |
| | IIGDM | 46.4 | 0.382 | 0.678 | 46.7 | 0.307 | 0.729 | 65.2 | 0.450 | **0.695** | 59.3 | 0.398 | **0.655** | 59.3 | 0.374 | 0.741 |
| | DDRM | 124.0 | 0.704 | 0.384 | 50.2 | 0.267 | **0.787** | 68.9 | 0.455 | 0.684 | – | – | – | 65.7 | 0.367 | 0.768 |
| | MCG | 42.9 | 0.441 | 0.525 | 44.4 | 0.344 | 0.608 | 101.6 | 0.579 | 0.421 | – | – | – | 155.6 | 0.669 | 0.219 |
| | ILVR | 42.0 | 0.398 | 0.635 | 41.5 | 0.298 | 0.708 | 77.6 | 0.451 | 0.639 | – | – | – | 103.1 | 0.567 | 0.472 |
| | ReSample | 37.2 | 0.311 | 0.708 | 42.3 | 0.278 | 0.771 | 66.1 | 0.468 | 0.687 | 60.1 | 0.399 | 0.618 | 52.7 | 0.359 | 0.751 |
| | PnP-ADMM | 124.9 | 0.713 | 0.285 | 85.3 | 0.388 | 0.636 | 109.3 | 0.548 | 0.641 | – | – | – | 104.1 | 0.459 | 0.736 |
| | Score-SDE | 138.4 | 0.695 | 0.494 | 59.4 | 0.374 | 0.591 | 131.8 | 0.706 | 0.419 | – | – | – | 186.4 | 0.741 | 0.231 |
| | ADMM-TV | 205.1 | 0.540 | 0.654 | 95.1 | 0.343 | 0.760 | 170.3 | 0.618 | 0.611 | – | – | – | 143.1 | 0.554 | 0.654 |
| | **C-DPS** | **35.4** | **0.236** | **0.729** | **36.4** | **0.254** | 0.781 | **60.2** | **0.417** | 0.691 | **55.1** | **0.372** | 0.630 | **50.1** | **0.331** | **0.772** |
| | C-DPS vs. Best | -1.8 | -0.075 | 0.016 | -5.1 | -0.013 | -0.006 | -5.0 | -0.033 | -0.004 | -4.2 | -0.027 | -0.025 | -2.6 | -0.028 | 0.004 |

| | | Inpaint (Random) | | | Inpaint (Box) | | | Deblur (Gaussian) | | | Deblur (Motion) | | | SR ($4\times$) | | |
|---|---|---|---|---|---|---|---|---|---|---|---|---|---|---|---|---|
| | | \multicolumn{15}{c}{Latent-Domain Methods} | | | | | | | | | | | | | |
| Dataset | Method | FID↓ | LPIPS↓ | SSIM↑ | FID↓ | LPIPS↓ | SSIM↑ | FID↓ | LPIPS↓ | SSIM↑ | FID↓ | LPIPS↓ | SSIM↑ | FID↓ | LPIPS↓ | SSIM↑ |
| FFHQ | PSLD | 52.6 | 0.242 | 0.786 | 47.9 | 0.177 | 0.791 | 97.3 | 0.338 | 0.600 | 104.9 | 0.360 | 0.648 | 81.4 | 0.308 | 0.618 |
| | ReSample | 44.3 | 0.157 | 0.718 | 59.2 | 0.201 | 0.722 | 79.6 | 0.276 | 0.686 | 49.6 | 0.214 | 0.800 | 102.2 | 0.423 | 0.562 |
| | LC-DPS | **40.7** | **0.153** | 0.780 | **46.2** | **0.158** | **0.799** | **72.8** | **0.256** | **0.730** | 51.8 | 0.294 | 0.793 | **64.5** | **0.220** | **0.764** |
| | LC-DPS vs. Best | -3.6 | -0.004 | 0.006 | -1.7 | -0.019 | 0.008 | -6.8 | -0.020 | 0.044 | 2.2 | 0.080 | -0.007 | -16.9 | -0.088 | 0.146 |
| ImageNet | PSLD | 91.5 | 0.362 | 0.764 | 160.1 | 0.489 | 0.660 | 100.8 | 0.414 | 0.663 | 137.5 | 0.549 | 0.568 | 106.2 | 0.384 | **0.671** |
| | ReSample | 64.2 | 0.158 | 0.732 | 139.9 | 0.285 | 0.608 | **71.4** | 0.276 | 0.684 | 73.0 | 0.246 | 0.714 | 123.8 | 0.390 | 0.558 |
| | LC-DPS | 64.7 | **0.154** | **0.764** | **125.8** | **0.274** | **0.690** | 73.8 | **0.265** | **0.700** | **69.3** | **0.233** | **0.728** | **96.7** | **0.312** | 0.653 |
| | LC-DPS vs. Best | 0.5 | -0.004 | 0.032 | -14.1 | -0.011 | 0.030 | 2.4 | -0.011 | 0.016 | -3.7 | -0.013 | 0.014 | -9.5 | -0.072 | -0.018 |

challenging noise conditions. The latent variant LC-DPS exhibits the same behaviour, confirming that the proposed framework is effective in both pixel and latent domains.

## H.2 More Qualitative Results

In this section, we present additional qualitative results on the FFHQ dataset across various inverse problem settings. These results serve to further demonstrate the flexibility and effectiveness of C-DPS in handling diverse measurement operators beyond those shown in the main text.

Figure 6 shows reconstructions for the Gaussian deblurring task, where C-DPS successfully restores high-frequency details that are often lost in baseline methods. Figure 7 presents results for motion deblurring, highlighting C-DPS's ability to recover sharp facial features under structured blur.

In Figure 8, we evaluate C-DPS on an 8× super-resolution task. Despite the aggressive downsampling, our method reconstructs globally consistent and perceptually plausible images. Figures 9 and 10 display results on random inpainting and box inpainting, respectively. In both cases, C-DPS generates semantically coherent completions even under extreme occlusions.

These qualitative results further support the claims made in the main paper, showing that C-DPS generalizes well across a wide range of inverse problems with minimal task-specific tuning.

# I Limitations and Failure Cases

While C-DPS demonstrates strong performance across a variety of inverse problems, we observe that it can struggle in certain edge cases. For example, when the measurement matrix $\mathbf{A}$ is nearly rank-deficient or the noise level is extremely high, the propagated measurement information $\{\boldsymbol{y}_t\}$

Table 8: Quantitative results for Gaussian noise level $\sigma = 0.03$ on the 1k validation sets of FFHQ $256 \times 256$ and ImageNet $256 \times 256$. **Bold** and underline indicate the best and second-best results, respectively.

| | | Pixel-Domain Methods | | | | | | | | | | | | | | | |
|---|---|---|---|---|---|---|---|---|---|---|---|---|---|---|---|---|---|
| **Dataset** | **Method** | **Inpaint (Random)** | | | **Inpaint (Box)** | | | **Deblur (Gaussian)** | | | **Deblur (Motion)** | | | **SR (4×)** | | |
| | | FID↓ | LPIPS↓ | SSIM↑ | FID↓ | LPIPS↓ | SSIM↑ | FID↓ | LPIPS↓ | SSIM↑ | FID↓ | LPIPS↓ | SSIM↑ | FID↓ | LPIPS↓ | SSIM↑ |
| FFHQ | DPS | 19.4 | 0.196 | 0.872 | 29.8 | 0.151 | 0.891 | 36.2 | 0.230 | 0.842 | 32.9 | 0.216 | 0.882 | 32.6 | 0.192 | 0.873 |
| | IIGDM | 19.6 | 0.204 | 0.861 | 31.1 | 0.160 | 0.878 | 33.5 | 0.218 | 0.857 | 27.6 | 0.198 | 0.904 | 29.1 | 0.181 | 0.875 |
| | DDRM | 61.3 | 0.516 | 0.363 | 36.7 | 0.175 | 0.895 | 63.2 | 0.291 | 0.807 | – | – | – | 52.3 | 0.268 | 0.861 |
| | MCG | 24.9 | 0.249 | 0.792 | 32.4 | 0.266 | 0.745 | 86.9 | 0.301 | 0.069 | – | – | – | 74.2 | 0.466 | 0.604 |
| | ILVR | 22.3 | 0.208 | 0.709 | 31.9 | 0.157 | 0.874 | 44.6 | 0.257 | 0.816 | – | – | – | 39.8 | 0.217 | 0.865 |
| | ReSample | 19.2 | 0.181 | 0.868 | 29.7 | 0.145 | 0.886 | 30.9 | 0.225 | 0.836 | 26.1 | 0.203 | 0.906 | 27.2 | 0.186 | 0.872 |
| | PnP-ADMM | 110.7 | 0.622 | 0.345 | 136.2 | 0.363 | 0.694 | 82.1 | 0.392 | 0.845 | – | – | – | 59.4 | 0.315 | **0.889** |
| | Score-SDE | 61.3 | 0.498 | 0.530 | 49.2 | 0.286 | 0.746 | 101.1 | 0.347 | 0.141 | – | – | – | 85.7 | 0.471 | 0.658 |
| | ADMM-TV | 155.7 | 0.406 | 0.807 | 59.6 | 0.283 | 0.840 | 160.5 | 0.454 | 0.832 | – | – | – | 95.4 | 0.385 | 0.835 |
| | **C-DPS** | **18.3** | **0.174** | **0.898** | **23.7** | **0.118** | 0.894 | **27.6** | **0.209** | **0.856** | **23.5** | **0.189** | **0.933** | **24.9** | **0.171** | 0.880 |
| | C-DPS vs. Best | -0.9 | -0.007 | 0.030 | -6.0 | -0.027 | 0.008 | -3.3 | -0.016 | 0.020 | -2.6 | -0.014 | 0.027 | -2.3 | -0.015 | 0.008 |
| ImageNet | DPS | 31.4 | 0.257 | 0.767 | 33.9 | 0.223 | 0.821 | 54.9 | 0.385 | 0.726 | 49.0 | 0.336 | 0.676 | 44.8 | 0.289 | 0.807 |
| | IIGDM | 37.0 | 0.303 | 0.736 | 36.5 | 0.247 | 0.785 | 52.3 | 0.367 | **0.736** | 47.3 | 0.322 | **0.709** | 48.1 | 0.299 | 0.790 |
| | DDRM | 94.6 | 0.548 | 0.438 | 40.1 | 0.208 | **0.842** | 56.8 | 0.373 | 0.729 | – | – | – | 51.6 | 0.305 | 0.817 |
| | MCG | 33.6 | 0.361 | 0.565 | 34.1 | 0.290 | 0.649 | 82.2 | 0.486 | 0.466 | – | – | – | 124.0 | 0.550 | 0.271 |
| | ILVR | 32.9 | 0.324 | 0.684 | 33.1 | 0.239 | 0.753 | 61.4 | 0.376 | 0.690 | – | – | – | 82.7 | 0.454 | 0.533 |
| | ReSample | 29.8 | 0.245 | 0.758 | 34.2 | 0.220 | 0.828 | 54.6 | 0.389 | 0.731 | 48.5 | 0.318 | 0.680 | 43.5 | 0.290 | 0.803 |
| | PnP-ADMM | 94.4 | 0.556 | 0.346 | 71.2 | 0.335 | 0.700 | 87.6 | 0.461 | 0.699 | – | – | – | 84.7 | 0.373 | 0.787 |
| | Score-SDE | 104.2 | 0.573 | 0.543 | 48.1 | 0.304 | 0.649 | 105.7 | 0.596 | 0.451 | – | – | – | 149.8 | 0.625 | 0.294 |
| | ADMM-TV | 151.3 | 0.458 | 0.692 | 72.3 | 0.287 | 0.812 | 133.2 | 0.502 | 0.671 | – | – | – | 118.6 | 0.492 | 0.704 |
| | **C-DPS** | **28.6** | **0.178** | **0.779** | **29.9** | 0.205 | 0.835 | **50.1** | **0.345** | 0.735 | **45.6** | **0.300** | 0.688 | **41.2** | **0.263** | **0.823** |
| | C-DPS vs. Best | -1.2 | -0.067 | 0.012 | -4.0 | -0.015 | 0.007 | -2.2 | -0.022 | -0.001 | -1.7 | -0.018 | -0.021 | -2.3 | -0.027 | 0.006 |

| | | Latent-Domain Methods | | | | | | | | | | | | | | | |
|---|---|---|---|---|---|---|---|---|---|---|---|---|---|---|---|---|---|
| **Dataset** | **Method** | **Inpaint (Random)** | | | **Inpaint (Box)** | | | **Deblur (Gaussian)** | | | **Deblur (Motion)** | | | **SR (4×)** | | |
| | | FID↓ | LPIPS↓ | SSIM↑ | FID↓ | LPIPS↓ | SSIM↑ | FID↓ | LPIPS↓ | SSIM↑ | FID↓ | LPIPS↓ | SSIM↑ | FID↓ | LPIPS↓ | SSIM↑ |
| FFHQ | PSLD | 43.5 | 0.203 | 0.830 | 39.1 | 0.142 | 0.833 | 78.3 | 0.291 | 0.652 | 85.5 | 0.310 | 0.702 | 66.8 | 0.262 | 0.669 |
| | ReSample | 36.9 | 0.126 | 0.776 | 49.1 | 0.166 | 0.779 | 59.4 | 0.234 | 0.736 | 41.1 | **0.189** | **0.845** | 83.6 | 0.344 | 0.619 |
| | LC-DPS | 34.3 | **0.123** | 0.843 | 37.6 | 0.130 | 0.841 | 54.2 | 0.215 | 0.777 | 42.9 | 0.258 | 0.837 | 58.7 | 0.176 | 0.812 |
| | LC-DPS vs. Best | -2.6 | -0.003 | 0.013 | -1.5 | -0.012 | 0.008 | -5.2 | -0.019 | 0.041 | 1.8 | 0.069 | -0.008 | -8.1 | -0.086 | 0.143 |
| ImageNet | PSLD | 70.8 | 0.284 | 0.813 | 124.4 | 0.415 | 0.720 | 86.1 | 0.352 | 0.702 | 112.9 | 0.446 | 0.612 | 92.2 | 0.326 | **0.714** |
| | ReSample | 49.1 | 0.118 | 0.782 | 111.9 | 0.223 | 0.659 | **57.3** | 0.225 | 0.724 | 61.5 | 0.200 | 0.745 | 107.2 | 0.332 | 0.609 |
| | LC-DPS | 49.5 | **0.115** | 0.815 | 101.4 | 0.215 | 0.740 | 59.1 | 0.216 | 0.740 | 58.4 | 0.188 | 0.755 | 82.9 | 0.252 | 0.701 |
| | LC-DPS vs. Best | 0.4 | -0.003 | 0.033 | -10.5 | -0.008 | 0.020 | 1.8 | -0.009 | 0.016 | -3.1 | -0.012 | 0.010 | -9.3 | -0.074 | -0.013 |

may not sufficiently constrain the posterior, resulting in oversmoothing or loss of fine details in the reconstruction.

This behavior is not unique to C-DPS. Prior methods such as DPS and ReSample also degrade under similar conditions, as they rely on learned priors and approximate measurement integration that become less reliable when the inverse problem is highly ill-posed. In our case, the Gaussian approximation to the posterior—while principled and efficient—can be limiting when the true posterior is strongly multi-modal or non-Gaussian.

These limitations arise from current design choices that favor tractability and generality. Future extensions could incorporate adaptive diffusion schedules, refined measurement dynamics, or non-Gaussian approximations to further improve performance in these extreme regimes.

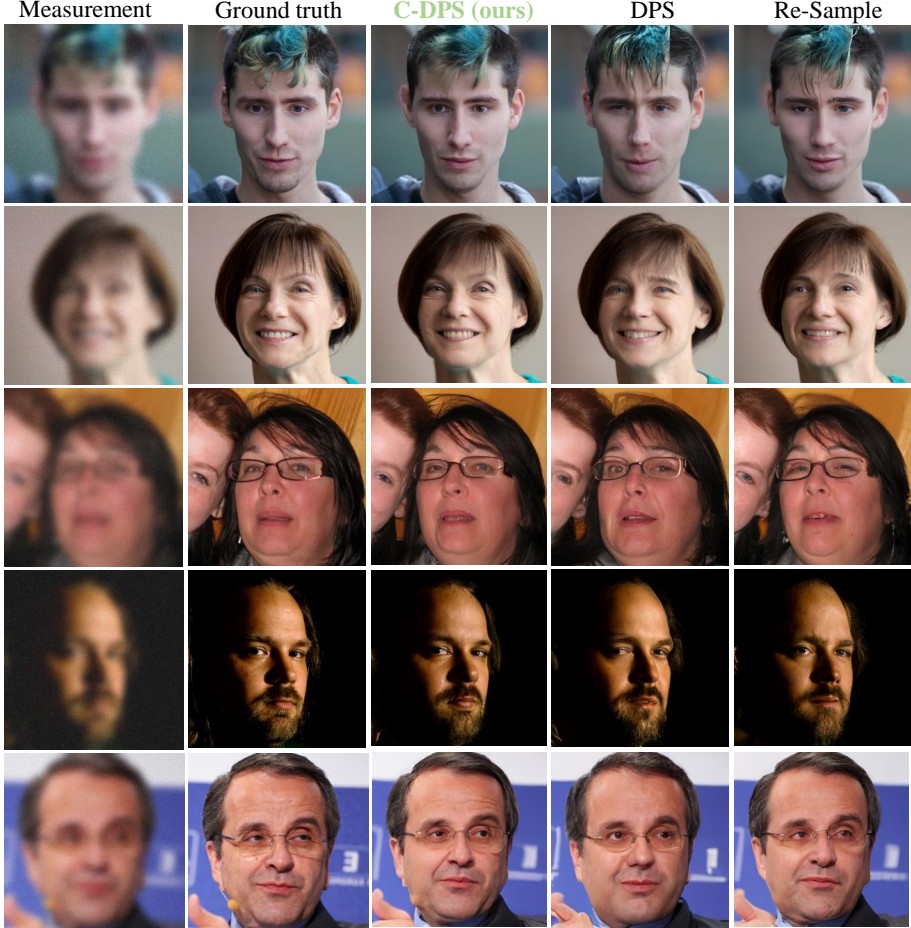

Figure 6: Qualitative results on FFHQ dataset, Gaussian deblurring.

Measurement    Ground truth    C-DPS (ours)    DPS    Re-Sample

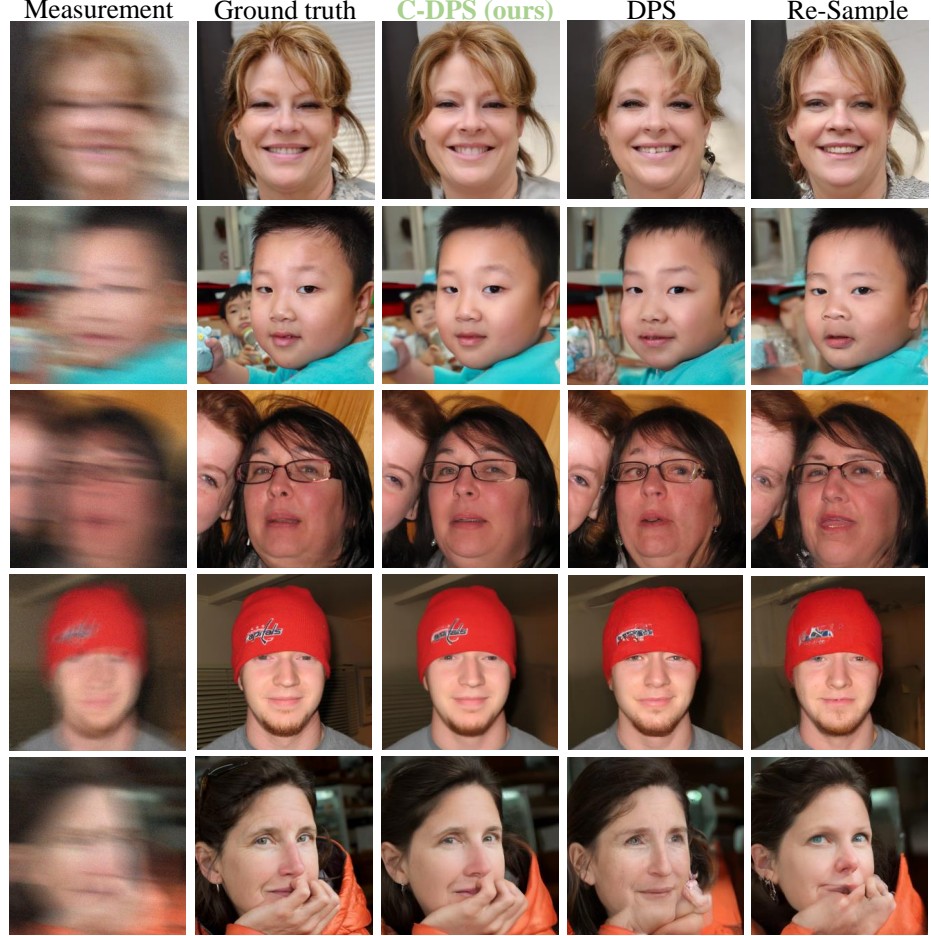

Figure 7: Qualitative results on FFHQ dataset, motion deblurring.

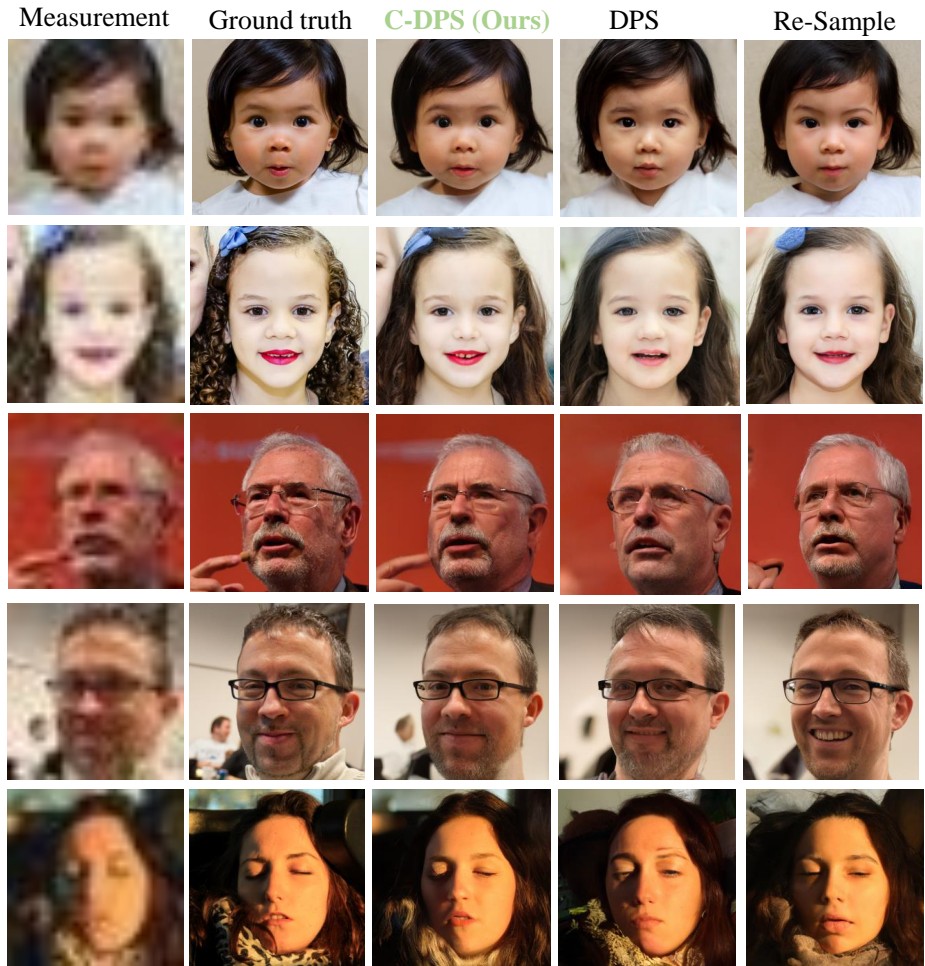

Figure 8: Qualitative results on FFHQ dataset, super-resolution task ($\times$ 8).

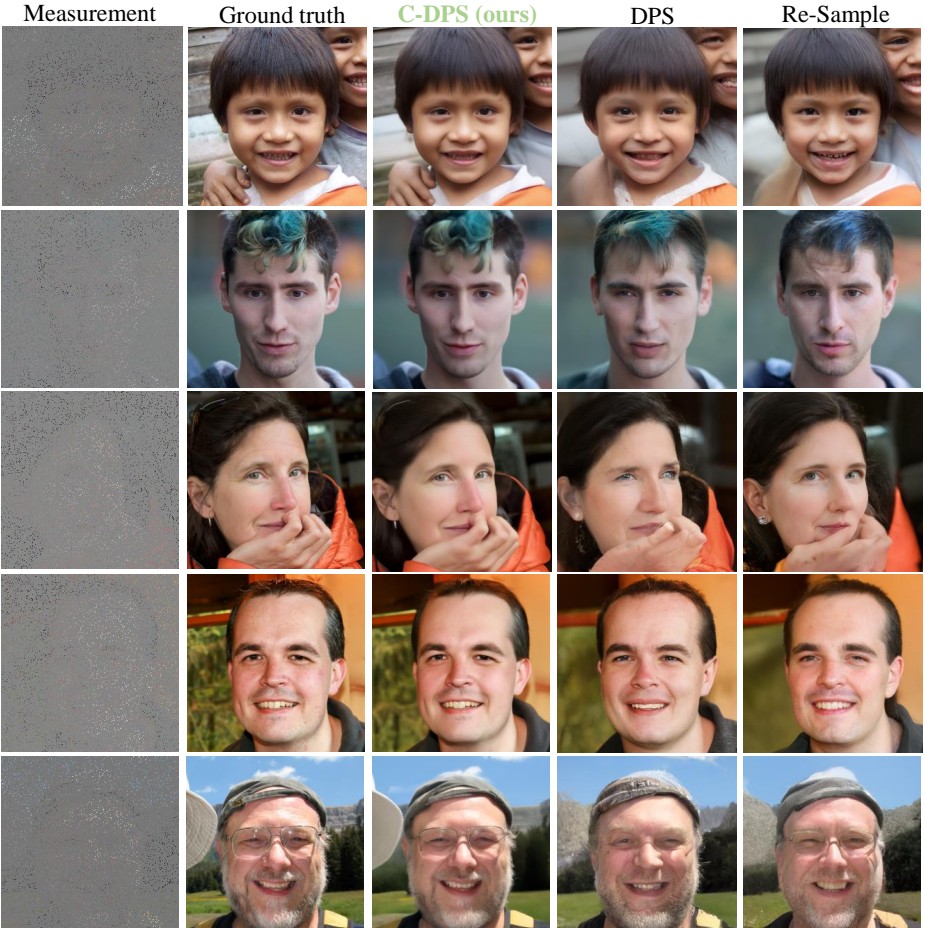

Figure 9: Qualitative results on FFHQ dataset, inpaiting task (random).

| Measurement | Ground truth | C-DPS (ours) | DPS | Re-Sample |
|:---:|:---:|:---:|:---:|:---:|

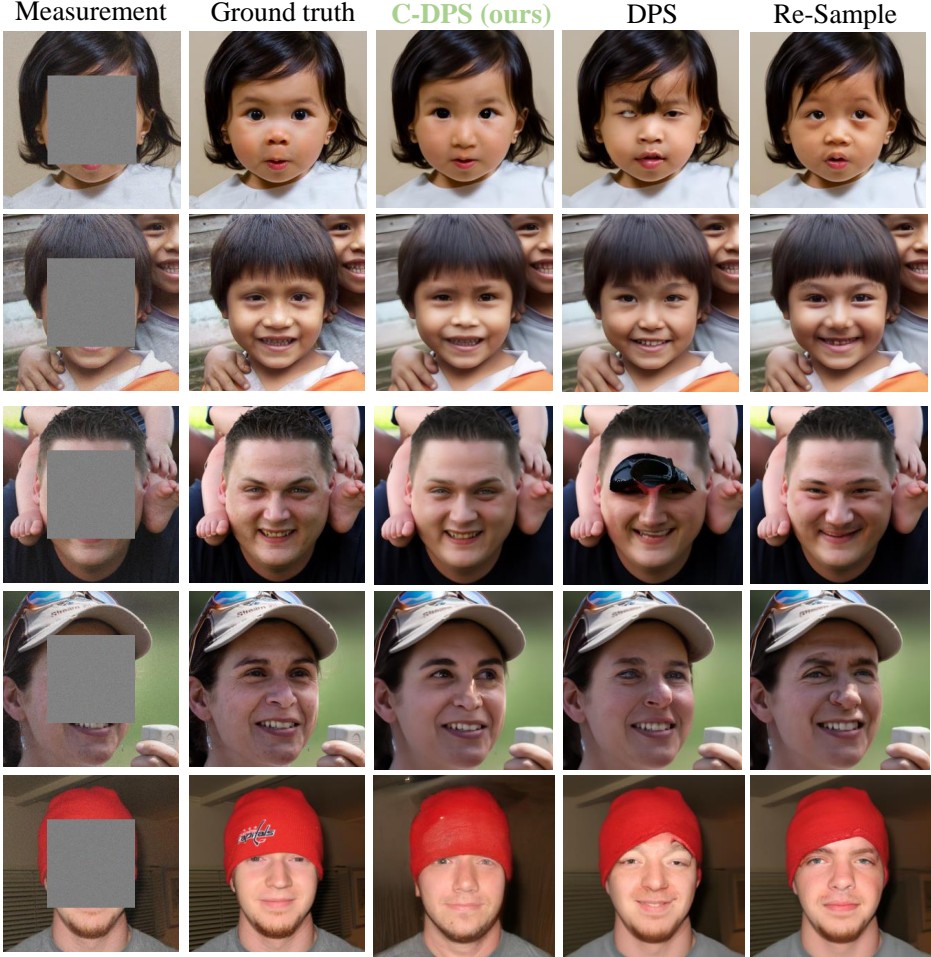

Figure 10: Qualitative results on FFHQ dataset, inpaiting task (box).

# Supplementary Materials (Evaluation on Medical Imaging Inverse Problems)

In this section, we extend our study by applying the proposed diffusion-based approach to inverse problems in the *medical imaging* domain. Medical images often contain complex structures and are typically acquired under constraints that make problems such as reconstruction, denoising, or inpainting particularly challenging. By evaluating our method on representative medical imaging tasks, we aim to demonstrate its effectiveness and generalization ability in real-world scenarios where accurate recovery of fine details is critical.

### Datasets and Preprocessing.

We evaluate our proposed method, C-DPS, on three representative medical imaging tasks using public datasets: undersampled MRI, sparse-view CT, and super-resolution. For MRI reconstruction, we use the fastMRI knee dataset [51]. Following standard preprocessing [26], we crop the raw k-space data to $320 \times 320$ resolution and reconstruct single-coil images using a minimum variance unbiased estimator. Undersampling is simulated using 1D Gaussian and Uniform sampling masks with acceleration rates (ACR) of 4 and 8.

For sparse-view CT, we utilize the LIDC dataset [52]. Two-dimensional CT slices of size $320 \times 320$ are extracted from volumetric scans. Sinograms are generated using a parallel-beam geometry with either 23 or 10 projections evenly spaced over 180 degrees to simulate sparse-view acquisition.

The super-resolution task is evaluated on the fastMRI brain dataset [51]. We downsample 2D full-resolution images to generate low-resolution inputs for $2 \times 2$ and $4 \times 4$ upscaling. We select approximately 63% of the slices labeled as `reconstruction_rss`, yielding a total of 34,698 training samples.

### Baselines and Evaluation Metrics

We compare our method, C-DPS, with strong training-free diffusion-based baselines including DPS [7], Score-MRI [2], DDS [46], and ScoreMed [26], depending on the task. All methods are implemented with their publicly available code and evaluated under consistent experimental conditions.

We report results using standard image quality metrics: Peak Signal-to-Noise Ratio (PSNR) and Structural Similarity Index (SSIM). Each task is evaluated on a test set of 1,000 images. For CT experiments, we follow prior work and set the likelihood step size $\zeta = 0$ where applicable. In all experiments, our method outperforms the baselines across all configurations in terms of both PSNR and SSIM, as shown in Tables 9, 11, and 10.

### Architecture and Sampling Settings

All models are based on the ADM architecture [28], without classifier-free guidance or dropout. Separate diffusion priors are trained for each task. For sampling, we use 100 timesteps for MRI and super-resolution, and 350 timesteps for sparse-view CT.

Our method introduces a measurement-consistent update through a bi-level guidance scheme. We tune the likelihood step size $\zeta$ and refinement weight $\gamma$ within a restricted search space, $\zeta \in [0, 2]$, $\gamma \in [0, 4.5]$, and fix $\lambda = 10^{-3}$ for the outer optimization. No extensive hyperparameter tuning is required to achieve strong performance.

All experiments are conducted on a NVIDIA P100 GPU with 12 GB of memory.

### Results

We assess the effectiveness of our method, C-DPS, across three key inverse problems: MRI reconstruction, CT reconstruction, and super-resolution. In all cases, C-DPS consistently outperforms prior training-free diffusion-based baselines, demonstrating both robustness and accuracy across varying acquisition settings.

**MRI Reconstruction.**    Table 9 reports results on the fastMRI knee dataset under Uniform1D and Gaussian1D masks at acceleration rates of $4\times$ and $8\times$. Our method achieves the highest PSNR and SSIM across all configurations. Notably, under more aggressive undersampling ($8\times$), C-DPS shows significant improvements over both DPS and DDS, highlighting its resilience in challenging reconstruction scenarios. The gains in SSIM, particularly under the Gaussian1D mask, indicate improved structural fidelity and reduced aliasing artifacts.

**Sparse-View CT Reconstruction.**    As shown in Table 11, C-DPS achieves state-of-the-art results on the LIDC dataset with both 23 and 10 projections. It outperforms ScoreMed and BGDM by a notable margin in PSNR and SSIM, despite using the same number of diffusion steps (350). The improvement is especially pronounced under the extremely sparse 10-view setting, suggesting that C-DPS maintains strong data consistency even under severe measurement limitations.

**Super-Resolution.**    On the fastMRI brain dataset, Table 10 demonstrates that C-DPS achieves the highest reconstruction quality for both $2\times2$ and $4\times4$ super-resolution tasks. Compared to DPS and DDS, our method produces sharper images with higher structural similarity, particularly in the more difficult $4\times4$ setting. These results affirm the generalization capability of C-DPS across different types of inverse problems.

**Overall Observations.**    Across all tasks, C-DPS improves upon existing baselines without requiring task-specific training or extensive parameter tuning. The results highlight the advantage of incorporating measurement-aware updates through our bi-level guidance framework, which enhances both reconstruction accuracy and robustness under limited data.

Table 9: Quantitative results for the fastMRI knee dataset across different sampling masks and acceleration rates.

| Method | Uniform1D $4\times$ ACR | | Uniform1D $8\times$ ACR | | Gaussian1D $4\times$ ACR | | Gaussian1D $8\times$ ACR | |
|---|---|---|---|---|---|---|---|---|
| | PSNR↑ | SSIM↑ | PSNR↑ | SSIM↑ | PSNR↑ | SSIM↑ | PSNR↑ | SSIM↑ |
| DPS [7] | 32.40±2.19 | 0.843±0.063 | 31.07±2.32 | 0.804±0.073 | 34.93±1.90 | 0.882±0.063 | 33.72±1.97 | 0.853±0.071 |
| Score-MRI [2] | 31.95±1.45 | 0.812±0.036 | 27.97±2.03 | 0.738±0.053 | 33.96±1.27 | 0.858±0.028 | 30.82±1.37 | 0.762±0.034 |
| DDS [46] | 33.83±2.54 | 0.859±0.045 | 32.09±2.84 | 0.822±0.060 | 37.61±2.29 | 0.900±0.045 | 35.82±2.42 | 0.874±0.052 |
| C-DPS (ours) | **35.63±2.47** | **0.877±0.057** | **33.33±2.66** | **0.842±0.077** | **38.05±2.43** | **0.918±0.0428** | **36.52±1.88** | **0.892±0.051** |

Table 10: Super-resolution results on the fastMRI brain dataset.

| Method | $2\times2$ SR | | $4\times4$ SR | |
|---|---|---|---|---|
| | PSNR↑ | SSIM↑ | PSNR↑ | SSIM↑ |
| DPS [7] | 35.44±3.71 | 0.931±0.027 | 30.29±2.84 | 0.854±0.034 |
| DDS [46] | 36.15±3.75 | 0.943±0.023 | 32.04±2.92 | 0.869±0.031 |
| C-DPS (ours) | **36.42±3.02** | **0.951±0.044** | **32.48±2.32** | **0.889±0.051** |

Table 11: Quantitative results of sparse-view CT reconstruction on the LIDC dataset using 350 NFEs.

| Method | 23 Projections | | 10 Projections | |
|---|---|---|---|---|
| | PSNR↑ | SSIM↑ | PSNR↑ | SSIM↑ |
| ScoreMed [26] | 35.24±2.71 | 0.905±0.046 | 29.52±2.63 | 0.823±0.061 |
| C-DPS (ours) | **35.92±2.33** | **0.925±0.046** | **30.59±2.21** | **0.842±0.058** |

