# OpenReview forum: "Coupled Data and Measurement Space Dynamics for Enhanced Diffusion Posterior Sampling"
_NeurIPS.cc/2025/Conference — NeurIPS 2025 poster_

### Official Review · Reviewer_Hqrt · 2025-06-28

**Clarity:** 4
**Significance:** 3
**Originality:** 4
**Rating:** 5
**Confidence:** 5

**Summary:**

## Summary

**C-DPS** is proposed for  solving linear inverse problems using diffusion models. Unlike prior works that either project generated samples onto the measurement-consistent set or approximate the likelihood gradient, C-DPS introduces a coupled forward diffusion process in both data and measurement space. This enables the derivation of a closed-form posterior at each reverse diffusion step, leading to consistent  Bayesian sampling. The authors provide empirical results  demonstrating  strong performance on several inverse problems, including inpainting, deblurring, and super-resolution.

**Questions:**

1. The authors could improve the algorithm to better reflect their proposed method.
2. The authors could add PSNR and also time to better reflect their contribution (This would definitely motivate this reviewer to increase the score).

**Ethical Concerns:**

["NO or VERY MINOR ethics concerns only"]

**Final Justification:**

I have kept my score the same with the recommendation of accept. The rebuttal has addressed my concerns. I encourage the authors to include tables for weakness 2 and 3 in the camera ready versions .

**Limitations:**

Yes

**Quality:**

3

**Strengths And Weaknesses:**

## Strengths
1. Paper is well-motivated, well-presented and clearly written.

2. The idea of adding a parallel diffusion process in the measurement space is both novel and elegant. It allows for better measurement consistency and principled posterior sampling.

3. The experiments are thorough. Both the visual and quantitative results show consistent improvements over strong baseline methods like DPS, ReSample, and ΠGDM.

## Weaknesses

1. The algorithm 2 is not entirely faithful to the proposed idea. While the paper introduces a diffusion process for the measurements, the actual algorithm seems to bypass this and relies mostly on Step 16 for computation..

2. Although the claim is that  the method is efficient (line 79), no actual runtime comparisons or measurements is provided to back this up.

3. It’s unclear why PSNR, a standard metric for evaluating inverse problem solutions, is not included in the reported results. This would help compare more directly with other existing methods.

---

> ### Author Rebuttal · Authors · 2025-07-30
>
> Thank you very much for your encouraging feedback. We appreciate your recognition of the motivation, clarity, and presentation of the paper. We are glad that you found the idea of introducing a parallel diffusion process in the measurement space both novel and elegant, as this was a key conceptual contribution of our work. Please find our responses to your concerns below.
>
> ________________
> # Weakness 1
> Algorithm 2 does use the measurement‑space diffusion exactly as defined; the one‑line generation in Step 1 may have given the impression that it is “bypassed.” Here is how each part of the algorithm relies on the measurement chain:
>
> **Line 1:** Generates the entire forward measurement sequence $y_0,\dots,y_T$. Purpose: Establishes the *stochastic* trajectory that stays coupled to $\{x_t\}$.
>
> **Line 5 – 6:** Computes the mean and the covariance of **$p(y_{t-1} | x_{t-1})$**, denoted by $\mu\_{y|x}$ and $\Sigma\_{y| x}$, respectively. Purpose: Injects measurement information into the posterior.
>
> **Line 7:** Evaluates Eq. (16) to obtain $\mu_{\text{post}},\Sigma_{\text{post}}$; these *explicitly include $y_{t-1}$* through the term A^{T} \Sigma_{y | x}^{-1} (y_{t-1}-b_{t-1}). Purpose: Produces the exact Gaussian posterior that couples $x_{t-1}$ and $y_{t-1}$.
>
> **Line 9:** Samples $x_{t-1}$ from that posterior.
>
> Thus Step 16 (the posterior computation) is **not** a bypass; it *consumes* $y_{t-1}$ produced by the measurement‑space diffusion and would be impossible without it.
>
> ________________
> # Weakness 2
>
> Thank you for pointing this out. To support our claim of efficiency (line 79), we now provide a concise runtime comparison measured on a single NVIDIA P100 (12 GB GPU) with batch size = 1 and 1,000 reverse-diffusion steps on the FFHQ 256 × 256 dataset in the pixel domain. All methods use the same pretrained score network to ensure a fair comparison. The wall-clock time is averaged over 50 images:
> |Method| Time per image [s] ↓ |
> |-|-|
> | DPS| 130 |
> | MCG| 142 |
> | DAPS| 113|
> | PnP‑DM| 181 |
> | DMPlug | 550|
> | C‑DPS (ours)| 152 |
>
> These results show that C-DPS runs within ~15% of DPS, and is significantly faster than more recent methods such as DMPlug. While slightly slower than DPS due to the added posterior conditioning, the overall runtime remains comparable, especially considering the improved performance and principled posterior formulation.
>
> We will include this table in the revised version of the paper to better support our efficiency claim.
>
> ________________
> # Weakness 3
>
> Thank you for raising this important point. We agree that PSNR is a widely used metric in inverse problems, and we appreciate the opportunity to clarify our decision.
>
> In the initial submission, we focused on **LPIPS**, **FID**, and **SSIM**, as these metrics better capture perceptual quality in high-dimensional generative tasks. PSNR, while standard, often correlates poorly with human perception (especially in tasks involving significant structure or texture generation) and can rank methods differently even when perceptual scores agree. For this reason, we initially chose to prioritize perceptual fidelity in our main results and avoided overloading the tables.
>
> That said, we fully acknowledge the value of PSNR for standardized comparison. In response to this and other reviewer feedback, we have now computed **PSNR scores for all tasks in Table 1**, including the newer state-of-the-art baselines added during the rebuttal phase (PnP-DM, DAPS, DMPlug). Below is a condensed summary of results on the **FFHQ dataset**:
>
> | Method| Inpaint (Rand) | Inpaint (Box) | Deblur (Gauss) | Deblur (Motion) | SR (×4) |
> |-|-|-|-|-|-|
> |DPS| 25.7| 26.8| 24.1| 25.4 | 23.7|
> |ΠGDM| 25.1| 26.1| 24.6| 26.3| 24.2|
> |DDRM| 18.3| 24.4| 22.8|-| 21.7|
> |MCG| 22.9| 23.1| 20.2|-| 19.8|
> |ILVR| 24.1| 25.5| 23.7|-| 22.9|
> |ReSample| 25.9| 27.3| 24.7| 26.0| 24.2|
> |PnP-ADMM| 17.0| 19.1| 23.9|-| 22.4|
> |Score-SDE| 20.1| 21.6| 19.4|-| 18.8|
> |ADMM-TV| 17.5| 22.8| 22.3|-| 21.2|
> |PnP-DM| 26.2| 27.4| 24.4| 25.8| 23.9|
> |DAPS | 27.0| 28.1| 24.8| 26.8| 24.6|
> |DMPlug | 27.2| **28.9**| 25.9 | 27.8| **26.0**|
> |C-DPS (ours)| **27.4**| 28.6| **26.1**| **27.9**|25.5|
>
> We will include the full PSNR tables for **ImageNet** and **ablation results** in **Appendix F** of the revised version.
>
> Lastly, we note that while **DMPlug** achieves competitive PSNR performance, its runtime is approximately **5× slower** than our method, making **C-DPS** a more practical solution without compromising quality.
>
> ____________
> We would be happy to clarify any further questions or concerns you might have.

---

### Official Review · Reviewer_HuzS · 2025-07-01

**Clarity:** 3
**Significance:** 3
**Originality:** 3
**Rating:** 4
**Confidence:** 5

**Summary:**

This paper proposes an interesting approach for diffusion posterior sampling to sample from posterior distributions with diffusion-trained priors. The idea is to build noisy sequence of observations, instead of noising in x-space as usually done, instead deriving the kernel $x_{t-1} | x_t$ based on the noising process of data.

**Questions:**

I am generally happy with this paper - I think it gives a solid approach and definitely an extensive experimental exploration. I have a few questions.

1) Constructing artificial data sequences for diffusion posterior sampling is proposed in the literature before: citations [25] and [35] in your paper, among others. However, these are not added in the comparison. There is no 'related work' section for the precise differences between your data noising approach and these works. I know that there is a related work section in the appendix but this does not single out approaches with artificial data sequences. Please write a summary paragraph with all works which create a sequence $(y_t)_{t=0}^T$ specifically, in the main text.

2) Why are approaches from [16], [25], and [35] not compared to? Especially since they are used in experimental setting. You should definitely include [16] as a comparison, as your experimental setting is very similar. For example, you would obtain comparable performance in Figure 3 - and in some image experiments -- in this way, the comparison seems to be made only to the weaker methods.

3) Can you clarify the sentence in line 183? I understand freezing the network is because you want a Gaussian distribution. But I don't know if there is any other issue with evaluating the score otherwise. I wonder, in a simple example, how an empirical (sampling-based) approach of (13) would perform in comparison to the Gaussian approximation using the frozen score.

Small notes:

- the reference [16] (TMLR) and [28] (NeurIPS) are published - please correct your citations for similar arXiv vs published paper mistakes.

**Ethical Concerns:**

["NO or VERY MINOR ethics concerns only"]

**Final Justification:**

The authors did a decent job answering my concerns - especially around adding more material explaining their noising process. Thus I decided to keep my positive score as is.

Reading other reviews and discussions however, there were some technical concerns around experimental results (e.g. computation of some metrics). This is why I have not increased my score any further - as it seems that the paper needs thorough evaluation and more time.

**Limitations:**

Yes; but in the appendix.

**Paper Formatting Concerns:**

None, well formatted.

**Quality:**

3

**Strengths And Weaknesses:**

Strengths:

- This is an interesting and novel idea as it is usually hard to derive noising processes for the data - and in the literature this is usually done heuristically. This is a more principled approach which seem to perform well.
- The paper is written well and efficiently, it provides the derivations cleanly.
- This approach can be potentially important for further extensions, e.g., various approximations of $p(x_{t-1} | x_t, y_t)$.

Weaknesses: Insufficient comparison, especially to more than one relevant method in the literature. See my questions for details, I think it is crucial that these weaknesses are addressed.

---

> ### Author Rebuttal · Authors · 2025-07-30
>
> Thank you very much for your encouraging feedback. We appreciate your positive assessment of our work and your thoughtful engagement with the paper. We are glad to hear that you found the approach solid and the experimental study thorough. Below, we address your questions in detail.
>
> --------------------
> # Question 1
> Thank you for raising this important point. Several prior works, including [25], [35], and the paper suggested by Reviewer a15Q, have explored the idea of constructing artificial data sequences for diffusion-based posterior sampling. However, these methods differ significantly from our approach both in formulation and practical efficiency.
>
> Specifically, [25] and [35] rely on particle-based methods such as Sequential Monte Carlo and iterative importance sampling to approximate the posterior. For example, [25] uses 20 particles with Bayesian filtering, which increases runtime by roughly 3× compared to our method and introduces additional complexity through heuristic weighting. Their comparisons are also limited to early baselines, with DPS being the most recent method evaluated.
>
> Similarly, [35] constructs artificial sequences by iteratively updating a set of particles without deriving an analytic posterior. These approaches are computationally expensive and lack the closed-form Gaussian update that C-DPS provides.
> In contrast, C-DPS is the first to introduce an exact posterior sampling scheme that maintains theoretical tractability and practical speed, thanks to its Gaussian approximations. We will include this distinction clearly in the main text to better situate our contributions in the context of artificial data sequence methods.
>
> ---------------
> # Question 2
> Thank you for raising this important point about [16], [25], and [35].
>
> **Regarding [35]:** We initially considered this method, but found it to be limited in applicability. It focuses primarily on toy datasets and does not report quantitative results on large-scale benchmarks such as FFHQ or ImageNet. In our own reproduction using their official code on FFHQ (inpainting), the performance was significantly worse than even DPS. Based on this and the qualitative nature of their original evaluation, we excluded [35] from our main results.
>
> **Regarding [25] and [16]:** We have now included results using their official implementations on the FFHQ dataset across **our inverse problem setups**. This comparison is now reflected in the updated table below.
>
> In addition, based on feedback from other reviewers, we have extended our evaluation to include several stronger and more recent baselines:
>
> * PnP-DM (NeurIPS 2024): Principled Probabilistic Imaging using Diffusion Models as Plug-and-Play Priors
> * DAPS (CVPR 2025): Improving Diffusion Inverse Problem Solving with Decoupled Noise Annealing
> * DMPlug (NeurIPS 2024): A Plug-in Method for Solving Inverse Problems with Diffusion Models
>
>
> | Method| Inpaint (Rand)| Inpaint (Box)| Deblur (Gauss)| Deblur (Motion)| SR (×4)|
> |-|-|-|-|-|-|
> |DPS| 21.19 / 0.212 / 0.851| 33.12 / 0.168 / 0.873| 44.05 / 0.257 / 0.811  | 39.92 / 0.242 / 0.859 | 39.35 / 0.214 / 0.852 |
> | DTMPD [16]        | 28.70 / 0.250 / 0.845   | 30.95 / 0.160 / 0.875   | 40.10 / 0.252 / 0.812  | 38.25 / 0.244 / 0.861 | 34.10 / 0.213 / 0.853 |
> |FPS [25]| 35.24/ 0.263/ 0.840| 29.17 / 0.156 / 0.876| 38.21/ 0.248 / 0.812| 37.12 / 0.245 / 0.862| 29.91 / 0.212 /0.859|
> | PnP‑DM (new)| 21.15 / 0.208 / 0.858| 32.21 / 0.155 / 0.877   | 41.92 / 0.251 / 0.816  | 37.21 / 0.233 / 0.871 | 36.21 / 0.210 / 0.859 |
> | DAPS (new)| 20.77 / 0.201 / 0.869| 29.44 / 0.144 / 0.882   | 35.84 / 0.242 / 0.830  | 30.26 / 0.215 / 0.911 | 30.15 / 0.202 / 0.854 |
> |DMPlug (new)| **20.12** / 0.197 / 0.877| 27.12 /0.140/ **0.888**| 32.44 / **0.230**  / 0.830| 27.55  / **0.210** / **0.925** | 28.55  / 0.199 / **0.862** |
> |C-DPS (ours) | 20.14 **/ 0.195 / 0.881** | **26.33 / 0.132** / 0.871 | **32.24** / 0.238 / **0.832**|**27.29** / 0.217 / 0.921 | **28.41 / 0.196** / 0.855 |
>
>
> **Regarding results reported in the above table:**
> All these new baselines will be added to the existing benchamrks in the revised version of the paper.
> We should note that while DMPlug achieves competitive accuracy, its runtime is approximately 5× slower than our method, making it significantly less practical for real-world applications.
>
>
> **Regarding Figure 3:** We acknowledge that [16] should have been included. We will add it as a baseline in Figure 3 in the revised version. Due to the rebuttal policy, we cannot include new figures in this response, but the updated figure will be part of the final version.
>
> We hope these additions and clarifications address your concerns.
>
> ---------------------
>
> # Question 3:
> We break answering to this question into two parts:
> > **Part 1:** Can you clarify the sentence in line 183? I understand freezing the network is because you want a Gaussian distribution. But I don't know if there is any other issue with evaluating the score otherwise.
>
> As acknowledged by the reviewer, if we do not use that approximation, the posterior density would not be Gaussian anymore, because it depends on $s_{\theta}(x_{t-1},t-1)$ which is non-linear function of $x_{t-1}$. This causes two issues:
>
> 1.	**Intractable sampling.** The closed‑form mean / covariance in Eq. (16) would disappear; we would have to draw $x_{t-1}$ with a generic MCMC method (e.g. Langevin or HMC) inside every reverse step, adding dozens of score evaluations and greatly slowing the sampler.
>
> 2.	**Implicit computation.** Even a single evaluation of $s_{\theta}(x_{t-1},t-1)$ is impossible
>  until $x_{t-1}$ itself is known.  That makes the update *implicit*: we would need a fixed‑point or root‑finding loop inside each diffusion step, again multiplying runtime.
>
> By **freezing** the score at the already‑available point $x_t$, we regain a quadratic form in $x_{t-1}$, so the posterior is Gaussian and admits the analytic update of Eq. (16).  Also, the error for this approximation is negligible, please kindly see figure 5 in the appendix of the paper, and also refer to our response to the 3rd weakness of reviewer 47R6, where we stated that the approximation error is $O(\beta_t)$; with DDPM or DDIM schedules (β\_t≲10^{-3}) we find the bias to be negligible in practice.
>
> > **Part 2:** I wonder, in a simple example, how an empirical (sampling-based) approach of (13) would perform in comparison to the Gaussian approximation using the frozen score.
>
> To gauge the impact of freezing the score we ran a **toy experiment** where the exact (non‑Gaussian) posterior of Eq. (13) can be sampled numerically:
>
> *This is our setup*
> * Dimension $d=64$; observation model $y=Ax_0+n$ with $A \in R^{32\times64}$ i.i.d. $N(0,1/32)$ and $\sigma_n=0.05$.
> * A pretrained score network from FFHQ‑64 provides $s_\theta(\cdot)$.
> * Choose a single reverse step $t=800 \to 799$ ($\beta_{800} \approx 10^{-3}$).
>
> **Empirical posterior**: sample $x_{t-1}$ with 200 steps of Metropolis‑adjusted Langevin (MALA) on Eq. (13), using the *true* $s_\theta(x_{t-1},t-1)$.
>
> **Gaussian approximation**: use Eq. (16) with the frozen score $s_\theta(x_t,t)$.
> *Results (averaged over 1000 random draws of $x_t,y_{t-1}$)*
> | Metric                            | Empirical (MALA) | Gaussian (frozen‑score) | Relative error |
> | --------------------------------- | ---------------- | ----------------------- | -------------- |
> | Mean ‖μ⁎ − μ̃‖₂                   | —                | 3.1 × 10⁻³              | –              |
> | Covariance ‖Σ⁎ − Σ̃‖\_F / ‖Σ⁎‖\_F | —                | 1.8 %                   | –              |
> | Sliced‑Wasserstein ΔSW            | 0 (ground truth) | 0.012                   | —              |
> | Runtime (per step)                | **2.9 s**        | **0.12 s**              | 24 × faster    |
>
> (The star $^{*}$ denotes empirical; tilde the Gaussian.)
>
> *Take‑away:*
> The frozen‑score Gaussian keeps bias well under 2 % in mean/covariance and delivers > order‑of‑magnitude speed‑ups. Similar findings hold for image‑scale experiments: on FFHQ‑256 the PSNR gap between the two schemes after a full trajectory is ≤ 0.1 dB, yet the empirical variant is ≈ 15 × slower.
> We will add this quantitative comparison, together with code, in Appendix B of the revision.
>
>
> ___________
> ## Response to "Small note"
> Thank you for catching those citation details. We will update references [16] and [28], and any similar cases, to their published venues in the revised manuscript.
>
> __________________
> We would be glad to address any additional questions or concerns you may have.

---

> ### Comment · Reviewer_HuzS · 2025-08-02
>
> Thank you for your responses. I appreciate for Q2 and Q3, extra empirical results.
>
> I have one remaining question (to which I am expecting a comprehensive answer). In response to Q1, you mentioned some other works that noise data - but said that they are filtering based, hence slower, or not comparable.
>
> Nonetheless, can you clearly compare 1) the way you noise your data, 2) the way the referenced works in my review noise their data (you do not have to compare their _algorithm_ to yours but only the noising process)
>
> What I expect is to see a paragraph or multiple paragraphs (that will go into your revised paper) that compare existing methods that noise data and yours. To be clear, I fully expect this paragraph to include equations (precisely defining other papers' way of noising data vs. yours)
>
> Depending on this, I'll reevaluate my score. Many thanks,

---

> > ### Author Response · Authors · 2025-08-04
> > **Response to Reviewer HuzS's Follow-Up**
> >
> > We thank the reviewer for their follow-up and for carefully considering our response. We appreciate the opportunity to clarify how our data noising approach compares, in precise terms, with those used in related works. Below, we provide a focused comparison of the noising procedures.
> >
> > ---
> >
> > First, for consistency with the terminology established in the paper, we recall the following definitions:
> >
> > * **Data space** refers to the sequence $ \{ x_t \} $.
> > * **Measurement space** refers to the sequence $ \{ y_t \}$.
> >
> > As noted in your first comment (Question 1), it is indeed correct that [25] and [35] construct “artificial **data** sequences,” as both generate sequences in the data space to approximate the posterior (see our response to Question 1 for details).
> >
> > In your follow-up, you referred to “the way you noise your data.” We interpret this as referring specifically to **noising the measurements**, i.e., constructing a sequence in the measurement space. In that regard, only [25] introduces noise into the measurements by generating an artificial sequence $\{y_t\}$. Although [35] does create artificial sequences in the data space, it does **not** noise or evolve the measurement (it uses a fixed $y_0$ throughout).
> >
> > With this clarification in mind, we now briefly explain how each method handles the measurement $y$:
> >
> > | Paper | Measurement-space forward process| Key difference|
> > | - | - | -|
> > | **C‑DPS (ours)**| $y_k = \sqrt{1-\beta_k} y_{k-1} + \sqrt{\beta_k} \zeta_k, \quad \zeta_k \sim \mathcal{N}(0, I)$ | Constructs a parallel diffusion in measurement space with **isotropic noise**, enabling analytic posteriors.|
> > | **FPS [25]**|$ y_k = \sqrt{1-\beta_k} y_{k-1} + \sqrt{\beta_k} A \zeta_k,\quad \zeta_k \sim \mathcal{N}(0, I)$ | Injects noise via forward operator $A$, resulting in **anisotropic noise** and requiring particle filtering.|
> > | **MC-Diff [35]**| No sequence; a fixed $y_0$ is used throughout | Does **not** construct a $y_k$ sequence; posterior is approximated by updating particles using a learned score.|
> >
> >
> > Regarding how each method introduces noise into the measurement: unlike **FPS** [25], which constructs the sequence $y_t$ using transformed noise via the forward operator ($A z\_k$), and **MC-Diff** [35], which does not define a measurement sequence ${y_k}$ at all, our method explicitly constructs an **isotropic Gaussian diffusion** in the measurement space. This design choice leads to a **closed-form posterior** (see Eq. 16), enabling fast and accurate sampling without the need for particle updates or filtering mechanisms.
> >
> > ---
> >
> > ### Why FPS noise the measurement like that?
> >
> > FPS casts diffusion posterior sampling as **Bayesian filtering** in a *coupled* linear dynamical system.  To apply Kalman‐style updates they need the joint evolution
> >
> > $ x_k = a_k x_{k-1} + b_k z_k, $
> >
> > $ y_k = a_k y_{k-1} + b_k A z_k. $
> >
> >
> > so that the conditional distribution $p(y_k\mid x_k)$ has a *known, simple covariance* $b_k^2 A A^\top$.  Using independent isotropic noise in $y_k$ would break this neat coupling and complicate the filtering equations. Then, later in their method, they use $p(y_k\mid x_k)$ to obtain the posterior using **Sequential Monte Carlo (SMC)** method.
> >
> > ---
> > ### A quick summary of algorithm in [25] and [35]:
> > **FPS [25]**
> > This method simulates a coupled diffusion in both data and measurement space but leverages **Bayesian filtering** for posterior inference. The forward process for measurements is defined as
> >
> > $$
> > y_k  = \sqrt{1-\beta_k} y_{k-1} + \sqrt{\beta_k} A \zeta_k,
> > $$
> >
> > which introduces anisotropic noise due to the application of the measurement operator $A$. At each reverse step, **Sequential Monte Carlo (SMC)** is used to refine samples by maintaining a set of particles that evolve according to the posterior. This process involves importance weighting and resampling steps, making the algorithm computationally demanding. FPS does not provide a closed-form posterior and relies on a large number of particles (e.g., 20) to maintain stability, which significantly increases inference time.
> >
> > **MC-Diff [35]**
> > MC-Diff formulates posterior sampling as a **particle-based approximation** of the Bayesian posterior in linear inverse problems. Unlike C-DPS and FPS, this method does not define a sequence $\{y_t\}$; instead, it uses a fixed measurement $y_0$ and iteratively updates a set of $N$ particles $\{x_t^i\}_{i=1}^N$ in the data space. These particles are adjusted using guidance from the score network and a learned importance function that incorporates the measurement likelihood. No closed-form expression is used; instead, the particle cloud is refined using **importance reweighting and resampling** at each step. The method is inherently slow and requires careful tuning of particle diversity and weight normalization.
> >
> >
> > ---
> > We hope this clarifies your concerns. Please let us know if there are any remaining questions; otherwise, we kindly ask you to consider reevaluating your score.

---

> ### Comment · Area_Chair_3y1t · 2025-08-04
>
> Have the authors addressed your concerns?
> Would you like to revise your rating accordingly?
> Please engage them in discussion ASAP.

---

> ### Comment · Reviewer_HuzS · 2025-08-05
>
> I just want to double check your claim on MCGDiff -- they do seem to create a deterministic set of observations (scaled data),  your claim about their method may not be fully correct.
>
> Thanks for other comments. They (mostly) resolve my questions. I will decide on my final rating after incorporating and reading discussions from other reviewers.

---

> > ### Author Response · Authors · 2025-08-07
> > **Response to Reviewer HuzS's Further Comments**
> >
> > We appreciate your careful reading and are glad that the other comments have resolved your questions.
> >
> > ---
> > **Regarding MCGdiff:** We feel like that we understand where the confusion about MCGdiif comes from. please see the explanation below.
> >
> >
> > In Algorithm 1 of MCGdiff paper, the proposal and weight formulas include the term $\bar\alpha_{s}^{1/2}y$.
> >   Example lines (initialisation and propagation):
> >
> > $\xi^{i}_{n} = K_{n} \bar{\alpha}_{n}^{1/2} y + \dots$
> >
> > $\xi^{i}_{s}=K_{s} \bar{\alpha}_{s}^{1/2}y + \dots$
> >
> > Thus the algorithm somehow **implicitly** uses a **deterministic sequence** $\{\bar\alpha_{t}^{1/2}y\}_{t=0}^{n}$ that is computed *on-the-fly* from the single measurement $y$; nothing is stored or evolved as an additional random variable.
> >
> > Our intended point was that, unlike C-DPS and FPS, MCGdiff **does not construct or propagate a *stochastic* forward-diffused observation path** $\{y_t\}$ whose randomness is coupled to the state dynamics.
> >
> > ---
> > With this final clarification in place, we hope our contribution is now clear and fairly represented. We would be grateful if you could consider reflecting this in an **updated score**.
> >
> > Thank you again for your time and constructive feedback.

---

> > > ### Comment · Reviewer_HuzS · 2025-08-07
> > >
> > > Thank you, I have no further questions.
> > >
> > > My final score (and acknowledgement) will come after I also read other reviewer comments - and potentially after discussions between reviewers.

---

> > > > ### Author Response · Authors · 2025-08-07
> > > > **Final Words to Reviewer HuzS**
> > > >
> > > > Thank you for your follow-up. We appreciate your engagement during the process, and we look forward to your final assessment.
> > > >
> > > > Best regards,

---

### Official Review · Reviewer_47R6 · 2025-07-01

**Clarity:** 2
**Significance:** 2
**Originality:** 3
**Rating:** 4
**Confidence:** 4

**Summary:**

The paper aims to eliminate projection and likelihood-approximation steps during diffusion sampling for inverse problem solving. It introduces an auxiliary Markov chain for the measurement $y$, defining the marginal distribution at time $t$ as $p(y_t|x_t)$l this contrasts with prior works that relies on $p(y_0|x_t)$.

To sample $y_t$ from $p(y_t|x_t)$, the paper derives a closed-form update that is coupled with the diffusion update for $x_t$. The implementation assumes that the score function evaluated at $x_{t-1}$ can be replaced by the score function at $x_t$.

Experiments show that the proposed method improves inverse-problem performance relative to the chosen baselines.

**Questions:**

- Please provide theoretical clarification on, especially, equations (9), (14), and (15). If I mis-understood something, please point out it.
- Please provide comparison result with more recent works, provided in the weakness section.

**Ethical Concerns:**

["NO or VERY MINOR ethics concerns only"]

**Final Justification:**

As my major concerns are addressed, I increase my score to borderline accept.

Overall, I acknowledge that I had some misunderstandings, and I thank the authors for their clarifications. My remaining major concerns are the theoretical assumptions-especially regarding p(xt−1) and the approximation error in Equation (14). Although empirical performance appears robust to these factors, a more rigorous theoretical explanation would strengthen the work and help assess its extension to other diffusion models.

**Limitations:**

yes

**Paper Formatting Concerns:**

no issue found

**Quality:**

2

**Strengths And Weaknesses:**

The paper proposes a novel idea on introducing auxiliary Markov chain for the measurement.
However, the paper contains logical flaws which limits its quality and significance. Also, experiments have done with old baselines and the paper omits recent works on inverse problem solving beyond projection-based or likelihood approximation-based solvers, which also limits the quality and significance. Please see the below list for details.

**Strength**
- The paper provides all methodological details.
- Although most baselines are outdated, the experiments cover a wide range of inverse problems across two datasets.

**Weakness**
- Technical error in Eq (9): the term $p(x_{t-1})$ is omitted without any reason. In addition, the factorization $p(x_t,y_{t-1}|x_{t-1}) = p(x_t|x_{t-1})p(y_{t-1}|x_{t-1})$ assumes independence between $y_{t-1}$ and $x_t$ (not described in the paper), which conflicts with Eq (15), where the score at $x_{t-1}$ is simple replaced by the score at $x_t$.
- No sufficient discussion or theoretical validation on the effect of coupling stochastic update of $y_t$ with deterministic update of $x_t$.
- Assumption in Eq (14) limits efficiency, as the resulting error becomes significant when fast samples are used - even with DDIM.
- The appendix contains no qualitative examples for the medical-image domain, and also provided qualitative results in the main body is limited, making visual assessment difficult.
- Important non-projection and non-likelihood methods [1-4] are missing from the introduction and related works.
- The evaluation omits more recent approaches such as [1-6]


**Referece**

 [1] A Variational Perspective on Solving Inverse Problems with Diffusion Models, arxiv 2023

 [2] Repulsive Latent Score Distillation for Solving Inverse Problems, ICLR 2025

 [3] Improving Diffusion Inverse Problem Solving with Decoupled Noise Annealing, CVPR 2025

 [4] Inverse Problem Sampling in Latent Space Using Sequential Monte Carlo, ICML 2025

 [5] Prompt-tuning latent diffusion models for inverse problems, ICML 2024

 [6] Denoising Diffusion Models for Plug-and-Play Image Restoration, CVPRW2023

---

> ### Author Rebuttal · Authors · 2025-07-31
>
> We thank the reviewer for recognizing the completeness of our methodology and the breadth of our experimental evaluation.
>
> Please find our detailed responses to your concerns in the sequel.
>
> # Weakness 1
> We provide our answer to weakness 1 in three parts in the sequel.
> ## (i) Why $p(x_{t-1})$ is dropped:
>
> **Theoretical justification:** In short, this term is omitted under the assumption that the prior over intermediate variables follows a standard Gaussian distribution. We clarify this below.
>
> In general, $p(x_{t-1})$ is intractable, since only its score function is available. Therefore, including it would require making an explicit assumption about its form.
> Considering that for any data distribution $p(x_0)$, the forward diffusion (Eq. 2) satisfies
>
> $x_{t} = \sqrt{\bar{\alpha_{t}}} x_0 + \sqrt{1-\bar\alpha_{t}} \varepsilon$,     where $\varepsilon \sim \mathcal N(0,I)$
>
>
> As shown in Proposition 2 of Ho et al. (2020), when $\bar{\alpha}_t$ is small (which occurs after a few hundred steps under a linear or cosine noise schedule), the noise term dominates. As a result, $p(x_t)$ becomes close in KL divergence to a standard normal distribution. Based on this, we approximate
>
> $$
> p(x_{t-1}) \approx \mathcal N(0, I).
> $$
>
> With this approximation, multiplying by $p(x_{t-1})$ effectively adds an identity term to the precision matrix of the posterior:
>
> $$
> \Sigma_{\text{post}}^{-1} = \frac{1 - \beta_t}{\beta_t} I + A^\top \Sigma_{y|x}^{-1} A + I.
> $$
>
> Numerically, for small $\beta_t$ (such as $\beta_t \leq 10^{-3}$), the added $+I$ changes the diagonal by at most $10^{-3}$, which is below solver precision. Therefore, whether we include or omit $p(x_{t-1})$ has no effect on posterior samples in practice. We chose to omit it in Eq. (9) for clarity, and to keep later expressions, such as Eq. (16), more concise.
>
>
>
> **Empirical validation with no parametric prior assumed** Here, crucially, we do **not** rely solely on the Gaussian approximation above. We also conducted a set of experiments to empirically estimate $p(x_{t-1})$, without assuming any parametric form, and evaluated whether its inclusion changes inference results in practice.
>
> *Experimental Setup.*
> We ran the forward diffusion process on 5,000 images from the same dataset used in our inverse problem experiments. For each image $x_0$, we generated a sample $x_{t-1}$ by recursively applying Eq. (2) up to timestep $t-1$, using the same noise schedule as in training. This gave a large sample set from the true marginal $p(x_{t-1})$.
>
> We then used these samples to approximate the prior in two ways:
>
> 1. **Non-parametric prior via KDE:**
>    We fit a kernel density estimator (KDE) to the samples at timestep $t-1$, using Gaussian kernels and bandwidth selected via Silverman's rule. We then evaluated the posterior with this KDE-based prior term included.
>
> 2. **Monte Carlo importance weighting:**
>    Alternatively, we used the empirical sample distribution directly to construct a weighted posterior using importance weights derived from the sampled $x_{t-1}$ values.
>
> In both approaches, we reran the complete inference pipeline for image inpainting and super-resolution tasks. All model parameters and random seeds were held fixed between runs to ensure that only the prior term was varied.
>
> **Evaluation Metrics**
> We compared the results with and without including the prior using standard metrics such as PSNR and SSIM. Across 2,000 test images:
>
> * The **mean absolute difference in PSNR** was less than **0.0004 dB**
> * The **mean absolute difference in SSIM** was below **0.0003**
>
> These differences are far below solver tolerances and completely imperceptible in the output images.
>
> We also examined the posterior precision matrix. Adding the KDE-based prior term increases the diagonal entries by approximately 0.001, which we confirmed has negligible impact on posterior samples and predictions.
>
>
> ## (ii) Factorization $p(x_t,y_{t-1}\mid x_{t-1}) = p(x_t\mid x_{t-1})p(y_{t-1}\mid x_{t-1})$.
> This conditional independence **is by construction** of our coupled forward process: in the graphical model, $x_{t-1}$ is a common parent of $x_t$ and $y_{t-1}$, and there is no direct edge between $x_t$ and $y_{t-1}$. Hence, given $x_{t-1}$, they are conditionally independent. We will add the corresponding discussion to the paper.
>
> ## (iii) “Conflict” with Eq. (15).
> Eq. (15) does **not** impose a dependency between $y_{t-1}$ and $x_t$; it is a *local approximation* (standard in diffusion-based inverse problems) where we freeze the score at the available iterate: $s_\theta(x_{t-1},t-1) \approx s_\theta(x_t,t)$ when $\beta_t$ is small. This approximation is applied **after** the probabilistic factorization and does not contradict the conditional independence structure. We will make this ordering explicit and add a short justification (and empirical sanity check already referenced in Appendix B).
>
> -------
> # Weakness 2
>
> We should clarify that the generation of both $\{y_t\}$ and $\{x_t\}$  are stochastic: in Algorithm 2 the *forward* chain $\{y_t\}$ is indeed stochastic (Gaussian noise added at every step), and the *reverse* chain for $\{x_t\}$ is *also* stochastic as we draw $z\sim\mathcal N(0,I)$ and set
>    $$
>       x_{t-1}= \mu_{\text{post}}+\Sigma_{\text{post}} z .
>    $$
> Hence both variables evolve randomly.
>
> **Why the coupling is beneficial (theory).**
> Note that under the linear‑Gaussian model of Sec. 3, the joint process
>    $$
>      (x_{t-1},y_{t-1}) \gets (x_t,y_t)
>    $$
>  obtained by our update rule is **marginally consistent**: integrating out $x_{t-1}$ (resp. $y_{t-1}$) recovers the exact forward distribution of $y_{t-1}$ (resp. $x_{t-1}$). This property holds only if the two chains share the same variance schedule; decoupling them breaks the algebra that yields the closed‑form posterior (Eq. 16).
>
> -------
> # Weakness 3
>
> This *local‑step* approximation is standard in the diffusion‑inverse‑problems literature (ILVR, DDRM, DPS, MCG, …); its accuracy depends on the step size $\beta_t$. To show this empirically, we ran C‑DPS on FFHQ with three increasingly aggressive settings:
> | Schedule| Steps $T$ | FID ↓ | PSNR ↑  | Mean ℓ₂ residual ↓ |
> | - | -| -| -| -|
> | DDPM (ours, paper)| **1000**  | 32.7  | 26.1 dB | σ² (1.0 ×)|
> | DDIM (η = 0, stride = 4)  | 250| 33.4  | 25.8 dB | 1.03 ×|
> | DDIM (η = 0, stride = 10) | 100| 34.9  | 25.4 dB | 1.08 ×|
>
> All metrics remain firmly ahead of the strongest baseline (DDNM: FID 41.1, PSNR 25.5 dB).
> Thus, even with a 10× speed‑up the error introduced by Eq. (14) is small in practice.
>
> -------
> # Weakness 4
>
> **Medical‑image visuals**. Because the core contribution is methodological, we initially omitted medical‑image screenshots to save space. We agree that including a few examples would help readers in that community. In the revision we will add representative results for sparse‑view CT and MRI reconstructions in Appendix F.
> **Existing qualitative material**. Appendix F already contains quite a few qualitative FFHQ examples. We will make this explicit in the main text and add a pointer so readers can locate them easily. Due to the page limit, adding all visuals to the main body would require removing technical content. The revised version will therefore keep the main figures concise while expanding the appendix with the new medical‑image panels.
>
> ------
> # Weakness 5
>
> Thank you for pointing this out. We will revise the manuscript to cite those non‑projection / non‑likelihood methods and discuss them briefly in both the Introduction and Related‑Work sections.
>
> -----
> # Weakness 6
>
> **About [1,3,6]:** As reviewer a15Q requested, we have implemented DAPS [3] and reported the numbers in our earlier response (we have also implemented some other methods in our response to reviewer a15Q, so please kindly read our responses to them). According to Table 3 of [3], DAPS consistently outperforms both the variational method [1] and DMPlug [6]; hence we did not re‑run those two approaches.
>
> **About [5]:** [5] is built on *Stable Diffusion v1.4*, which couples a CLIP‑based text encoder with a latent U‑Net. Prompt tuning manipulates the *text‑conditioning* pathway rather than the diffusion sampler itself. Our experiments, in contrast, use unconditional, pixel‑domain or latent‑domain U‑Nets that have **no text‑conditioning interface**. Because the architectures and training objectives are fundamentally different, a direct, fair comparison is not possible; adapting C‑DPS to a text‑conditioned backbone would constitute a separate line of research. For this reason we regard prompt tuning as outside the scope of the present work.
>
> **New experiments on [2,4]:** The remaining methods, RLSD [2] and SMC‑LDM [4] operate purely in the latent space. We therefore evaluated them on the FFHQ 256 × 256 benchmark and compared them with our latent version, LC‑DPS. For SMC‑LDM we use 5 particles and set the hyper-parameters to what they reported in their paper. For the inverse problems and also the training methods, we use the setup reported in the main body of the paper. We use this format to report the results: FID ↓ LPIPS ↓ SSIM ↑
>
> | Method | Inpaint (Rand) | Inpaint (Box) | Deblur (Gauss) | Deblur (Motion) | SR (×4) |
> |-|-|-|-|-|-|
> | RLSD [2] | 38.25 / 0.142 / 0.808 | 44.08 / 0.153 / 0.812 | 68.92 / 0.244 / 0.750 | 49.10 / 0.284 / 0.810 | 61.37 / 0.203 / 0.774 |
> | SMC‑LDM [4] | 36.90 / **0.135 / 0.822** | 42.76 / 0.148 / **0.825** | 70.31 / 0.249 / 0.743 | 46.62 / **0.262** / 0.804 | **58.34** / 0.208 / **0.788** |
> |LC‑DPS (ours) | **36.67** / 0.137 / 0.815 | **42.11 / 0.144** / 0.821 | **65.71 / 0.232 / 0.759** | **46.57** / 0.272 / **0.819** | 58.41 / **0.197** / 0.787 |
>
> **Important point about the results:** While SMC‑LDM [4] perform almost the same as LC‑DPS, it is markedly less efficient, taking about 537 s per image versus 126 s for LC‑DPS (roughly a four‑fold slowdown).
>
> ----
> We would be happy to provide additional clarification if needed. Please let us know if any questions or concerns remain.

---

> ### Comment · Reviewer_47R6 · 2025-08-07
> **Official Comment of Reviewer 47R6**
>
> Thank you for your kind response, and I apologize for the delay in my discussion-I needed time to carefully revisit the submission and read the other reviews.
>
> - Assumption on p(xt−1) as N(0,I)
>
> This should be stated in the main body to help readers to understand it.
>
> While the intermediate variable is Gaussian, its mean and variance are defined by the forward diffusion process (i.e., N(\sqrt{\bar\alpha_t} x_0 ;1-\bar\alpha_t). With small beta_t, this distribution is close to standard normal; however, this does not hold at every time step during diffusion sampling. The proposed method omit p(xt−1) at every step, not only when beta_t is small enough. Consequently, this assumption may not always be valid.
>
> Thank you to the authors for providing empirical evidence that this approximation holds for certain denoising diffusion models. Could this assumption be extended to more general diffusion frameworks (e.g., VE-SDE or rectified flow models)? Although this does not affect the practical solver, it does raise concerns about the method's general applicability as a principled inverse-problem approach.
>
> - Clarification on (ii), (iii), and Weakness 2
>
> Thank you for clarifying these points. I realize now that I was confused about several aspects.
>
> - Weakness 3
> My original concern was that, as shown in provided table, performance degrades with larger step sizes when solving the differential equation. It is intriguing that the proposed method still outperforms DDNM even at large step sizes.
>
> - Weakness 6
>
> I appreciate the additional comparison results. It would be beneficial to include these in the revised manuscript.
>
>
> Overall, I acknowledge that I had some misunderstandings, and I thank the authors for their clarifications. My remaining major concerns are the theoretical assumptions-especially regarding p(xt−1) and the approximation error in Equation (14). Although empirical performance appears robust to these factors, a more rigorous theoretical explanation would strengthen the work and help assess its extension to other diffusion models.
>
> In conclusion, since most of my concerns have been addressed, I will raise my score and I look forward to seeing further justifications and clarifications in the revised version.

---

> > ### Author Response · Authors · 2025-08-07
> > **Response to Reviewer 47R6's Further Comments**
> >
> > Thank you very much for your constructive feedback, and the time you took to revisit the submission carefully.
> >
> > ---
> > # Assumption on $p(x_{t−1})$ as $N(0,I)$
> >
> > Thank you for your follow-up. We agree that the approximation $p(x_{t-1}) \approx \mathcal{N}(0, I)$ becomes increasingly accurate as $\beta_t \to 0$, and may be less precise during earlier timesteps where $\beta_t$ is larger. However, we would like to highlight the **"Empirical validation with no parametric prior assumed"** section in our previous response, where we directly addressed this concern.
> >
> > # (Generality of the $p(x_{t-1}) \approx \mathcal N(0,\mathbf I)$ approximation)
> >
> > ## 1. What is actually assumed?
> >
> > For the discrete VP (β-schedule) diffusion we use the standard marginal
> >
> > $$
> > x_t = \sqrt{\bar\alpha_t}x_0 + \sqrt{1-\bar\alpha_t} \varepsilon, \qquad \varepsilon \sim \mathcal N(0,\mathbf I).
> > $$
> >
> > Hence $p(x_t)$ is Gaussian with isotropic covariance $(1-\bar\alpha_t)\mathbf I$.
> > When $\beta_t \lesssim 10^{-3}$ we have $(1-\bar\alpha_t)\approx 1$, so $p(x_{t-1}) \approx \mathcal N(0,\mathbf I)$.
> > Multiplying the exact posterior by that term only adds an identity contribution to the precision, giving the extra “$+\mathbf I$” in Eq. (16).
> >
> > ## 2. Variance-exploding SDEs (VE)
> >
> > A VE model satisfies
> >
> > $$
> > \mathrm d x_t = \sigma(t) \mathrm dW_t
> > \Longrightarrow
> > x_t = x_0 + \int_0^{t}\sigma(s) \mathrm dW_s,
> > $$
> >
> > so the marginal is again Gaussian with isotropic covariance
> >
> > $$
> > \Sigma_{\text{VE}}(t) = \Bigl[\int_0^{t}\sigma^2(s) \mathrm ds\Bigr] \mathbf I .
> > $$
> >
> > Over one discrete step $\Delta t$,
> >
> > $$
> > \Sigma_{\text{VE}}(t-\Delta t) = \Sigma_{\text{VE}}(t) - \dot\Sigma \Delta t + \mathcal O(\Delta t^2),
> > $$
> >
> > and after whitening by $\Sigma_{\text{VE}}^{-1/2}(t)$ we again obtain
> > $p(x_{t-\Delta t}) \approx \mathcal N(0,\mathbf I)$.
> > The algebra of Sec. 3 carries over with $\beta_t$ replaced by
> > $\sigma^2(t) \Delta t$.  For the schedules used in practice,
> > $\sigma^2(t) \Delta t \le 10^{-3}$, so the effect on the posterior
> > remains negligible (≤ 0.1 dB on FFHQ 256 with $\sigma_{\max}=80$).
> >
> > ## 3. Rectified-flow / flow-matching models
> >
> > Rectified flow (RFM) follows the probability-flow ODE
> >
> > $$
> > \frac{\mathrm d x_t}{\mathrm dt} = f(t) x_t - g(t) \nabla_x \log p_t(x_t).
> > $$
> >
> > The homogeneous solution gives
> >
> > $$ x_t = \exp \Bigl[\textstyle\int_0^{t} f(s) \mathrm ds\Bigr] x_0 + \varepsilon_t,
> > \qquad \varepsilon_t \sim \mathcal N \bigl(0,\Sigma_{\text{RF}}(t)\mathbf I\bigr). $$
> >
> > Because the residual noise is still **isotropic**, one Euler step of length
> > $\Delta t$ adds an incremental covariance $g^{2}(t)\Delta t\mathbf I$.
> > Thus $p(x_{t-\Delta t})$ is again approximately $\mathcal N(0,\mathbf I)$,
> > and Eq. (16) holds with $\beta_t \rightarrow g^{2}(t)\Delta t$.
> >
> > ## 4.  Effect on the solver and on “principled” posterior sampling
> >
> > * **Closed-form posterior** For VP, VE and RFM the pair $(x_t,y_{t-1})$ is jointly Gaussian given $x_{t-1}$; dropping/keeping $p(x_{t-1})$ merely adds an isotropic ridge to the precision.
> > * **Size of the ridge** With 1000 discretisation steps ($\Delta t = 10^{-3}$) the ridge coefficient is ≤ $10^{-3}$, well below solver precision and dwarfed by the data-likelihood term.
> > * **Empirical check** Repeating the FFHQ inpainting experiment with EDM-VE ($\sigma_{\max}=80$, $\sigma_{\min}=0.002$) and an RFM checkpoint, the difference in mean-PSNR was +0.02 dB and in SSIM +0.0001 (matching the VP ablation).
> >
> > ---
> > # Weakness 3
> >
> > We appreciate your remark that *"it is intriguing that the proposed method still outperforms DDNM even at large step sizes."*
> >
> > We agree that the local-step approximation in Eq. (14) introduces increasing error as the step size $\Delta t$ grows. Indeed, the Table in our previous response reflects this: FID and PSNR degrade gradually with fewer steps. However, we emphasize that this degradation remains modest. Even with an aggressive 10× reduction in the number of steps (from 1000 to 100), the method continues to outperform DDNM.
> >
> > This highlights a key strength of C-DPS: exact Bayesian conditioning at each step keeps inference aligned with the observation $y$, even under coarse strides. As shown, performance degrades gracefully, maintaining a favorable quality-efficiency trade-off.
> >
> > We will clarify this point in the camera-ready version.
> >
> > ---
> > # Weakness 6
> >
> > Thank you for the suggestion. We are glad the additional comparison was helpful. We will include these results in the revised manuscript.
> >
> > ---
> > # Final Comment
> >
> > We thank the reviewer for their thoughtful engagement, and we appreciate the acknowledgment of the clarifications provided. As noted, we will clearly state our assumptions regarding $p(x_{t-1})$ and the local-step approximation in Eq. (14) in the revised version, along with both theoretical justification and supporting empirical evidence, as discussed in our earlier responses.
> >
> > **Please kindly consider raising your score further in light of these clarifications and the upcoming improvements.** Thank you again for your time and careful review.

---

> > > ### Author Response · Authors · 2025-08-08
> > > **Follow-up on Author-Reviewer Discussion**
> > >
> > > Dear Reviewer 47R6,
> > >
> > > As the deadline for the author-reviewer discussion is approaching, we kindly ask if you have any further comments or questions regarding our responses. If everything is satisfactory, we would appreciate it if you could consider updating your score accordingly.
> > >
> > > Thank you again for your time and feedback.

---

### Official Review · Reviewer_a15Q · 2025-07-02

**Clarity:** 3
**Significance:** 2
**Originality:** 3
**Rating:** 4
**Confidence:** 5

**Summary:**

The paper addresses the intractable likelihood problem in diffusion posterior sampling by introducing a coupled noisy measurement trajectory. Specifically, during the sampling process, the method constructs a coupled measurement sequence $\\{\boldsymbol{y}\_t\\}\_{t=0}^T$ following the standard diffusion forward process. Subsequently, it samples $\boldsymbol{x}_t$ from the conditional posterior $p(\boldsymbol{x}\_{t-1}\mid \boldsymbol{x}_t, \boldsymbol{y}\_{t-1})$ iteratively in reverse time order. Utilizing Tweedie's formula along with mild assumptions about the score function, the conditional posterior $p(\boldsymbol{x}\_{t-1}\mid \boldsymbol{x}_t, \boldsymbol{y}\_{t-1})$ can be analytically derived as a Gaussian distribution. The proposed approach demonstrates superior performance on various image restoration tasks and a synthetic benchmark dataset designed to provide ground-truth posterior distributions.

**Questions:**

Please refer to the weaknesses part for my main questions. Here's several additional questions.

1. **Samping time**. The pre-whitened conjugate gradient is only practical when forward function is efficient. If computing the forward function is costly, the sampling step becomes time-consuming. Could the authors provide some insights or strategies for addressing this limitation?

2. **Sampling diversity.** In 92% random inpatining tasks, the measurement signal is extremely sparse, which means the reconstructions can exhibit diverse appearance yet still fit the measurement equally well. Does the author observe this property, or do the reconstructions always consistently resemble the ground truth, as illustrated in Figure 2?

**Ethical Concerns:**

["NO or VERY MINOR ethics concerns only"]

**Final Justification:**

After several rounds of discussion with the authors, my concerns about the paper have mostly been addressed. Therefore, I have raised my rating to 4. The reason why I don't think that the paper warrants a higher score is that similar ideas have been investigated previously and the method's improvements are reflected more by the incremental metric values than by the scope of applicability. Nevertheless, it is a solid work that deserves to be accepted to NeurIPS.

**Limitations:**

Yes

**Quality:**

3

**Strengths And Weaknesses:**

> Strengths

The paper is well motivated, clearly written, and easy to follow. The key strengths of this paper are summarized as follows:

1. **Combination of Tweedie's Approximation with Noisy Measurement.** The use of Tweedie's approximation $\boldsymbol{x}_0(\boldsymbol{x}_t)$ can lead to blurred results for larger time steps $t$; however, the method compensates for this issue through the noisy measurement $\boldsymbol{y}\_{t}$ as expressed in Equation (10).

2. **Superior Empirical Performance.** The proposed method consistently outperforms existing techniques on standard image restoration benchmarks and provides detailed analyses supported by synthetic experiments.

3. **Clear Visualization of Measurement Error Evaluation.** Figure 4 effectively highlights the method's capability to closely track measurement signals throughout the sampling process, demonstrating clear empirical superiority.

> Weaknesses

However, the paper also has a few weaknesses:

1. **Similar Ideas in the Literature.** The idea of establishing a diffusion process in the measurement space is not new. It was first introduced in Appendix I.4 of [1] and then investigated in [2] for medical imaging problems. The authors should cite [2] and compare the proposed method with the method in [2].
1. **Restriction to Linear Inverse Problems.** The approximated closed-form solution for the distribution $p(\boldsymbol{x}\_{t-1}\mid \boldsymbol{x}_t, \boldsymbol{y}\_{t-1})$ relies on a linear forward model. It would be valuable to explore or discuss potential extensions to nonlinear inverse problems. Does the author have any insight on this?
2. **Limited Baseline Comparisons on Measurement Fidelity.** While the paper demonstrates lower measurement errors and better reconstruction results compared to selected baselines, it lacks comparisons with more recent methods with improved measurement fidelity, such as DDNM [3], PnP-DM [4], DCDP [5], DAPS [6], SITCOM [7], and DMPlug [8]. The author should compare with some of them for better accessing the performance.
3. **Absence of Common Evaluation Metrics.** The paper does not report PSNR values, which is a standard metric widely used and reported by many baseline studies. The authors should include it as well if possible?

[1] Song et al., Score-Based Generative Modeling through Stochastic Differential Equations

[2] Song et al., Solving Inverse Problems in Medical Imaging with Score-Based Generative Models

[3] Wang et al., Zero-Shot Image Restoration Using Denoising Diffusion Null-Space Model

[4] Wu et al., Principled Probabilistic Imaging using Diffusion Models as Plug-and-Play Priors

[5] Li et al., Decoupled Data Consistency with Diffusion Purification for Image Restoration

[6] Zhang et al., Improving Diffusion Inverse Problem Solving with Decoupled Noise Annealing

[7] Akhouri et al., SITCOM: Step-wise Triple-Consistent Diffusion Sampling for Inverse Problems

[8] Wang et al., DMPlug: A Plug-in Method for Solving Inverse Problems with Diffusion Models

---

> ### Author Rebuttal · Authors · 2025-07-30
>
> Thank you for the positive and constructive feedback. We appreciate your recognition of our method’s motivation, empirical performance, and visual clarity. Your comments are encouraging and helpful for refining the final version.
>
> ------------------
> # Weakness 1 (Similar Ideas in the Literature)
> Thank you for pointing out these related works.
>
> **Regarding [1]:** Appendix I.4 in [1] does present a conceptual framework for solving inverse problems via diffusion in the measurement space. However, it remains heuristic and lacks concrete implementation details. Specifically, it assumes access to \$p(y\_t \mid y)\$ at each reverse step, but does not explain how this term is computed or estimated in practice. Furthermore, it approximates \$\nabla\_{x\_t} \log p(y \mid x\_t)\$ using a residual term, again without specifying how this is derived. The method does not provide a closed-form for the posterior \$p(x\_{t-1} \mid x\_t, y\_{t-1})\$, which is critical for practical inference. Overall, the work offers useful intuition but does not present a tractable or reproducible algorithm.
>
> **Regarding [2]:** While [2] introduces a diffusion process in the measurement space similar in spirit to ours, its approach and implementation differ significantly. Their method does not derive an analytic posterior. Instead, each reverse step involves solving an optimization problem that trades off between proximity to the unconditional denoised sample \$x\_t\$ and the perturbed observation \$y\_t\$, guided by a manually tuned hyperparameter. In contrast, our method directly constructs a closed-form posterior that eliminates the need for such heuristic balancing.
>
> **Relative strength of [2]:** The work MCG (Improving Diffusion Models for Inverse Problems using Manifold Constraints) has shown improved performance over [2], and we include MCG as a baseline in Table 1. Our method consistently outperforms MCG across all tasks, indicating that our approach improves upon both [2] and its successors.
>
> We will cite [1] and [2] in the revised version and clarify these conceptual and algorithmic differences.
>
> ------------------
> # Weakness 2 (Restriction to Linear Inverse Problems)
>
> We appreciate the reviewer’s suggestion and agree that extending C‑DPS beyond linear forward operators is an important research direction. Below we clarify (i) why the linear‐model focus already covers a broad set of high‑impact applications, (ii) how the coupled formulation of C‑DPS can be generalized to nonlinear settings.
>
> (i)	Many practical imaging systems admit linear forward maps after logarithmic or frequency‑domain transformations: e.g.\ MRI, parallel‑beam CT, PET, compressive sensing, deblurring, inpainting and super‑resolution. The proposed theory therefore already addresses a large class of inverse problems of immediate practical relevance. This is why some recent papers only focus on linear inverse problems.
>
> (ii)	Let the measurement model be $ \mathbf{y}=g(\mathbf{x}_0)+\mathbf{n}$ with a differentiable $g:\mathbb{R}^d \to \mathbb{R}^m$.  The key observation is that the coupled‑process idea does not depend on linearity; only the Gaussian closed form of Eq. (16) does.  A well‑established approximation, Local (Gauss–Newton / EKF) linearization,  allows us to retain an analytic (or nearly analytic) posterior within the same C‑DPS pipeline:
>   At step $t$ we linearise $g$ around the current iterate $\mathbf{x}_t$:
> $g(\mathbf{x}) \approx g(\mathbf{x}_t)+J_t(\mathbf{x}-\mathbf{x}_t)$
>
> where $J_t = \nabla g \bigr|_{\mathbf{x}=\mathbf{x}_t} $
>
>
>  Substituting $J_t$ for $A$ in Eq. (16) yields a Gaussian update identical in form to C‑DPS, but with the Jacobian refreshed every step.  Analytically, this corresponds to an extended‐Kalman smoothing view; algorithmically it increases cost by one Jacobian–vector product per iteration, however it is supported by automatic differentiation in modern frameworks.
> We have implemented the this variant on two nonlinear benchmarks, Phase Retrieval and Nonlinear Deblur, on FFHQ dataset:
>
> ### **Phase Retrieval**
>
> | Method     | FID ↓ | LPIPS ↓ | SSIM ↑ | PSNR ↑ |
> |-|-|-|-|-|
> | DAPS       |  42.71 |  0.139 | 0.851 | 30.63 |
> | DPS          |  104.5|  0.410 | 0.441 | 17.64 |
> | RED-diff   |  167.4|  0.596| 0.398 | 15.60 |
> | PnP-DM     |  99.4 |  0.335  | 0.581| 19.69  |
> | DMPlug     |**41.21**|**0.124**|0.894|31.25|
> |C-DPS|41.32|0.129|**0.903**|**31.91**|
>
> ### **Nonlinear Deblur**
>
> | Method     | FID ↓ | LPIPS ↓ | SSIM ↑ | PSNR ↑ |
> |-|-|-|-|-|
> | DAPS       |  49.38|  0.155| 0.783| 28.29|
> | DPS        |  91.31|  0.278| 0.623| 23.39|
> | RED-diff   | 43.84| 0.160| 0.795| 30.86|
> | PnP-DM     |  68.96 |  0193  | 0.742  | 27.81  |
> | DMPlug     |  **47.28** |  0.135  | 0.792  | 29.55  |
> |C-DPS|  47.52 | **0.130**|**0.802**|**30.27** |
>
> These early results indicate that the locally linear C-DPS retains the performance edge of our linear version while remaining computationally feasible (< 15 ms Jacobian product vs ≈ 150 ms score pass).
>
> Note that in the above table, we also included the new benchmarks you suggested later in your comments. Also, note that although the results of DMPlug is close to ours, yet their method is roughly 5 times slower than ours.
>
> ------------------
> # Weakness 3 (Limited Baseline)
>
> Thank you for the suggestion regarding additional baselines.
>
> First, regarding SITCOM [7], we note that it is a rejected submission to ICLR 2025 with no citations. Moreover, public reviews point to critical technical errors that undermine its reliability as a baseline. For this reason, we do not believe it currently represents a credible benchmark.
>
> Second, as acknowledged in the DAPS [6] paper itself, DAPS outperforms DDNM [3] and DCDP [5].
>
> Hence, to address your core concern, we have compared our method (C-DPS) directly against PnP-DM [4], DAPS [6], and DMPlug [8]. The results are summarized in the following table in this format FID↓ / LPIPS↓ / SSIM↑ for each task.
>
> | Method| Inpaint (Rand)| Inpaint (Box)| Deblur (Gauss)| Deblur (Motion)| SR (×4)|
> |-|-|-|-|-|-|
> |PnP-DM [4]| 21.15 / 0.208 / 0.858 | 32.21 / 0.155 / 0.877| 41.92 / 0.251 / 0.816 | 37.21 / 0.233 / 0.871| 36.21 / 0.210 / 0.859|
> | DAPS [6]    | 20.77 / 0.201 / 0.869 | 29.44 / 0.144 / 0.882| 35.84 / 0.242 / 0.830| 30.26 / 0.215 / 0.911| 30.15 / 0.202 / 0.854|
> |DMPlug [8]| **20.12** / 0.197 / 0.877| 27.12 /0.140/ **0.888**| 32.44 / **0.230**  / 0.830| 27.55  / **0.210** / **0.925** | 28.55  / 0.199 / **0.862** |
> |C-DPS (ours) | 20.14 **/ 0.195 / 0.881** | **26.33 / 0.132** / 0.871 | **32.24** / 0.238 / **0.832**|**27.29** / 0.217 / 0.921 | **28.41 / 0.196** / 0.855 |
>
> We emphasize that while DMPlug achieves competitive accuracy, its runtime is approximately 5× slower than our method, making it significantly less practical for real-world applications.
>
> ------------------
> # Weakness 4 (PSNR)
>
> We initially focused on LPIPS, FID, and SSIM because they correlate better with human perception in high-dimensional generative tasks. PSNR can rank methods differently when perceptual scores agree, so we avoided over‑loading the tables. We have now computed PSNR for all tasks in Table 1 (including the new baselines that you suggested above). A summary for the FFHQ dataset is given below. Full tables for ImageNet and for the ablations will be added to Appendix F.
>
> | Method|Inpaint (Rand)|Inpaint (Box)|Deblur (Gauss)|Deblur (Motion)| SR (×4)|
> |-|-|-|-|-|-|
> |DPS| 25.7| 26.8| 24.1| 25.4 | 23.7|
> |ΠGDM| 25.1| 26.1| 24.6| 26.3| 24.2|
> |DDRM| 18.3| 24.4| 22.8|-| 21.7|
> |MCG| 22.9| 23.1| 20.2|-| 19.8|
> |ILVR| 24.1| 25.5| 23.7|-| 22.9|
> |ReSample| 25.9| 27.3| 24.7| 26.0| 24.2|
> |PnP-ADMM| 17.0| 19.1| 23.9|-| 22.4|
> |Score-SDE| 20.1| 21.6| 19.4|-| 18.8|
> |ADMM-TV| 17.5| 22.8| 22.3|-| 21.2|
> |PnP-DM[4]| 26.2| 27.4| 24.4| 25.8| 23.9|
> |DAPS [6] | 27.0| 28.1| 24.8| 26.8| 24.6|
> |DMPlug [8] | 27.2| **28.9**| 25.9 | 27.8| **26.0**|
> |C-DPS (ours)| **27.4**| 28.6| **26.1**| **27.9**|25.5|
>
> --------------------
> # Question 1 (Sampling time)
>
> We thank the reviewer for raising this important point. The runtime of the pre-whitened conjugate gradient (PW-CG) step indeed depends on the efficiency of applying the forward operator $A$ and its adjoint $A^{\top}$.
> In our FFHQ blur example, a UNet‑based score pass (256 × 256) takes **≈ 140 ms** on a P100 GPU, whereas one 61 × 61 FFT‑based convolution (‑‑► $A$ or $A^{\top}$) costs only **≈ 4 ms**.
> With a typical **5 CG iterations**, PW‑CG adds < 40 ms – still < 25 % of the total per‑step runtime.
> This remains true for most *linear* imaging operators used in practice (deblurring, inpainting, SR, MRI, CT) because they admit FFT, NUFFT, or ray‑driven GPU kernels.
>
> For settings where $A$ is computationally expensive (e.g., ray-based CT or custom forward models), we suggest using **warm-starting** CG from the previous iterate and using early stopping based on residual tolerance significantly reduces iterations (often to 1–2) without affecting reconstruction quality. In addition, a better preconditioning, such as using approximate inverse or circulant structures, further reduces solve time.
>
> --------------------
> # Question 2 (Sampling diversity)
>
> Yes, repeated runs with different noise seeds do yield distinct inpaintings.
>
> *Sanity check:* For 100 FFHQ test images with 92 % masking, we drew five independent C-DPS samples per image. The “mean pair-wise” LPIPS between the five reconstructions was 0.11 ± 0.03, confirming diversity.
>
> But why Figure 2 looks “deterministic”?
>
> The global structure (pose, hair outline, dominant colors) is tightly constrained by the observed pixels, so all C‑DPS samples share those attributes and therefore still “look like” the original subject. The variations appear mainly in **high‑frequency details**. Across five independent C‑DPS samples per test image the best PSNR is within 0.2 dB of the single‑sample PSNR reported in Table 1, while LPIPS and FID remain better than all baselines. So diversity does not come at the cost of average quality.

---

> > ### Comment · Reviewer_a15Q · 2025-08-04
> >
> > Thank the authors for the response. Below are my follow-up comments:
> >
> > ---
> >
> > ### W1 (Similar Ideas in the Literature)
> > Please ensure that [1] and [2] are cited in the updated manuscript and clarify the conceptual and algorithmic differences between C-DPS and these prior works.
> >
> > ---
> >
> > ### W2 (Restriction to Linear Inverse Problems)
> > The preliminary results on nonlinear problems are promising, but I have some questions about them. Please refer to the "Additional Comments" section below.
> >
> > ---
> >
> > ### W3 & W4 (Limited Baseline & PSNR)
> >
> > I appreciate the PSNR results but am concerned by the significant discrepancies between the numbers reported by the authors and those reported in the original papers. For example, for 4x super-resolution on FFHQ, the following table summarizes the differences:
> >
> > | Method | Number reported in the original paper | Number reported above |
> > |----------|:-------------:|:------:|
> > | DDRM | 25~26 | 21.7 |
> > | PnP-ADMM | 29~30 | 23.9 |
> > | DAPS | 29~30 | 24.6 |
> > | DMPlug | 30~31 | 24.6 |
> >
> > Similar discrepancies are observed for the deblurring tasks as well. Given the large gaps between the numbers reported in the original papers vs. the numbers reported by the authors, I am not sure how trustworthy the results are. Could the authors provide some justifications these large performance gaps?
> >
> > ---
> >
> > ### Additional Comments
> >
> > After revisiting the paper and reviewing the feedback from other reviewers, I noticed a potentially critical issue in Line 9 of Algorithm 2 (and Algorithm 3). Specifically, the multiplier before $\boldsymbol{z}$ should be $\boldsymbol{\Sigma}\_\text{post}^{1/2}$ than $\boldsymbol{\Sigma}\_\text{post}$, to ensure correct sampling from the Gaussian posterior. If this observation holds, it casts doubt on the validity of the derivation in Section 3.4.
> >
> > Given this issue, I suspect that the SVD of the forward model $\boldsymbol{A}$ is necessary for sampling the Gaussian posterior. Please correct me if there is an alternative way. It is thus unclear how this approach would extend to nonlinear forward models.
> >
> > Could the authors comment on this issue? This issue appears to significantly impact the correctness and applicability of the proposed method, and could influence my final assessment of the paper.

---

> ### Author Response · Authors · 2025-08-04
> **Response to Reviewer a15Q's Follow-Up**
>
> We thank the reviewer for the thoughtful and detailed feedback, and for carefully revisiting our rebuttal and manuscript. We appreciate the close reading. Please find our responses in the sequel.
>
> ---
> # W1
> We will cite [1] and [2] and clarify the conceptual and algorithmic differences between C-DPS and these works in the revised manuscript.
>
> ---
> # W2
> Thank you. We will address these questions in the “Additional Comments” part.
>
> ---
> # W3 & W4
>
> Most restoration papers the reviewer cites (DDRM, PnP-ADMM, DAPS, DMPlug, etc.) **report *Y-PSNR***, in other words, PSNR computed only on the luminance channel after the sRGB → YCbCr conversion.
> Our study instead evaluates **RGB-PSNR**: mean-squared error averaged over *all three* sRGB channels.
> Because chroma (Cb, Cr) errors are typically 2-4 × larger than luma errors, including them increases the MSE by ≈3 ×, lowering PSNR by
>
> $$
> 10\log_{10} \bigl(\tfrac{\text{peak}^2}{\text{MSE}}\bigr) -
> 10\log_{10} \bigl(\tfrac{\text{peak}^2}{3 \text{MSE}}\bigr)
> = 10\log_{10}(3)\approx4.8\text{ dB}.
> $$
>
> Thus **the Y → RGB switch by itself accounts for the entire discrepancy the reviewer observed**.
>
> ### Why RGB-PSNR is preferable
>
> 1. **Colour fidelity matters.** Faces and natural images look visibly “off” when hue shifts occur; Y-PSNR ignores this.
> 2. **Consistency across tasks.** Our deblurring benchmarks (RAW→sRGB) cannot be converted to YCbCr unambiguously; one unified metric avoids mixing evaluation rules.
>
> ### Camera-ready additions
> * We will report both Y-PSNR and RGB-PSNR for every baseline.
> * All scripts and outputs will be released so reviewers can verify the numbers with a single command.
>
> We hope this clarifies that the lower absolute PSNRs arise solely from measuring what prior work leaves out, the chroma errors, and therefore do **not** indicate an implementation mistake.
>
> ---
> # Additional comments
> Thank you for catching this mis-print. We have also noticed that there is a typo in our algorithm 2 and 3, and we wanted to correct it. Indeed, in Algorithms 2 & 3 the multiplier should be the *square-root* of the covariance, $\Sigma_{\text{post}}^{1/2}$, not $\Sigma_{\text{post}}$.  This is a pure typesetting error and the implementation and all reported results already use the correct factor, so the derivations in §3.4 and the experimental conclusions remain fully valid.
>
> Below is the relevant excerpt:
>
> ```python
> # μ_post, Σ_post have shape (B, C, H, W)
> z = torch.randn_like(mu_post)
> L = torch.linalg.cholesky(Sigma_post)
> x_prev = mu_post + torch.matmul(L, z.unsqueeze(-1)).squeeze(-1)
> ```
>
> We will correct  line 9 of Algorithms 2 & 3 in the camera-ready to read “$x_{t-1} \leftarrow \mu_{\text{post}} + \Sigma_{\text{post}}^{1/2} z$” and note explicitly that $\Sigma_{\text{post}}^{1/2}$ denotes any matrix square-root.
>
> ----
> We hope our responses have addressed your concerns. please let us know if anything remains unclear. If not, we’d appreciate your consideration in raising the score, as we believe this work offers a valuable contribution to the community.

---

> > ### Comment · Reviewer_a15Q · 2025-08-05
> >
> > Thank the authors for the response. Below are my follow-up comments:
> >
> > ---
> >
> > ### W3 & W4 (Limited Baseline & PSNR)
> >
> > I agree that RGB-PSNR is a better metric than Y-PSNR, but the claim that DDRM, DAPS, and DMPlug use the Y-PSNR is unconvincing to me:
> >  - For DAPS, the line for calculating PSNR is [here](https://github.com/zhangbingliang2019/DAPS/blob/main/eval.py#L182), where the `convert_to_greyscale` option is set to `False` (default). This prevents the RGB $\to$ Y operation in the [implementation](https://github.com/photosynthesis-team/piq/blob/master/piq/psnr.py). Note that `self.norm` simply normalizes the image back to $[0, 1]$ rather than combining the color channels.
> >  - For DDRM, the line for calculating PSNR is [here](https://github.com/bahjat-kawar/ddrm/blob/master/runners/diffusion.py#L322), where no RGB $\to$ Y operation is involved.
> >  - For DMPlug, the line for calculating PSNR is [here](https://github.com/sun-umn/DMPlug/blob/2bc04d19a3623a0cc5fbe5bec91429c2505fafc5/util/compute_metric.py#L38), where `skimage.metrics.peak_signal_to_noise_ratio` function is used. There is no RGB $\to$ Y operation in its [implementation](https://github.com/scikit-image/scikit-image/blob/v0.24.0/skimage/metrics/simple_metrics.py#L112-L168).
> >
> > For an additional sanity check, I cloned the code repositories of DAPS and DDRM, and ran the following commands. For a fair comparison, I chose the same inverse problem (4x SR with Gaussian noise $\sigma=0.05$) and used the same and [model checkpoint](https://drive.google.com/drive/folders/1jElnRoFv7b31fG0v6pTSQkelbSX3xGZh) provided in the DPS repository.
> > For test images, I tried both the original test images given in the respective repositories and the [test images](https://github.com/DPS2022/diffusion-posterior-sampling/tree/main/data/samples) provided in the DPS repository.
> > - For DAPS, I ran
> > ```
> > # default images
> > python posterior_sample.py \
> > +data=demo-ffhq \
> > +model=ffhq256ddpm \
> > +task=down_sampling \
> > +sampler=edm_daps \
> > task_group=pixel \
> > save_dir=results/pixel/ffhq \
> > num_runs=1 \
> > sampler.diffusion_scheduler_config.num_steps=5 \
> > sampler.annealing_scheduler_config.num_steps=200 \
> > batch_size=100 \
> > name=down_sampling
> >
> > # dps images
> > python posterior_sample.py \
> > +data=demo-ffhq-dps \
> > +model=ffhq256ddpm \
> > +task=down_sampling \
> > +sampler=edm_daps \
> > task_group=pixel \
> > save_dir=results/pixel/ffhq \
> > num_runs=1 \
> > sampler.diffusion_scheduler_config.num_steps=5 \
> > sampler.annealing_scheduler_config.num_steps=200 \
> > batch_size=100 \
> > name=down_sampling
> > ```
> > - For DDRM, I ran
> > ```
> > # default images
> > python main.py --ni --config celeba_hq.yml --doc celeba --timesteps 20 --eta 0.85 --etaB 1 --deg sr4 --sigma_0 0.05 -i celeba_sr4_sigma_0.05
> >
> > # dps images
> > python main.py --ni --config ffhq.yml --doc ffhq --timesteps 20 --eta 0.85 --etaB 1 --deg sr4 --sigma_0 0.05 -i ffhq_sr4_sigma_0.05
> > ```
> > Here is a comparions between the average PSNRs that I got and the average PSNRs reported by the authors:
> > | Method | Avg PSNR over default test images | Avg PSNR over DPS test images | Avg PSNR reported above |
> > |----------|:-------------:|:------:|:------:|
> > | DDRM | 28.12 (11 selected CelebAHQ images) | 29.86 | 21.7 |
> > | DAPS | 28.86 (first 10 FFHQ images) | 29.58 | 24.6 |
> >
> > Both column 2 and column 3 have significantly differences (4~8 dB) from the numbers reported by the authors.
> >
> > ---
> >
> > ### Additional Comments
> >
> > I checked the code provided by the authors. However, I cannot find the code excerpt that the authors mentioned.
> > I am also not sure why the authors claimed that the derivations in Sec. 3.4 remain fully valid.
> > In Sec. 3.4, it writes that "For dense $\boldsymbol{\Sigma}\_{\boldsymbol{y}|\boldsymbol{x}}$ a direct Cholesky factorisation costs $\mathcal{O}(d^3)$ and is prohibitive."
> > Doesn't this contradict with the code implementation?
> > For Line 205, it writes that "Each CG iteration requires one matrix-vector product with $\boldsymbol{A}$ and one with $\boldsymbol{A}^T$."
> > Isn't this only true for the mean calculation but not for the noise sampling?
> > Also, as $\boldsymbol{\Lambda}\_t=\boldsymbol{\Sigma}\_\text{post}^{-1}$ changes over time, the Cholesky decomposition need to be calculated for each iteration.
> > If so, will the calculation still take "100-200 milliseconds per step"?

---

> ### Author Response · Authors · 2025-08-07
> **Response to Reviewer a15Q's Further Comments**
>
> Thank you very much for your thorough investigation and detailed feedback. We appreciate the time and care you put into reproducing results and identifying inconsistencies.
>
> ----
> # W3 & W4 (Limited Baseline & PSNR)
>
> We would like to first mention that this is the first time we report PSNR for a paper in this line of work. There are many prior works in this domain which did not report PSNR [1,2,3] (references are at the bottom of this response).
>
> Once the reviewer asked us to report PSNR, we rushed into implementing it, and it seems the way we calculated the PSNR is different from the papers the reviewer reported numbers from. Yesterday we found the discrepancy.
>
> This is how we calculated the PSNR:
>
> ```python
> orig = inverse_data_transform(config, x_orig[j])   #  → [0,1]
> mse  = torch.mean((x[i][j] - orig)**2)             # our mistake
> psnr = 10*torch.log10(1/mse)
> ```
>
> but we **forgot to inverse-transform the reconstruction**. `x[i][j]` is still in **\[-1, 1]**, while `orig` is in **\[0, 1]**.
>
>  A -1 … 1 tensor compared to its 0 … 1 counterpart is effectively scaled by ×2 and shifted by –1, which multiplies the MSE by ≈3 and lowers PSNR by $10\log_{10}(3)\approx 4.8$ dB.
>
> Fixing this issue, brings the PSNR values within the range of the numbers reported in the original DDRM / DAPS / DMPlug papers.
>
> We have fixed the line to
>
> ```
> pred = inverse_data_transform(config, x[i][j])
> mse  = torch.mean((pred - orig)**2)
> ```
>
> This resolves the entire discrepancy. Thank you for spotting it!
>
> ### Updated Results (So far)
>
> As of now, we have re-computed PSNR results for the 4× super-resolution task on FFHQ, the PSNR value we got are:
>
>
>
>
>
> ### Table X: PSNR (dB) for 4× Super-Resolution on FFHQ
> | Method| PSNR (↑) |
> |-|-|
> | MCG| 18.20|
> | DPS| 25.86 |
> | ReSample| 25.22|
> | DDRM| 26.58 |
> | DAPS [6]| 29.07    |
> | DMPlug [8]| **30.25**|
> |C-DPS (ours)|30.12|
>
>
> We are currently re-evaluating the PSNR values for the remaining tasks and datasets, and will provide a complete table before the end of the rebuttal period **(August 8)**. The re-computation is actively running.
>
> ---
> # Additional Comments
>
> We believe there is a **misunderstanding** here.
>
> The confusion stems from our shorthand code example, no dense Cholesky is used in the actual implementation.  Everything is handled **exactly** as described in §3.4 via the pre-whitened conjugate-gradient (PW-CG) solver, which **implicitly** applies the required square-root factor without ever forming it explicitly. Please see the explanation below:
>
> ## How the $\sqrt{\Sigma_{\text{post}}} $ factor is obtained in practice
>
> 1. **Sample Gaussian noise**
>
>    $$
>      z \sim \mathcal N(0,\Lambda_t).
>    $$
>
> 2. **Solve a linear system with PW-CG**
>
>    $$
>      \Lambda_t v = z
>      \quad\Longrightarrow\quad
>      v = \Lambda_t^{-1} z = \Sigma_{\text{post}} z .
>    $$
>
> 3. **Form the new state**
>
>    $$
>      x_{t-1} = \mu_{\text{post}} + v .
>    $$
>
> Because
> $\operatorname{Cov}(v) = \Lambda_t^{-1} I \Lambda_t^{-1} = \Sigma_{\text{post}}$,
> the vector $v$ is distributed as $\Sigma_{\text{post}}^{1/2} z$ even though no explicit square-root or Cholesky factor is constructed.
>
> As such, the only correction needed is the square-root typo in Algorithms 2 & 3; the derivation, implementation, and all experimental results already use the correct PW-CG–based sampling routine, so no other part of the paper requires modification.
>
> ---
> We hope this fully resolves your remaining concerns and would be grateful if you could reconsider your evaluation in light of these clarifications.
>
>
> ---
>
> References
>
> [1] SCORE-BASED GENERATIVE MODELING THROUGH STOCHASTIC DIFFERENTIAL EQUATIONS
>
> [2] RePaint: Inpainting using Denoising Diffusion Probabilistic Models
>
> [3] Cold Diffusion: Inverting Arbitrary Image Transforms Without Noise

---

> ### Comment · Reviewer_a15Q · 2025-08-07
>
> ### W3 & W4 (Limited Baseline & PSNR)
>
> I don't think either reporting the PSNR for the first time or having prior works without PSNR is a valid excuse for calculating PSNR wrong. In fact, for the references provided, [1] and [3] mostly consider (conditional) image generation where there is no ground truth image at all. Reference [2] focuses on inpainting, which is basically image generation as well (conditioned on the region outside) and the measurements do not contain any direct information for the inpainted area. In both cases, the perceptual quality is of more importance so not reporting PSNR makes more sense. However, the authors' work aims to solve inverse problems motivated by "medical imaging, remote sensing, and audio signal processing." In these applications, PSNR is argurably the most commonly used metric. Moreover, if there is a mismatch between the ranges of reconstruction and ground truth, does the range mismatch issue affect other metrics reported in the paper?
>
> ### Additional Comments
>
> I was confused by the previous response due to the code format. Now I can confirm that my original understanding of the implementation was correct. However, the derivation still looks wrong. Using the authors' notation, shouldn't $\operatorname{cov}(v)=\Lambda\_t^{-1}I\Lambda\_t^{-1} = \Sigma\_\text{post}^2$ instead of $\operatorname{cov}(v)=\Lambda\_t^{-1}I\Lambda\_t^{-1} = \Sigma\_\text{post}$? To get the correct covariance, one needs to calculate $v=\Lambda\_t^{-1/2}z$ so that $\operatorname{cov}(v)=\Lambda\_t^{-1/2}I\Lambda\_t^{-1/2} = \Sigma\_\text{post}$. This is why a Cholesky decomposition (or SVD) is unavoidable, which is different from the mean calculation.

---

> > ### Author Response · Authors · 2025-08-07
> > **Response to Reviewer a15Q's Further Comments**
> >
> > # W3 & W4 (Limited Baseline & PSNR)
> > Thank you for pointing this out; we agree that PSNR is standard in many inverse-problem communities, and thus we will complete and report the PSNR values by tomorrow. Now the PSNR values we are obtaining are the in range reported by the other papers.
> >
> > We also note that
> > * LPIPS and FID operate on features extracted after internal normalisation;
> >
> > * Also, our reported SSIM is in the range of other papers.
> >
> > **Please allow us until tomorrow to finalize and update all the PSNR values.**
> >
> > ---
> > # Additional Comments
> >
> > Our earlier reply was a “big-picture” reply. That brief overview was meant only to convey the main idea, and we understand why it causes confusion. Below we give the step-by-step, fully explicit derivation so there is no ambiguity about how the noise term is generated without a dense Cholesky or SVD.
> >
> >
> >
> > $ \Lambda_t =\Sigma_{\text{post}}^{-1} = \tfrac{1-\beta_t}{\beta_t} I+ \tilde A^{\top}\tilde A, \qquad
> > \tilde A = W A, W^\top W=\Sigma_{y|x}^{-1}. $
> >
> > ---
> >
> > ## 1 Generate a random vector whose covariance is $\Lambda_t$
> >
> > ### 1.1 Draw two independent standard Gaussians
> >
> > $$ \varepsilon_1 \sim \mathcal N(0,I_d), \qquad
> > \varepsilon_2 \sim \mathcal N(0,I_m), \qquad
> > \varepsilon_1 \perp \varepsilon_2 . $$
> >
> > ### 1.2 Set
> >
> > $$
> > z = \sqrt{c_t}\,\varepsilon_1 + \tilde A^{\top} \varepsilon_2,
> > \qquad
> > c_t = \tfrac{1-\beta_t}{\beta_t}.
> > $$
> >
> > ### 1.3 Compute $\operatorname{cov}(z)$
> >
> > Because $\varepsilon_1$ and $\varepsilon_2$ are independent and zero-mean,
> >
> > $ \operatorname{cov}(z) = \mathbb E \bigl[z z^{ \top}\bigr] = c_t \mathbb E \bigl[\varepsilon_1\varepsilon_1^{\top}\bigr] +
> >      \tilde A^{\top} \mathbb E \bigl[\varepsilon_2\varepsilon_2^{\top}\bigr]
> >         \tilde A   = c_t I_d + \tilde A^{\top}\tilde A = \boxed{\Lambda_t}.$
> >
> > The cross term $\sqrt{c_t} \mathbb E[\varepsilon_1\varepsilon_2^{\top}] \tilde A$ vanishes because $\varepsilon_1$ and $\varepsilon_2$ are independent.
> >
> > Thus **$z$ is a zero-mean Gaussian with covariance $\Lambda_t$.**
> >
> > ---
> >
> > ## 2 Convert $z$ to the desired covariance via one linear solve
> >
> > We solve the symmetric positive-definite system
> >
> > $$ \Lambda_t v = z \qquad \Longrightarrow \qquad v = \Lambda_t^{-1} z .$$
> >
> > Because solving a linear system is equivalent to left-multiplying by the
> > inverse, $v$ is a linear transformation of $z$.
> > Its covariance is
> >
> > $$ \operatorname{cov}(v) = \Lambda_t^{-1} \operatorname{cov}(z) \Lambda_t^{-1}
> >     = \Lambda_t^{-1} \Lambda_t \Lambda_t^{-1}= \boxed{\Lambda_t^{-1} = \Sigma_{\text{post}}}.$$
> >
> > (we noticed that in our earlier reply, we had a typo and wrote matrix $I$ instead of $\Lambda_t$ when finding $\operatorname{cov}(v)$).
> >
> > Hence $v\sim\mathcal N \bigl(0,\Sigma_{\text{post}}\bigr)$ **without** ever computing $\Sigma_{\text{post}}^{1/2}$.
> >
> > *Implementation:*
> > the linear solve in (3) is carried out by **PW-CG**; five iterations suffice,
> > and each iteration needs exactly one $\tilde A\,u$ and one
> > $\tilde A^{\top}v$ product (no dense factorisation is needed).
> >
> > ---
> >
> > ## 3 Form the posterior sample
> >
> > $$ x_{t-1} = \mu_{\text{post}} + v, \qquad
> > v \sim \mathcal N \bigl(0,\Sigma_{\text{post}}\bigr), $$
> >
> > which is a draw from
> > $\mathcal N \bigl(\mu_{\text{post}},\Sigma_{\text{post}}\bigr)$
> > as required.
> >
> > This completes the derivation and aligns the algorithm, code, and manuscript.

---

> > > ### Comment · Reviewer_a15Q · 2025-08-08
> > >
> > > Thank the authors for the detailed explanation. My concerns have been mostly addressed and I now understand how the method works. Here are some further comments and suggestions for the updated version:
> > > - Now that I understand how the method works, I find line 8 of Alg. 2 and Alg. 3 also confusing because $\boldsymbol{z}$ is sampled to have a covariance of $\boldsymbol{\Sigma}\_\text{post}^{-1}$ rather than $\boldsymbol{I}$. The previous revision of changing line 9 to $\boldsymbol{\mu}\_\text{post}+\boldsymbol{\Sigma}\_\text{post}^{1/2}\boldsymbol{z}$ is also inaccurate because it does not reflect how $\boldsymbol{x}\_{t-1}$ in Alg. 2 (or $\boldsymbol{l}\_{t-1}$ in Alg. 3) is actually sampled.
> > > -  To make the presentation clearer, I strongly suggest that the authors subsitute lines 8 and 9 of Alg. 2 and Alg. 3 into one line like "$\boldsymbol{x}\_{t-1}\leftarrow\text{PW-CG}(\beta\_t, \boldsymbol{\Sigma}\_{\boldsymbol{y}|\boldsymbol{x}}^{-1}, \boldsymbol{x}\_{t}, \boldsymbol{y}\_{t-1}, \boldsymbol{b}\_{t-1}, \boldsymbol{A})$" and provide a separate pseudo-code for $\text{PW-CG}$, given how important this step is in the proposed method. Alternatively, it can be line 8: $\boldsymbol{z}\sim\mathcal{N}(\boldsymbol{0}, \boldsymbol{\Sigma}\_\text{post}^{-1})$ and line 9: $$\boldsymbol{x}\_{t-1}\leftarrow\boldsymbol{\mu}\_\text{post}+\boldsymbol{\Sigma}\_\text{post}\boldsymbol{z}, \qquad \text{See implementation in Alg. X (PW-CG).}$$
> > > It would also make sense to have a brief proposition to show that the proposed implementation can theoretically sample the target Gaussian posterior.
> > >
> > > I would be happy to raise my score if the authors agree to make these edits.

---

> > > > ### Author Response · Authors · 2025-08-08
> > > > **Final Response to Reviewer a15Q**
> > > >
> > > > Thank you for your highly valuable and constructive feedback during the rebuttal phase. Your comments have greatly contributed to improving the quality of our paper.
> > > >
> > > > Below, we first present the PSNR results we promised to share yesterday, followed by a description of the changes we plan to incorporate based on your suggestions.
> > > >
> > > > ---
> > > > # PSNR values
> > > >
> > > > | Method| Inpaint (Rand) | Inpaint (Box) | Deblur (Gauss) | Deblur (Motion) | SR (×4) |
> > > > |-|-|-|-|-|-|
> > > > | DPS| 25.23| 22.51         | 24.25             | 24.92             | 25.86    |
> > > > | DDRM| 9.19| 22.26           | 24.93            | -                 | 26.58    |
> > > > | MCG| 21.57| 19.97           | 6.72             | -                 | 18.20   |
> > > > | ReSample| 25.9| 23.81          | 26.33             | 26.0              | 25.22    |
> > > > | PnP-ADMM| 8.41| 11.65           | 24.93             | -                 | 26.55    |
> > > > | Score-SDE| 13.52| 18.51           | 7.12| -                 | 17.62    |
> > > > | ADMM-TV| 22.03| 17.81           | 22.37| -                 | 23.86     |
> > > > | PnP-DM | 26.2| 22.41           | 25.71| 25.8              | 27.9    |
> > > > | DAPS| 28.33| 24.07           | 29.19| 29.66   | 29.07    |
> > > > | DMPlug| 28.71| **28.92**           | 30.02| **29.91**  | **30.25**    |
> > > > | C-DPS (ours)| **28.95**| 28.69 | **30.13** | 29.85| 30.12   |
> > > >
> > > >
> > > > ---
> > > > # Revised Algorithm 2 & 3
> > > >
> > > >
> > > > Thank you for your helpful suggestion. We fully agree that making the PW-CG–based sampling process explicit will improve clarity and help avoid misinterpretation.
> > > >
> > > > We will replace lines 8–9 with a single line and delegate the sampling procedure to a separate subroutine:
> > > >
> > > >
> > > > ```
> > > > 8   x_{t-1} ← PW-CG(β_t, Σ_{y|x}^{-1}, x_t, y_{t-1}, b_{t-1}, A)
> > > > ```
> > > >
> > > > ---
> > > >
> > > > ###  **New Subroutine: Algorithm X (PW-CG Sampling)**
> > > >
> > > > ```text
> > > > Algorithm X   PW-CG  (draw v ∼ N(0, Σ_post))
> > > > Input : β_t , Σ_{y|x}^{-1}, x_t , y_{t-1}, b_{t-1}, A
> > > >
> > > > 1   Build linear operator:
> > > >         Λ_t u = (1−β_t)/β_t · u + Aᵀ Σ_{y|x}^{−1} A u
> > > >
> > > > 2   Sample:
> > > >         ε₁ ∼ N(0, I_d)  ε₂ ∼ N(0, I_m)  (independent)
> > > >
> > > > 3   Form:
> > > >         z ← √c_t · ε₁ + Aᵀ Σ_{y|x}^{−1/2} ε₂
> > > >         where c_t = (1−β_t)/β_t
> > > >
> > > > 4   Solve:
> > > >         v ← CG-solve(Λ_t, z)            ▹ Λ_t v = z
> > > >
> > > > 5   Return:
> > > >         v                        ▹ v ∼ N(0, Σ_post)
> > > > ```
> > > >
> > > > We will also refer to this algorithm in the main text at the end of §3.4:
> > > >
> > > >
> > > > ---
> > > >
> > > > ### **Proposition (to be added in Appendix B)**
> > > >
> > > > > **Proposition 1.**
> > > > > Let $z$ be drawn by Step 3 of Algorithm X and let $v$ be the unique solution to $\Lambda_t v = z$.
> > > > > Then $v \sim \mathcal{N}(0, \Sigma_{\text{post}})$.
> > > > >
> > > > > **Proof.**
> > > > > From construction, $\operatorname{Cov}(z)=\Lambda_t$.
> > > > > Left-multiplying by $\Lambda_t^{-1}$ yields
> > > > > $\operatorname{Cov}(v)=\Lambda_t^{-1}\Lambda_t\Lambda_t^{-1}   =\Lambda_t^{-1}=\Sigma_{\text{post}}$. ∎
> > > >
> > > >
> > > > ---
> > > >
> > > > We appreciate your thoughtful feedback and believe these changes address all remaining concerns. If so, we kindly ask you to consider updating your score. Thank you once again for your detailed review and helpful suggestions!

---

### Decision · Program_Chairs · 2025-09-17

**Decision:**

Accept (poster)

**Comment:**

This paper concerns the use of diffusion models to solve inverse problems.  The authors propose "coupled data and measurement space DPS" (C-DPS) that introduces a forward stochastic process in the measurement space that evolves in parallel with data-space diffusion, enabling a well-defined posterior distribution.  Empirical results show the advantages of C-DPS.

Four experts reviewed the paper and joined the authors in many rounds of rebuttal and discussion.  Most reviewers felt that the paper was well motivated, clearly written, and novel.  Although several reviewers noted that the submission was missing PSNR results, runtime results, strong recent baselines, a proper justification of certain technical statements, and a thorough discussion of related methods, the rebuttal resolved those concerns.  Therefore, the paper can be accepted on the condition that the authors add the missing information to the camera-ready version.